# Unified Analysis of Continuous Weak Features Learning with Applications to Learning from Missing Data

**Kosuke Sugiyama** [1]  **Masato Uchida** [1]

## Abstract

This paper addresses weak features learning (WFL), focusing on learning scenarios characterized by low-quality input features (weak features; WFs) that arise due to missingness, measurement errors, or ambiguous observations. We present a theoretical formalization and error analysis of WFL for continuous WFs (continuous WFL), which has been insufficiently explored in existing literature. A previous study established formalization and error analysis for WFL with discrete WFs (discrete WFL); however, this analysis does not extend to continuous WFs due to the inherent constraints of discreteness. To address this, we propose a theoretical framework specifically designed for continuous WFL, systematically capturing the interactions between feature estimation models for WFs and label prediction models for downstream tasks. Furthermore, we derive the theoretical conditions necessary for both sequential and iterative learning methods to achieve consistency. By integrating the findings of this study on continuous WFL with the existing theory of discrete WFL, we demonstrate that the WFL framework is universally applicable, providing a robust theoretical foundation for learning with low-quality features across diverse application domains.

## 1. Introduction

The performance and explainability of machine learning models are strongly influenced by the quality of input features. However, in many real-world applications, constraints such as high observation costs, low observation precision, and privacy concerns often result in features deviating from their true values. These deviations manifest as missing values, erroneous observations, or ambiguous information. Such low-quality input features are termed Weak Features (WFs) (Sugiyama & Uchida, 2025). Due to the absence of direct observation of true values, WFs destabilize the learning process of predictive models, leading to performance degradation. Furthermore, the inaccuracies and incompleteness inherent in WFs significantly hinder the explainability of model outputs and decision-making processes. To address these challenges, methods such as impute-then-regress (ItR) (Josse et al., 2024; Bertsimas et al., 2021; Le Morvan et al., 2020a;b; 2021), which imputes missing data prior to prediction, and complementary features learning (CFL) (Sugiyama & Uchida, 2024), which utilizes values differing from the true values, have been proposed. Additionally, a generalized framework, weak features learning (WFL) (Sugiyama & Uchida, 2025), has been developed to systematically address a broader range of WFs.

WFL aims to mitigate the impact of WFs and improve the generalization performance and explainability of predictive models (Sugiyama & Uchida, 2025). A common approach involves constructing *feature estimation models* $g$ to estimate the true values of WFs (termed exact values) from observed features (ordinary features; OFs) that are not WFs, and then learning a *label prediction model* $f$ using both the estimated values for the WFs and the OFs to predict the labels of downstream tasks. In practice, ItR and CFL employ sequential learning, where $g$ and $f$ are optimized consecutively, or iterative learning, where they are optimized alternately (Yoon et al., 2018; Mattei & Frellsen, 2019; Le Morvan et al., 2020a; Ipsen et al., 2021; 2022; Zaffran et al., 2023; Sugiyama & Uchida, 2024).

However, fundamental questions remain unanswered, such as how the interaction between $g$ and $f$ influences learning errors and under what conditions these methods can be guaranteed to be effective. Specifically, these questions can be framed as: (1) How do $g$ and $f$ interact and influence each other's error bound? and (2) What are the precise conditions under which consistency with the optimal hypothesis is theoretically guaranteed? For WFL with all WFs restricted to discrete (discrete WFL), a unified formulation has been

[1]Major in Computer Science and Communications Engineering, Waseda University, Tokyo, Japan. Correspondence to: Kosuke Sugiyama <kohsuke0322@asagi.waseda.jp>, Masato Uchida <m.uchida@waseda.jp>.

proposed, offering systematic answers to these questions via generalization error analysis within a generalized class of learning algorithms (Sugiyama & Uchida, 2025). Discrete WFs include cases such as missing or noisy discrete features, sets that contain the exact discrete value, and observations consisting solely of incorrect discrete values. Discrete WFL is designed to address the presence of such discrete WFs.

In contrast, the theoretical framework for continuous WFL that uniformly addresses continuous WFs, such as missing or noisy continuous values and interval observations that contain the exact continuous value, is not yet well established. This is primarily because the theoretical framework for discrete WFL heavily depends on the mathematical properties of discreteness, making it difficult to directly extend to continuous WFL. Although some theoretical analyses exist for ItR applied to missing data, they lack finite-sample generalization error analyses for arbitrary distributions and fail to accommodate diverse types of continuous WFs (Josse et al., 2024; Bertsimas et al., 2021; Le Morvan et al., 2020a;b; 2021). Additionally, while cases involving erroneous observations (Ristovski et al., 2010; Hu et al., 2020; Berikov & Litvinenko, 2021) or interval-based observations containing exact values (Cheng et al., 2023) have been analyzed in contexts where they are observed as target labels, no analyses have addressed scenarios where such observations serve as input features affecting a downstream task. Therefore, constructing a more general and systematic theoretical framework for continuous WFL remains a critical challenge. Such a framework should not only function equivalently to the discrete WFL framework and provide consistent guarantees across various types of WFs, but also adapt flexibly to arbitrary generative distributions of continuous WFs.

Our main contributions in this paper are the following:

1. We propose a risk-based formulation to unify the treatment of arbitrary continuous WFs. By employing a novel proof technique distinct from that used in discrete WFL, we theoretically demonstrate that the proposed objective function is valid for modeling the true input-output relationship (Section 3.1). This establishes the theoretical justification of our formulation. Based on this, we introduce the *learning algorithm class for continuous WFL (LAC-cWFL)*, a counterpart to LAC-dWFL (Sugiyama & Uchida, 2025). LAC-cWFL offers a consistent theoretical framework for continuous WFs and flexibly incorporates three learning steps: (i) Learning $g$ using WFs as weak supervision, (ii) Learning $f$ with $g$ fixed, and (iii) Learning $g$ with $f$ fixed.

2. We derive new inequalities to analyze the interaction between $g$ and $f$ in continuous WFL (Section 4.1). These inequalities quantitatively characterize how the mutual dependence between $g$ and $f$ influences learning errors, forming a foundation for comprehensively understanding WFL algorithms. In Appendix A, we leverage these inequali-

ties to investigate the theoretical integration of discrete and continuous WFL.

3. We conduct a theoretical investigation into the influence of $g$ on $f$'s learning via the generalization error analysis in step (ii) of LAC-cWFL (Section 4.2). Furthermore, our theoretical framework can be combined with the theories of existing methods for step (i) to establish conditions under which sequential learning, executing (i) and (ii) consecutively, achieves consistency.

4. We Theoretically analyze the influence of $f$ on $g$'s learning in step (iii) of LAC-cWFL (Section 4.3). By integrating the analyses in Contributions 3 and 4, we derive the conditions under which iterative learning, alternating between steps (ii) and (iii), achieves consistency.

5. We extend our analysis to derive analogous theoretical results for ItR in scenarios where all features are continuous and could be missing (Appendix B). This extension establishes a comprehensive foundation for the generalization error analysis of ItR with continuous missing features.

Our results are correspond in parallel with those of discrete WFL (Sugiyama & Uchida, 2025). Consequently, the analyses in this paper demonstrate that the WFL framework (Sugiyama & Uchida, 2025) serves as a universal framework, equally applicable to both discrete and continuous WFs.

## 2. Related work

### 2.1. Review of Ordinary Supervised Learning

This paper builds on the foundational principles of empirical risk minimization (ERM) in supervised learning (Shalev-Shwartz & Ben-David, 2014; Mohri et al., 2018), which we briefly revisit here. Let the input space be defined as $\mathcal{X}^d \in \mathbb{R}^d$ and the label space as $\mathcal{Y} \in \mathbb{R}$. Here, $d \in \mathbb{N}_+$ represents an input dimension. Denote the random variables representing instances by $\boldsymbol{X}$ and the random variable for labels by $Y$. Their realizations $(\boldsymbol{x}, y)$ are assumed to independently follow the true distribution $p_*(\boldsymbol{x}, y)$ over $\mathcal{X} \times \mathcal{Y}$. The goal of ERM is to learn a *label prediction model* $f : \mathcal{X} \to \mathcal{Y} \in \mathcal{F}$ that minimizes the expected risk with respect to a loss function $l : \mathcal{Y} \times \mathcal{Y} \to \mathbb{R}_+$:

$$R_l(f) := \mathbb{E}_{p_*(\boldsymbol{x},y)}[l(f(\boldsymbol{X}), Y)], \quad (2.1)$$

where $\mathcal{F}$ is the hypothesis set of label prediction models. Since $p_*(\boldsymbol{x}, y)$ is unknown, ERM approximates $R_l$ with the empirical risk computed as an average with finite samples and learns $f$ by minimizing this empirical risk.

### 2.2. Discrete WFL

Discrete WFL was formulated as follows (Sugiyama & Uchida, 2025). In WFL, the random variables represent-

ing an instance $\boldsymbol{X}$ are partitioned into the exact values of WFs, denoted by $\boldsymbol{X}^{\mathrm{w}}$, and OFs, denoted by $\boldsymbol{X}^{\mathrm{o}}$, such that $\boldsymbol{X} = (\boldsymbol{X}^{\mathrm{w}}, \boldsymbol{X}^{\mathrm{o}})$. The possible value sets of $\boldsymbol{X}^{\mathrm{w}}$ and $\boldsymbol{X}^{\mathrm{o}}$ are denoted by $\mathcal{X}^{\mathrm{w}} = \prod_{j \in [F^{\mathrm{w}}]} \mathcal{X}_j^{\mathrm{w}}$ and $\mathcal{X}^{\mathrm{o}}$, respectively, where $\mathcal{X}^{\mathrm{w}} \times \mathcal{X}^{\mathrm{o}} = \mathcal{X}$, $F^{\mathrm{w}}$ is the number of WFs, and $[F^{\mathrm{w}}] = \{1, \dots, F^{\mathrm{w}}\}$. Notably, $\mathcal{X}^{\mathrm{w}}$ is a finite set. The observed values of WFs are denoted by the random variables $\overline{\boldsymbol{X}}^{\mathrm{w}}$. Feature estimation models, which estimate the exact values of WFs $\boldsymbol{X}^{\mathrm{w}}$, are defined as $\boldsymbol{g} = (g_1, \dots, g_{F^{\mathrm{w}}}) \in \mathcal{G} = \mathcal{G}_1 \times \cdots \times \mathcal{G}_{F^{\mathrm{w}}} : \mathcal{X}^{\mathrm{o}} \to \mathcal{X}^{\mathrm{w}}$, where $g_j \in \mathcal{G}_j : \mathcal{X}^{\mathrm{o}} \to \mathcal{X}_j^{\mathrm{w}}$ for all $j \in [F^{\mathrm{w}}]$. Here, $\mathcal{G}_j$ represents the hypothesis set for estimating $X_j^{\mathrm{w}}$.

In discrete WFL, where all WFs are discrete, the goal is to minimize the generalization error $R_{l,\boldsymbol{g}}$ of $f$, evaluated using the loss function $l$, and the estimation error $R_{01,j}$ of $g_j$, measured by the 0-1 loss (Sugiyama & Uchida, 2025):

$$R_{l,\boldsymbol{g}}(f) := \mathbb{E}_{p_*(\boldsymbol{x}^{\mathrm{o}}, y)}[l(f(\boldsymbol{g}(\boldsymbol{X}^{\mathrm{o}}), \boldsymbol{X}^{\mathrm{o}}), Y)], \quad (2.2)$$

$$R_{01,j}(g_j) := \mathbb{E}_{p_*(\boldsymbol{x})}[\mathbb{1}_{[g_j(\boldsymbol{X}^{\mathrm{o}}) \neq X_j^{\mathrm{w}}]}], \quad \forall j \in [F^{\mathrm{w}}]. \quad (2.3)$$

Minimizing these risks aims to achieve a highly accurate $f$ and improve the explainability of $f$ by mitigating inaccuracies or ambiguities in the input features. The objective function of discrete WFL is defined as the weighted sum of these risks using a weighting parameter $\lambda \in \mathbb{R}_+$:

$$R_{l,\lambda}^{\mathrm{dWFL}}(f, g) := R_{l,\boldsymbol{g}}(f) + \lambda \sum_{j \in [F^{\mathrm{w}}]} R_{01,j}(g_j). \quad (2.4)$$

It has been shown that for any $l$ bounded by $U_l < \infty$, $f \in \mathcal{F}$ and $\boldsymbol{g} \in \mathcal{G}$, the inequality $R_l(f) \leq R_{l,U_l}^{\mathrm{dWFL}}(f, \boldsymbol{g})$ holds (Theorem 3.1 in (Sugiyama & Uchida, 2025)). This result showed that minimizing $R_{l,\lambda}^{\mathrm{dWFL}}$ contributes to obtaining an $f$ that captures the true input-output relationship, thereby justifying the formulation. This work builds on (Sugiyama & Uchida, 2025) further conducted a theoretical analysis of a generalized learning algorithm class for discrete WFL under the formulation.

However, this analysis expresses the estimation errors of $\boldsymbol{g}$ using the 0-1 loss, rendering the same proof strategy inapplicable to continuous WFL. In this study, we construct a theoretical framework for continuous WFL that accounts for the unique characteristics of continuous WFs. Specifically, we represent the estimation errors of $\boldsymbol{g}$ using the mean squared error (MSE) instead of the 0-1 loss, thereby establishing a theoretical framework for continuous WFL that parallels that of discrete WFL.

### 2.3. Theoretical Analysis for ItR

ItR is a framework that imputes missing values in input features and subsequently performs predictions using the imputed data (Le Morvan et al., 2021). Approaches for handling missing values are generally categorized into two strategies: leveraging the absence of values as informative

signals and estimating the exact values. The former approach is implemented using decision-tree-based methods (Twala et al., 2008; Chen & Guestrin, 2016) or through imputation with placeholder values that signify missingness (Josse et al., 2024). However, when the presence or absence of missingness is independent of $Y$, this approach may not contribute meaningfully to predicting $Y$. In such cases, employing ItR with imputation through a feature estimation model $\boldsymbol{g}$ could become essential.

Several theoretical studies have investigated ItR. Josse et al. demonstrated that in regression or binary classification tasks where only one variable is missing and the missing mechanism is Missing at Random, there exists a predictive function $f$ such that $f \circ \boldsymbol{g}$ serves as the Bayes rule for any constant-value imputation $\boldsymbol{g}$ (Josse et al., 2024) Bertsimas et al., while restricting their analysis to regression tasks, demonstrated that for any almost everywhere continuous function $\boldsymbol{g}$, there exists an $f$ such that $f \circ \boldsymbol{g}$ is the Bayes rule (Bertsimas et al., 2021). Morvan et al. extended the work of Bertsimas et al., providing a comprehensive analysis of the Bayes rule under arbitrary missing mechanisms and any measurable $\boldsymbol{g}$, where all features could potentially be missing (Le Morvan et al., 2021). Morvan et al. also conducted generalization error analysis in the restricted setting where the true regression model is linear (Le Morvan et al., 2020b).

Despite these advances, the aforementioned studies did not investigate learning algorithms capable of asymptotically obtaining the optimal $f \circ \boldsymbol{g}$. Moreover, generalization error analysis under finite sample regimes, applicable to arbitrary data distributions, has yet to be unachieved. In this study, we conduct finite-sample generalization error analysis of ItR for any downstream task. This analysis encompasses situations where any continuous feature can be missing, for any measurable $\boldsymbol{g}$. Through this generalization error analysis, we further investigate the conditions under which sequential or iterative learning methods for $f$ and $\boldsymbol{g}$ achieve consistency. The scenario where only some features may be missing is addressed in Sections 3 and 4, while the case where all features are missing is discussed in Appendix B.

## 3. Formulation

### 3.1. Formulation of continuous WFL

In this section, we present the formulation of continuous WFL from the perspective of risk minimization. Most notations remain consistent with Section 2.2, but two key differences exist. First, in discrete WFL, $\mathcal{X}^{\mathrm{w}}$ is a finite set, whereas in continuous WFL, it is defined as $\mathcal{X}^{\mathrm{w}} \subseteq \mathbb{R}^{F^{\mathrm{w}}}$. Next, we define the probability density function (PDF) associated with the feature estimation models $\boldsymbol{g}$ as $q_{\boldsymbol{g}}(\boldsymbol{x}^{\mathrm{w}} | \boldsymbol{x}^{\mathrm{o}})$. The PDF $q_{\boldsymbol{g}}$ will later be utilized as a probabilistic model derived from the deterministic model $\boldsymbol{g}$. Similar to discrete

WFL (Sugiyama & Uchida, 2025), we focus on binary classification as the downstream task; but, our formulation and analysis can be readily extended to other downstream tasks without losing generality.

The main objectives of WFL are twofold: improving the generalization performance in a downstream task and restoring the explainability lost due to WFs. The loss of explainability stems from the imprecision of information encoded in WFs, $\overline{X}^{\mathrm{w}}$. A natural approach to address this issue is to accurately estimate the exact values of WFs, $X^{\mathrm{w}}$. Therefore, in continuous WFL, it is natural to learn $f$ and $g$ by minimizing the following two risks. The first risk evaluates the generalization error of $f$:

$$
\begin{aligned}
R_{l,\boldsymbol{g}}(f) &:= \mathbb{E}_{p_*(\boldsymbol{x}^{\mathrm{o}},y)q_{\boldsymbol{g}}(\boldsymbol{x}^{\mathrm{w}}|\boldsymbol{x}^{\mathrm{o}})}[l(f(\boldsymbol{X}),Y)] \\
&= \mathbb{E}_{p_*(\boldsymbol{x}^{\mathrm{o}},y)}[l(f(\boldsymbol{g}(\boldsymbol{X}^{\mathrm{o}}),\boldsymbol{X}^{\mathrm{o}}),Y)],
\end{aligned}
\tag{3.5}
$$

The second risk measures the estimation errors of $g$:

$$
R_{\mathrm{MSE},j}(g_j) := \mathbb{E}_{p_*(\boldsymbol{x})}[l_{\mathrm{MSE}}(g_j(\boldsymbol{X}^{\mathrm{o}}),X_j^{\mathrm{w}})], \forall j \in [F^{\mathrm{w}}],
\tag{3.6}
$$

where $l_{\mathrm{MSE}}(y,y') := (y-y')^2$ represents the mean squared error (MSE). The objective function for continuous WFL is defined as a linear combination of these two risks using a weight parameter $\lambda \in \mathbb{R}_+$:

$$
R_{l,\lambda}^{\mathrm{cWFL}}(f,g) := R_{l,\boldsymbol{g}}(f) + \lambda \sum_{j \in [F^{\mathrm{w}}]} R_{\mathrm{MSE},j}(g_j).
\tag{3.7}
$$

The objective function $R_{l,\lambda}^{\mathrm{cWFL}}$ facilitates the unified treatment of any continuous WFs. This is because the error of $g_j$ is defined through the risk $R_{\mathrm{MSE},j}$ to be minimized irrespective of the type of WF. For various types of continuous WFs, weakly supervised learning methods that minimize $R_{\mathrm{MSE},j}$ using $\overline{X}_j^{\mathrm{w}}$ as weak supervision have already been established (Wasserman & Lafferty, 2007; Li et al., 2017; Kostopoulos et al., 2018; Ristovski et al., 2010; Hu et al., 2020; Berikov & Litvinenko, 2021; Cheng et al., 2023). Therefore, to minimize $R_{l,\lambda}^{\mathrm{cWFL}}$, these WSL methods can be employed to learn $g_j$ based on the type of WF.

Moreover, $R_{l,\lambda}^{\mathrm{cWFL}}$ can be interpreted as a natural formulation, where $R_{01,j}$ replaced by $R_{\mathrm{MSE},j}$ in $R_{l,\lambda}^{\mathrm{dWFL}}$ defined in Eq. (2.4). However, justifying this formulation for continuous WFL is non-trivial. The justification for discrete WFL relies on the inequality derived using the properties of $R_{01,j}$ (Theorem 3.1 in (Sugiyama & Uchida, 2025)), which cannot be directly applied to continuous WFL. In the case of discrete WFL, the key approach involved expressing the upper bound of $R_l(f) - R_{l,\boldsymbol{g}}(f)$ in terms of $R_{01,j}(g_j)$. In this work, we successfully derive the upper bound of $R_l(f) - R_{l,\boldsymbol{g}}(f)$, expressed in terms of $R_{\mathrm{MSE},j}(g_j)$. The validity of our formulation is demonstrated by the following theorem. The proof can be found in Appendix C.1.

**Theorem 3.1.** *Let $\sigma \in \mathbb{R}_+$, and define $q_{\boldsymbol{g}}$ as the PDF of $\mathcal{N}(\boldsymbol{g}(\boldsymbol{x}^{\mathrm{o}}), \sigma^2 I_{F^{\mathrm{w}} \times F^{\mathrm{w}}})$. For any measurable $f \in \mathcal{F}$, $g \in \mathcal{G}$, and $l$ bounded by $U_l < \infty$, the following holds:*

$$
\begin{aligned}
R_l(f) &\leq \\
&R_{l,\boldsymbol{g}}(f) + U_l\big\{C_\sigma + \tfrac{1}{2\sigma^2}\sum_{j \in [F^{\mathrm{w}}]} R_{\mathrm{MSE},j}(g_j)\big\}^{\frac{1}{2}}.
\end{aligned}
\tag{3.8}
$$

Here, $C_\sigma := \mathbb{E}_{p_*(\boldsymbol{x},y)}[\log p_*(\boldsymbol{X}^{\mathrm{w}}|\boldsymbol{X}^{\mathrm{o}},Y)] + \log\sqrt{(2\pi)^{F^{\mathrm{w}}}\sigma^{2F^{\mathrm{w}}}}$. The definition of $q_{\boldsymbol{g}}$ corresponds to adding Gaussian noise with mean 0 and variance $\sigma^2$ to the output of $g$. This definition imposes no constraints on the learning of $g$, thus preserving the generality of our results. According to the definition of $C_\sigma$, when for any $j \in [F^{\mathrm{w}}]$, $\mathcal{X}_j^{\mathrm{w}}$ is a set of finite-precision decimals (in this case, $\mathbb{E}_{p_*(\boldsymbol{x},y)}[\log p_*(\boldsymbol{X}^{\mathrm{w}}|\boldsymbol{X}^{\mathrm{o}},Y)] \leq 0$ holds), and $q_{\boldsymbol{g}}$ is defined with $\sigma^2 = (2\pi)^{-1}$, it follows that $C_\sigma \leq 0$. Thus, under this mild assumption, $C_\sigma$ can be disregarded. The LHS of Eq. (3.8) quantifies how well $f$ captures the true input-output relationship. Thus, Theorem 3.1 establishes that minimizing the two components of our objective function ensures that $f$ effectively captures the true input-output information.

### 3.2. Learning Algorithm Class for continuous WFL

In this section, we introduce a unified learning algorithm class capable of handling not only arbitrary continuous WFs but also various methods in continuous WFL. Under the proposed formulation in Section 3.1, we define the following learning algorithm class:

**Definition 3.2** (LAC-cWFL). The *learning algorithm class for continuous WFL (LAC-cWFL)* refers to the set of algorithms that learn the feature estimation models $g$ and label prediction model $f$ through any combination of the following three steps:

(i) Learning $g$ by using $\overline{X}^{\mathrm{w}}$ as weak supervision and minimizing $\sum_{j \in [F^{\mathrm{w}}]} R_{\mathrm{MSE},j}$, either directly or indirectly.

(ii) Fixing $g$ and learning $f$ by minimizing $R_{l,\boldsymbol{g}}$.

(iii) Fixing $f$ and learning $g$ by minimizing $R_{l,\lambda}^{\mathrm{cWFL}}$.

LAC-cWFL acts as a continuous counterpart to LAC-dWFL (Sugiyama & Uchida, 2025). Similar to LAC-dWFL, LAC-cWFL unifies a broad range of methods, such as *sequential learning* (comprising steps (i) and (ii)) and *iterative learning* (comprising steps (ii) and (iii)) (Yoon et al., 2018; Mattei & Frellsen, 2019; Ipsen et al., 2021; Josse et al., 2024; Le Morvan et al., 2020a; 2021; Ipsen et al., 2022; Sugiyama & Uchida, 2025). Furthermore, various weakly supervised learning methods can be applied to step (i) by utilizing $\overline{X}^{\mathrm{w}}$ as weak supervision (Wasserman & Lafferty, 2007; Li et al., 2017; Kostopoulos et al., 2018; Ristovski et al., 2010;

Hu et al., 2020; Berikov & Litvinenko, 2021; Cheng et al., 2023). In Section 4, we perform a generalization error analysis of this unified class of learning algorithms, revealing the mutual influences between $g$ and $f$ during learning and the conditions under which WFL attains optimal models.

## 4. Theoretical analysis

### 4.1. Construction of an Analytical Tool

The formulation introduced in Section 3 and LAC-cWFL correspond to those of discrete WFL (Sugiyama & Uchida, 2025). However, determining whether similar theoretical analyses can be applied to them, as in the case of discrete WFL, is not straightforward. This difficulty arises because the analytical method for discrete WFL is derived using the properties of $R_{01,j}$, and the method cannot be applied directly to continuous WFL. In discrete WFL, deriving an inequality that expresses the upper bound of $|R_l(f) - R_{l,\boldsymbol{g}}(f)|$ in terms of $R_{01,j}(g_j)$ and $R_l(f)$ enabled discussions on how $\boldsymbol{g}$ influences the learning of $f$ and vice versa (Lemma 4.1 in (Sugiyama & Uchida, 2025)). Therefore, for continuous WFL, it is essential to identify an inequality that expresses the upper bound of $|R_l(f) - R_{l,\boldsymbol{g}}(f)|$ in terms of $R_{\text{MSE},j}(g_j)$ and $R_l(f)$. The challenge is distinct from Theorem 3.1, as the upper bound must explicitly depend on $R_l(f)$. Equation (3.8) in Theorem 3.1 does not express this dependency.

We derive an inequality that resolves this issue. First, we establish the following lemma, which is valid within a more general framework that is independent of the definition of $q_{\boldsymbol{g}}$ [1]. The proof is provided in Appendix C.2.

**Lemma 4.1.** *For any measurable $f \in \mathcal{F}$, $q_{\boldsymbol{g}}(\boldsymbol{x}^{\text{w}}|\boldsymbol{x}^{\text{o}})$ and $l$ bounded by $U_l < \infty$, the following inequality holds:*

$$|R_l(f) - R_{l,\boldsymbol{g}}(f)| \le \left(\sqrt{R_l(f)} + \sqrt{R_{l,\boldsymbol{g}}(f)}\right) \times$$
$$\left\{2U_l \mathbb{E}_{p_*(\boldsymbol{x}^{\text{o}},y)}\left[D_{\text{H}}^2(p_*(\boldsymbol{X}^{\text{w}}|\boldsymbol{X}^{\text{o}},Y), q_{\boldsymbol{g}}(\boldsymbol{X}^{\text{w}}|\boldsymbol{X}^{\text{o}}))\right]\right\}^{\frac{1}{2}}. \quad (4.9)$$

*Here, $D_{\text{H}}$ denotes the Hellinger distance.*

Although Lemma 4.1 has a broad scope, one of the our goals is analyzing the relationship between the estimation errors of $\boldsymbol{g}$, represented by $\sum_{j \in [F^{\text{w}}]} R_{\text{MSE},j}(g_j)$, and $f$, with the goal of constructing a theory that parallels discrete WFL. To this end, we define $q_{\boldsymbol{g}}$ as the PDF of $\mathcal{N}(\boldsymbol{g}(\boldsymbol{x}^{\text{o}}), \sigma^2 I_{F^{\text{w}} \times F^{\text{w}}})$ and derive the following lemma as

---

[1] Lemma 4.1 is valid for both discrete and continuous WFs, and it remains applicable even when the feature estimation models are probabilistic. However, this lemma cannot directly be used to derive the results of discrete WFL presented in (Sugiyama & Uchida, 2025). In contrast, the lemma has the potential to enable a unified analysis of discrete WFL and continuous WFL. This direction is explored in Appendix A.

a specific case of Lemma 4.1. The proof is provided in Appendix C.3.

**Lemma 4.2.** *Let $q_{\boldsymbol{g}}$ be the PDF of $\mathcal{N}(\boldsymbol{g}(\boldsymbol{x}^{\text{o}}), \sigma^2 I_{F^{\text{w}} \times F^{\text{w}}})$, where $\sigma^2 \in \mathbb{R}_+$. For any measurable $f \in \mathcal{F}$, $\boldsymbol{g} \in \mathcal{G}$ and $l$ bounded by $U_l < \infty$, the following inequality holds:*

$$|R_l(f) - R_{l,\boldsymbol{g}}(f)| \le \left(\sqrt{R_l(f)} + \sqrt{R_{l,\boldsymbol{g}}(f)}\right)$$
$$\times \left\{2U_l\left(C_\sigma + \tfrac{1}{2\sigma^2} \sum_{j \in [F^{\text{w}}]} R_{\text{MSE},j}(g_j)\right)\right\}^{\frac{1}{2}}. \quad (4.10)$$

Equation (4.10) of Lemma 4.2 corresponds to the inequality in discrete WFL, which expresses the upper bound of $|R_l(f) - R_{l,\boldsymbol{g}}(f)|$ in terms of $R_{01,j}(g_j)$ and $R_l(f)$ (Eq.(4.6) in (Sugiyama & Uchida, 2025)). This parallel suggests that Eq. (4.10) provide a sufficient foundation for deriving analogous analyses to those in discrete WFL. Using Lemma 4.2, we theoretically elucidate the mutual influence of $\boldsymbol{g}$ and $f$ in LAC-cWFL. In Section 4.2, we analyze the learning of $f$ under LAC-cWFL's step (ii), whereas in Section 4.3, we focus on the learning of $\boldsymbol{g}$ under the step (iii).

### 4.2. Analysis of Learning Label Prediction Model $f$

This section examines the learning of $f$ in LAC-cWFL's step (ii). As $f$ in WFL utilizes the output of $\boldsymbol{g}$ as input, the learning of $f$ inherently depends on $\boldsymbol{g}$. We theoretically investigate how the estimation errors of $\boldsymbol{g}$ influence the learning of $f$.

For theoretical analysis, we introduce the following definitions. Let $S := \{(\boldsymbol{x}_i, y_i)\}_{i=1}^n$ denote an *ordinary dataset* and $\overline{S} := \{(\bar{\boldsymbol{x}}_i^{\text{w}}, \boldsymbol{x}_i^{\text{o}}, y_i)\}_{i=1}^n$ a *weak dataset*, where both datasets contain $n \in \mathbb{N}_+$ samples, and the $i$-th sample in $S$ and $\overline{S}$ corresponds to the same instance, for any $i \in [n]$. Let $\widehat{R}_l$ and $\widehat{R}_{l,\boldsymbol{g}}$ denote the empirical risks computed by sample average over $S$ and $\overline{S}$, respectively. For any $\boldsymbol{g} \in \mathcal{G}$, the empirical risk minimizer of LAC-cWFL's step (ii) is defined as follows:

$$f_{\boldsymbol{g},\overline{S}} := \arg\min_{f \in \mathcal{F}} \widehat{R}_{l,\boldsymbol{g}}(f).$$

Using Lemma 4.2, the error bound for $f_{\boldsymbol{g},\overline{S}}$ learned in LAC-cWFL's step (ii) is established in the following theorem. The proof is provided in Appendix C.4.

**Theorem 4.3.** *Suppose $S$ and $\overline{S}$ are ordinary and weak datasets consisting of $n$ samples, respectively. Let $q_{\boldsymbol{g}}$ be the PDF of $\mathcal{N}(\boldsymbol{g}(\boldsymbol{x}^{\text{o}}), \sigma^2 I_{F^{\text{w}} \times F^{\text{w}}})$, where $\sigma^2 \in \mathbb{R}_+$. For any measurable $\boldsymbol{g} \in \mathcal{G}$, $L_l$-Lipschitz continuous $l$ bounded by $U_l < \infty$ and $\delta \in (0, 1)$, the following inequality holds with*

*probability at least* $1 - \delta$:

$$R_{l,\boldsymbol{g}}(f_{\boldsymbol{g},\overline{S}}) - R_l(f_{\mathcal{F}}) \leq$$

$$4\Big(L_l\mathfrak{R}_n^*(\mathcal{F}) + L_l\mathfrak{R}_n^{\boldsymbol{g}}(\mathcal{F}) + U_l\sqrt{\tfrac{\log(4/\delta)}{2n}}\Big)$$

$$+ \Big\{2\Big(R_l(f_{\mathcal{F}}) + 4L_l\mathfrak{R}_n^*(\mathcal{F}) + 2U_l\sqrt{\tfrac{\log(4/\delta)}{2n}}\Big)^{\frac{1}{2}}$$

$$+ \Big(2U_l\Big(C_\sigma + \tfrac{1}{2\sigma^2}\sum_{j\in[F^{\mathrm{w}}]} R_{\mathrm{MSE},j}(g_j)\Big)\Big)^{\frac{1}{2}}\Big\}$$

$$\times \Big\{2U_l\Big(C_\sigma + \tfrac{1}{2\sigma^2}\sum_{j\in[F^{\mathrm{w}}]} R_{\mathrm{MSE},j}(g_j)\Big)\Big\}^{\frac{1}{2}},$$

(4.11)

*where* $f_{\mathcal{F}} := \arg\min_{f\in\mathcal{F}} R_l(f)$ *represents the true risk minimizer in ordinary supervised learning.*

Here, $\mathfrak{R}_n^*(\mathcal{F})$ and $\mathfrak{R}_n^{\boldsymbol{g}}(\mathcal{F})$ denote the Rademacher complexities associated with the distributions $p_*(\boldsymbol{x})$ and $p_*(\boldsymbol{x}^{\mathrm{o}})q_{\boldsymbol{g}}(\boldsymbol{x}^{\mathrm{w}}|\boldsymbol{x}^{\mathrm{o}})$, respectively, and represent the complexity of $\mathcal{F}$. For kernel ridge regression and multilayer perceptrons, the order of the Rademacher complexity's upper bound is $\mathcal{O}_p(1/n^{1/2})$ (Mohri et al., 2018; Neyshabur et al., 2015). In the subsequent discussions, $\mathfrak{R}_n^*(\mathcal{F})$ and $\mathfrak{R}_n^{\boldsymbol{g}}(\mathcal{F})$ are assumed to have this order. Additionally, we assume that $\mathcal{F}$ is sufficiently expressive and that $R_l(f_{\mathcal{F}}) = 0$. Given that $\mathcal{X}_j^{\mathrm{w}}$ is a set of finite decimal numbers, for any $j \in [F^{\mathrm{w}}]$ and $\sigma^2 = (2\pi)^{-1}$, it follows that $C_\sigma \leq 0$. Therefore, under this mild assumption, $C_\sigma$ will be ignored in the subsequent discussion.

Theorem 4.3 establishes an error bound for learning $f$ under continuous WFL, which parallels the result for discrete WFL (Theorem 4.2 in (Sugiyama & Uchida, 2025)). This demonstrates that our framework successfully achieved results for learning $f$ in continuous WFL, analogous to those in discrete WFL. The main contributions of Theorem 4.3 are twofold:

The first contribution of Theorem 4.3 lies in elucidating how the estimation errors of fixed $\boldsymbol{g}$, measured by the MSE, influence the convergence rate of $f$'s generalization error $R_{l,\boldsymbol{g}}(f)$ with respect to the number of training samples $n$. Specifically, the orders of the first and second terms on the RHS of Eq. (4.11) are $\mathcal{O}_p(1/n^{1/2})$ and $\mathcal{O}_p(1/n^{1/4})$, respectively. Therefore, when $\sum_{j\in[F^{\mathrm{w}}]} R_{\mathrm{MSE},j}(g_j)$ is large, the second term, which decreases more slowly, becomes dominant.

The second contribution of Theorem 4.3 lies in its ability to combine Eq. (4.11) with the theoretical bounds for learning $\boldsymbol{g}$ in LAC-cWFL's step (i), thereby enabling an analysis of the effect of $\boldsymbol{g}$'s learning on $f$'s learning and the asymptotic properties of sequential learning. For example, in the case of WFs with missing data, if $\hat{g}_j$ is obtained as an empirical risk minimizer for $R_{\mathrm{MSE},j}$, an error bound of $R_{\mathrm{MSE},j}(\hat{g}_j)$ can be derived (e.g., Theorem 11.3 in (Mohri et al., 2018)) and subsequently incorporated into Eq. (4.11). This com-

bination demonstrates that learning $\boldsymbol{g}$ improves the error bound's order from $\mathcal{O}_p(1/n^{1/4})$ to $\mathcal{O}_p(1/(nm)^{1/4})$, where $m \leq n$ is the number of samples available for learning $\boldsymbol{g}$. Consequently, learning $\boldsymbol{g}$ improves convergence rates.

Additionally, this combination enables a consistent analysis of sequential learning as represented by LAC-cWFL's steps (i) and (ii), thereby elucidating conditions for achieving consistency as follows theorem. This theorem shows that sequential learning is asymptotically sufficient under these conditions. The proof is provided in Appendix C.5.

**Theorem 4.4.** *Suppose that $\mathcal{X}_j^{\mathrm{w}}$ is a set of finite-precision decimals, for any $j \in [F^{\mathrm{w}}]$. Assume the existence of true deterministic functions $g_j^* : \mathcal{X}^{\mathrm{o}} \to \mathcal{X}_j^{\mathrm{w}}$ exist for any $j \in [F^{\mathrm{w}}]$, such that $(g_1^*, \ldots, g_{F^{\mathrm{w}}}^*) \in \mathcal{G}$ and $f^* : \mathcal{X} \to \mathcal{Y}$ such that $f^* \in \mathcal{F}$. Furthermore, assume that $l$ bounded by $U_l < \infty$ is $L_l$-Lipschitz continuous, that $\mathfrak{R}_n^*(\mathcal{F})$ and $\mathfrak{R}_n^{\boldsymbol{g}}(\mathcal{F})$ asymptotically converge to $0$ as $n \to \infty$, and that for all $j \in [F^{\mathrm{w}}]$, the number of samples available for learning $g_j$ tends to infinity as $n \to \infty$. Then, using consistent methods for learning $\boldsymbol{g}$ and settings $\sigma^2 = (2\pi)^{-1}$ for $q_{\boldsymbol{g}}$, sequential learning under continuous WFL is consistent (i.e. as $n \to \infty$, $R_{l,\boldsymbol{g}}(f_{\boldsymbol{g},\overline{S}}) \to R_l(f_{\mathcal{F}})$).*

### 4.3. Analysis of Learning Feature Estimation Models $\boldsymbol{g}$

This section examines the learning of $\boldsymbol{g}$ in the context of LAC-cWFL's step (iii). We aim to elucidate how the generalization performance and properties of $f$ influence the learning of $\boldsymbol{g}$ when the objective function $R_{l,\lambda}^{\mathrm{cWFL}}$ is minimized with $f$ fixed.

LAC-cWFL's step (iii) focuses on minimizing the two terms on the RHS of Eq. (3.7) with respect to $\boldsymbol{g}$. This requires a distinct analytical approach compared to the single-risk minimization in Section 4.2. In the analysis of discrete WFL (Sugiyama & Uchida, 2025), structural risk minimization (SRM) was employed to handle the minimization of multiple terms (Mohri et al., 2018). To develop a theoretical framework of continuous WFL that corresponds in parallel with that of discrete WFL, we analyze the learning of $\boldsymbol{g}$ using SRM.

As a preparation, for any $j \in [F^{\mathrm{w}}]$, we introduce the following definitions according to the theory of discrete WFL (Sugiyama & Uchida, 2025). Let $l_j$ denote the loss function for $g_j$, calculated using $\overline{X}_j^{\mathrm{w}}$. Define the datasets $\overline{S}_j := \{(\bar{x}_{ij}^{\mathrm{w}}, \boldsymbol{x}_i^{\mathrm{o}})\}_{i=1}^n$. Let $\overline{R}_{l_j}(g_j) := \mathbb{E}_{p_*(\boldsymbol{x},y)\bar{p}_*(\bar{x}_j^{\mathrm{w}}|\boldsymbol{x},y)}[l_j(g_j(\boldsymbol{X}^{\mathrm{o}}), \overline{X}_j^{\mathrm{w}}))]$ represent the expected risk of $g_j$ computed using $\overline{X}_j^{\mathrm{w}}$ and $\widehat{\overline{R}}_{l_j}$ represent the empirical risk, which approximates $\overline{R}_{l_j}$ by taking the sample average over $\overline{S}_j$. We assume that $\overline{R}_{l_j}$ satisfies either $R_{01,j}(g_j) = \overline{R}_{l_j}(g_j)$ or $R_{01,j}(g_j) \leq \overline{R}_{l_j}(g_j)$ for any $g_j$, or that the optimal solutions of $\overline{R}_{l_j}$ equal those of $R_{01,j}$.

For any $\boldsymbol{r} = (r_1, \ldots, r_{F^{\mathrm{w}}}) \in \mathbb{R}_+^{F^{\mathrm{w}}}$ define the following hypothesis set:

$$\mathcal{G}(\boldsymbol{r}, \overline{S}) = \mathcal{G}_1(r_1, \overline{S}_1) \times \cdots \times \mathcal{G}_{F^{\mathrm{w}}}(r_{F^{\mathrm{w}}}, \overline{S}_{F^{\mathrm{w}}}),$$

where, $\mathcal{G}_j(r_j, \overline{S}_j) := \{g_j | g_j \in \mathcal{G}_j \wedge \widehat{\overline{R}}_{l_j}(g_j) \le r_j\}, \forall j \in [F^{\mathrm{w}}]$. In this section, $R_{l,f}(f)$ is expressed as $R_{l,f}(\boldsymbol{g})(\equiv R_{l,\boldsymbol{g}}(f))$ to emphasize the optimization with respect to $\boldsymbol{g}$. The empirical risk of $R_{l,f}(\boldsymbol{g})$ is described as $\widehat{R}_{l,f}(\boldsymbol{g})(\equiv \widehat{R}_{l,\boldsymbol{g}}(f))$. Using these definitions, the learning of $\boldsymbol{g}$ in LAC-cWFL's step (iii) is expressed as follows:

$$\boldsymbol{g}_{f,\overline{S}}^{(\boldsymbol{r})} = \arg \min_{\boldsymbol{g} \in \mathcal{G}(\boldsymbol{r}, \overline{S})} \widehat{R}_{l,f}(\boldsymbol{g}). \tag{4.12}$$

By applying Lemma 4.2, we derive the following error bound for $\boldsymbol{g}_{f,\overline{S}}^{(\boldsymbol{r})}$. The proof is provided in Appendix C.6.

**Theorem 4.5.** *Suppose $S$ and $\overline{S}$ are ordinary and weak datasets consisting of $n$ samples, respectively. Let $q_{\boldsymbol{g}}$ be the PDF of $\mathcal{N}(\boldsymbol{g}(\boldsymbol{x}^{\mathrm{o}}), \sigma^2 I_{F^{\mathrm{w}} \times F^{\mathrm{w}}})$, where $\sigma^2 \in \mathbb{R}_+$. For any measurable label prediction model $f \in \mathcal{F}$, $l$ bounded by $U_l < \infty$ and $\delta \in (0, 1)$, the following inequality holds with probability at least $1 - \delta$:*

$$R_{l,f}(\boldsymbol{g}_{f,\overline{S}}^{(\boldsymbol{r})}) - R_l(f) \le$$
$$\left( 4\mathfrak{R}_n^*(\widetilde{\mathcal{G}}_{l,f}(\boldsymbol{r}, \overline{S})) + 2U_l \sqrt{\tfrac{\log(2/\delta)}{2n}} \right) + \left\{ 2\sqrt{R_l(f)} \right.$$
$$\left. + \left( 2U_l \big( C_\sigma + \tfrac{1}{2\sigma^2} \sum_{j \in [F^{\mathrm{w}}]} R_{\mathrm{MSE},j}(g_{\overline{S},j}^{(r_j)}) \big) \right)^{\frac{1}{2}} \right\}$$
$$\times \left\{ 2U_l \big( C_\sigma + \frac{1}{2\sigma^2} \sum_{j \in [F^{\mathrm{w}}]} R_{\mathrm{MSE},j}(g_{\overline{S},j}^{(r_j)}) \big) \right\}^{\frac{1}{2}}. \tag{4.13}$$

*Here, $g_{\mathcal{G}(\boldsymbol{r}, \overline{S}),j} := \arg\min_{g_j \in \mathcal{G}_j(r_j, \overline{S}_j)} R_{\mathrm{MSE},j}(g_j)$ and $\widetilde{\mathcal{G}}_{l,f}(\boldsymbol{r}, \overline{S}) := \{(\boldsymbol{x}^{\mathrm{o}}, y) \mapsto l(f(\boldsymbol{g}(\boldsymbol{x}^{\mathrm{o}}), \boldsymbol{x}^{\mathrm{o}}), y) : \boldsymbol{g} \in \mathcal{G}(\boldsymbol{r}, \overline{S})\}$.*

The term $R_{\mathrm{MSE},j}(g_{\overline{S},j}^{(r_j)})$ in Eq. (4.13) can be further upper-bounded by defining $\mathcal{G}_j(r_j, \overline{S}_j)$ as the set of empirical risk minimizers obtained through a learning method using $\overline{S}_j$. Consider, for instance, the case where $\overline{X}_j^{\mathrm{w}}$ denotes features with missing values, and the missingness mechanism is missing completely at random (MCAR). Assume further that $\mathcal{G}_j$ is sufficiently expressive such that $\min_{g_j \in \mathcal{G}_j} R_{\mathrm{MSE},j}(g_j) = 0$, and that $\mathcal{Y}$ is a bounded interval such that $l_{\mathrm{MSE}} \le U_{l_{\mathrm{MSE}}}$. Suppose that $\mathcal{G}_j(r_j, \overline{S}_j)$ is defined as the set of empirical risk minimizers obtained using only the samples for which $\overline{X}_j^{\mathrm{w}}$ is observed, where these samples are independently and identically distributed. Then, for any $\delta \in (0, 1)$, $R_{\mathrm{MSE},j}(g_{\overline{S},j}^{(r_j)})$ can be upper bounded with probability at least $1 - \delta$, as follows (Mohri et al.,

2018):

$$R_{\mathrm{MSE},j}(g_{\overline{S},j}^{(r_j)}) \le 8U_{l_{\mathrm{MSE}}} \mathfrak{R}_{n_j'}^*(\mathcal{G}_j) + 2U_{l_{\mathrm{MSE}}} \sqrt{\tfrac{\log(1/\delta)}{2n_j'}}, \tag{4.14}$$

where $n_j'$ denotes the number of training samples used to obtain $g_{\overline{S},j}^{(r_j)}$. Under the assumption that the missingness mechanism is MCAR, it follows that $n_j' \to \infty$ as $n \to \infty$. Details are provided in Appendix C.7. Accordingly, if $\mathcal{G}_j(r_j, \overline{S}_j)$ is defined as the set of ERM solutions obtained by such a method achieving consistency, and if the order of $\mathfrak{R}_n^*(\mathcal{G}_j)$ is $\mathcal{O}_p(1/n^{1/2})$, then the order of the upper bound of $R_{\mathrm{MSE},j}(g_{\overline{S},j}^{(r_j)})$ is also $\mathcal{O}_p(1/n^{1/2})$. Moreover, if $\mathfrak{R}_n^*(\mathcal{G}_j) \to 0$ as $n \to \infty$, then it follows that $R_{\mathrm{MSE},j}(g_{\overline{S},j}^{(r_j)}) \to 0$ as $n \to \infty$. In the following discussion, we assume that $R_{\mathrm{MSE},j}(g_{\overline{S},j}^{(r_j)})$ can be upper bounded by a probability inequality of the same form as Eq.(4.14).

Similar to the analysis of $f$'s learning in Section 4.2, Theorem 4.5 yields results that closely align with those of discrete WFL (Theorem 4.4 in (Sugiyama & Uchida, 2025)). This confirms that the analysis of $\boldsymbol{g}$ learning in LAC-cWFL's step (iii) aligns with its counterpart in discrete WFL.

Theorem 4.5 elucidates how the generalization performance and properties of $f$ affect the learning of $\boldsymbol{g}$ in continuous WFL. Equation (4.13) expresses the effect of $f$'s generalization error $R_l(f)$ on the convergence rate of $R_{l,f}$'s error bound with $n$ training samples. Specifically, the orders of the first and second terms on the RHS of Eq. (4.13) are $\mathcal{O}_p(1/n^{1/2})$ and $\mathcal{O}_p(1/n^{1/4})$, respectively. Therefore, as $R_l(f)$ increases, the slower-decreasing second term dominates the reduction rate. Although $R_l(f)$ affects the learning of $\boldsymbol{g}$ and cannot be minimized directly, Theorem 3.1 demonstrates that the two terms in $R_{l,\lambda}^{\mathrm{cWFL}}$, as defined in Eq. (3.7), contribute to the minimization $R_l(f)$. Consequently, a reduction in $R_l(f)$ can be expected as a result of the iterative learning process.

Furthermore, Theorems 4.3 and 4.5 reveal the conditions under which the iterative learning composed of LAC-cWFL's steps (ii) and (iii) achieves consistency. The proof is provided in Appendix C.8.

**Theorem 4.6.** *In addition to the conditions stated in Theorem 4.4, assume that $f$ obtained in LAC-cWFL's step (ii) is Lipschitz continuous. Furthermore, for any $j \in [F^{\mathrm{w}}]$, define $\mathcal{G}_j(r_j, \overline{S}_j)$ as the set of empirical risk minimizers obtained by methods that use $\overline{S}_j$ and are guaranteed to achieve consistency. Moreover, suppose the Rademacher complexities about $\boldsymbol{g}$ converge to 0 as $n \to \infty$. Under these conditions, the iterative learning composed of LAC-cWFL's steps (ii) and (iii) achieves consistency.*

In the preceding discussion, for the sake of simplicity, the

variance $\sigma^2$ of $q_{\boldsymbol{g}}$ was defined as a constant independent of $\boldsymbol{x}^\circ$. In contrast, it is also possible to define this variance as a function $\sigma^2(\boldsymbol{x}^\circ)$. In this case, for Theorem 3.1, Lemma 4.2, Theorem 4.3, and Theorem 4.5, the inequalities presented in each result continue to hold when $\sigma^2$ in the upper bounds is replaced by $\max_{x^\circ} \sigma^2(x^\circ)$.

## 5. Experiments

In Section 4, it was theoretically demonstrated that sequential learning alone is sufficient for continuous WFL. In this section, we investigate whether the error bound for $f$, derived in Theorem 4.3, can adequately explain the relationship between the number of training samples $n$ and the generalization error $R_{l,\boldsymbol{g}}(f_{\boldsymbol{g},\overline{S}})$ in practical tasks. We adopt the experimental methodology of discrete WFL (Sugiyama & Uchida, 2025) to examine the relationship between them across various estimation errors for $\boldsymbol{g}$.

### 5.1. Experimental Settings

For real-world datasets, we utilize four datasets from OpenML (Vanschoren et al., 2013): *hls4ml_lhc_jets_hlf* (Pierini et al., 2020), *electricity* (Harries et al., 2014), *mv* (Luis, 2014), and *Run_or_walk_information* (Viktor, 2017). We refer to them as *Jets*, *Electricity*, *Mv*, and *Run-or-Walk*, respectively. Details of these datasets are summarized in Appendix F.1. Half of the samples in each dataset were used as test data to estimate the expected risk. This experiment focused on scenarios involving OFs alongside continuous features with missing values as WFs. Here, the missing mechanism was assumed to be *Missing Completely At Random*, with a missing rate of 50% for any WFs. For $\boldsymbol{g}$ and $f$, we used two-layer perceptrons with hidden layers of width 500 and ReLU as an activation function, following the experimental setup of discrete WFL (Sugiyama & Uchida, 2025). Logistic loss is employed as $l$. Details of the experimental settings are summarized in Appendix F.1. The experimental results shown below are the average of 5 trials. The experimental scripts used in this paper are available at the following URL: https://github.com/KOHsEMP/continuous_WFL

### 5.2. Impact of $\boldsymbol{g}$ on $f$'s learning

We examines whether Theorem 4.3 can explain the learning behavior of $f$ with $\boldsymbol{g}$ fixed in real-world data. To verify this, $\boldsymbol{g}$ must be created with several estimation errors. However, precisely controlling estimation errors of $\boldsymbol{g}$ via learning algorithms is challenging. In discrete WFL, since WFs are discrete, this issue was addressed by creating randomly incorrect models with certain error rates (Sugiyama & Uchida, 2025). In contrast, continuous WFL requires controlling the MSE of $\boldsymbol{g}$, making the same approach infeasible.

To address this limitation, we construct $g_j$ with MSE $\alpha \in$

$\mathbb{R}_+$ using the following sampling method to generate the predicted value $\hat{x}_j^{\mathrm{w}}$ for an exact value $x_j^{\mathrm{w}}$, where $j \in [F^{\mathrm{w}}]$:

$$\begin{aligned} &\beta \sim \mathrm{Bern}(1/2), \text{ where } \beta \in \{0,1\}, \\ &\hat{x}_j^{\mathrm{w}} \sim \beta \times \mathcal{N}(x_j^{\mathrm{w}} + \beta\sqrt{\alpha} + (1-\beta)\sqrt{\alpha}, \alpha/k^2), \end{aligned} \quad (5.15)$$

where $k \in \mathbb{R}_+$. In this study, $k$ is set to 2. This approach allows precise control over the estimation errors of $\boldsymbol{g}$, enabling an investigation of $f$'s learning results across varying levels of $\boldsymbol{g}$'s errors.

Figure 5.1 illustrates the relationship between the number of training samples $n$ and the generalization error $R_{l,\boldsymbol{g}}(f)$ when $f$ trained with $\boldsymbol{g}$ of varying estimation errors. Here, the MSE of $\boldsymbol{g}$ is controlled within the range $[0.001, 0.5]$. The results indicate that lower estimation error for $\boldsymbol{g}$ result in a faster decrease in $R_{l,\boldsymbol{g}}(f)$ as $n$ increases.

Additionally, Figure 5.2 compares the observed generalization error $R_{l,\boldsymbol{g}}(f)$ with the error bound in Theorem 4.3. From Figure 5.2, the decrease in $R_{l,\boldsymbol{g}}(f_{\boldsymbol{g},\overline{S}})$ and the error bound shows similar behavior, with greater reductions when the MSE of $\boldsymbol{g}$ is smaller. This indicates that Theorem 4.3 can well explain the relationship between $\boldsymbol{g}$'s MSE and $R_{l,\boldsymbol{g}}(f_{\boldsymbol{g},\overline{S}})$. Discrepancies between $R_{l,\boldsymbol{g}}(f_{\boldsymbol{g},\overline{S}})$ and the error bound can be caused by the fact that the error bound does not account for the importance of WFs in predicting $Y$, as in the case of discrete WFL (Sugiyama & Uchida, 2025). While our derived bound cannot account for the feature importance of WFs, it captures fundamental characteristics common across diverse scenarios. Additionally, our results are considered to provide an important foundation for deriving error bounds that reflect the feature importance of WFs.

## 6. Conclusion

In this paper, we presented a unified framework for the formulation and theoretical analysis of continuous WFL. First, we proposed a formulation of continuous WFL that is capable of handling arbitrary continuous WFs. We validated this formulation by demonstrating that minimizing the proposed objective function helps to the label prediction model $f$ to capture the true input-output relationships. Under this formulation, we conducted generalization error analyses for LAC-cWFL, which encompasses both the feature estimation models $\boldsymbol{g}$ and $f$. Through this analysis, we provided a detailed understanding of how $\boldsymbol{g}$ and $f$ affect each other's error bound in finite-sample settings. Furthermore, we identified the conditions under which sequential and iterative learning strategies within LAC-cWFL achieve consistency. Finally, through numerical experiments using real-world datasets, we confirmed that our theoretical findings align well with the actual learning behavior. The results of this study are shown to correspond parallelly with the

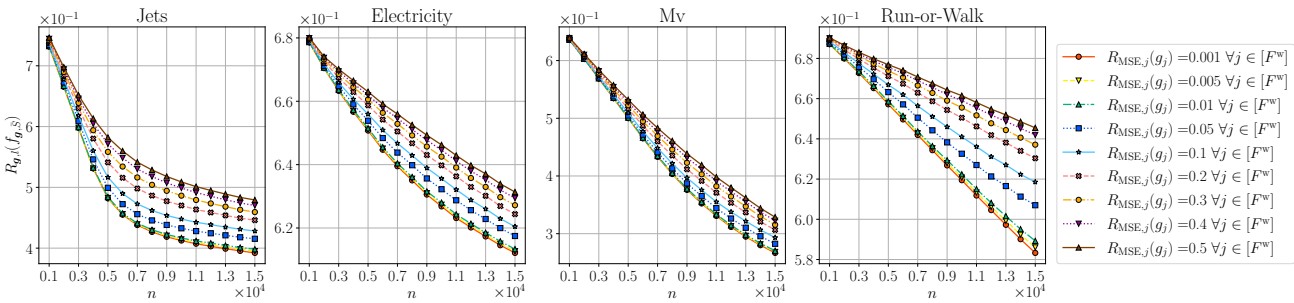

*Figure 5.1.* The relationship between $R_{l,\boldsymbol{g}}(f_{\boldsymbol{g},\overline{S}})$ and the training data size $n$ for various MSEs of $\boldsymbol{g}$.

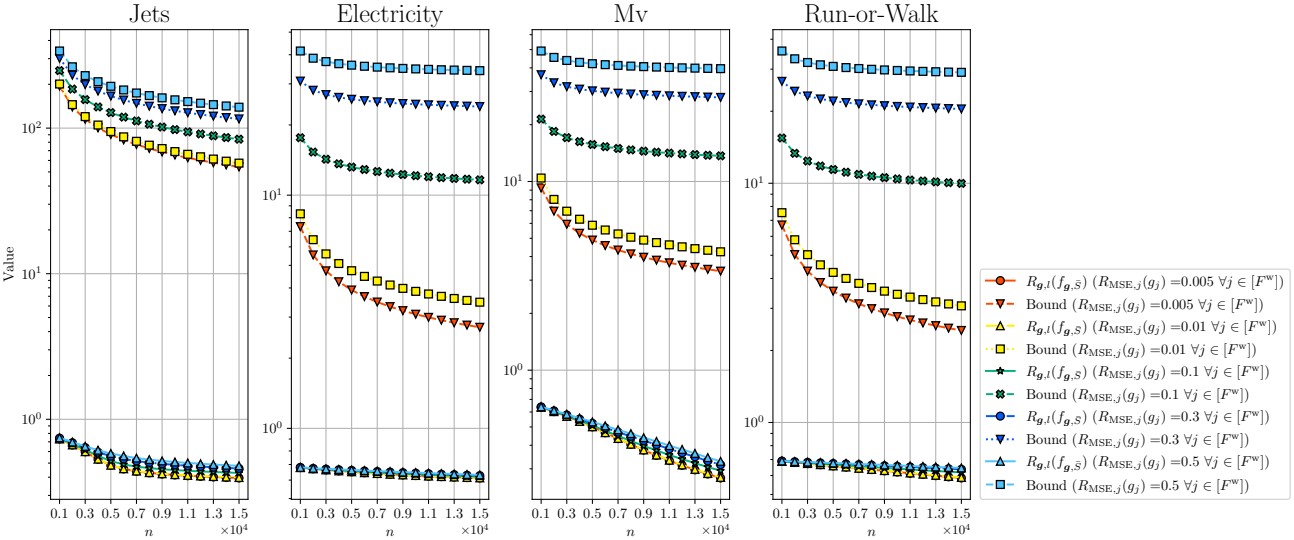

*Figure 5.2.* A comparison between $R_{l,\boldsymbol{g}}(f_{\boldsymbol{g},\overline{S}})$ for various MSEs of $\boldsymbol{g}$ and the error bound derived in Theorem 4.3.

findings of discrete WFL, which addresses discrete WFs. Thus, our findings demonstrate that the WFL framework operates universally for both discrete and continuous WFs, offering theoretical support for a wide range of application domains where low-quality features are prevalent.

## Impact Statement

Our paper presents a formalization and theoretical analysis of continuous WFL that accommodates arbitrary continuous WFs. The relationship revealed in this study between low-quality continuous features and the predictive performance in downstream tasks is also relevant to discussions on the safety of machine learning. For example, our findings may inform questions such as: what kinds of WFs tend to induce models with socially undesirable bias, or which WFs contribute to model vulnerabilities. The theoretical results presented in this work are considered to offer a critical foundation for such discussions.

## Acknowledgements

This work was supported in part by the Japan Society for the Promotion of Science through Grants-in-Aid for Scientific Research (C) (23K11111).

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

# A. Considerations for the theoretical integration of discrete WFL and continuos WFL

The results presented in the main body of this paper provide a unified theoretical foundation for WFL in a wide range of scenarios where some features are continuous WFs. Combined with the analysis results from discrete WFL (Sugiyama & Uchida, 2025), our findings clarify numerous theoretical properties of WFL across various problem settings. However, these frameworks do not address situations where discrete and continuous WFs coexist. This limitation arises because discrete WFL represents the estimation errors of the feature estimation models $g$ using the 0-1 loss, whereas continuous WFL expresses them using the mean squared error (MSE).

On the other hand, Lemma 4.1, which is fundamental to the results in this paper, is applicable regardless of the nature of the WFs (discrete or continuous). Hence, in this section, we explore a methodology for extending Lemma 4.1 to enable an analysis of an *integrated WFL* framework that unifies discrete and continuous WFL. We aim to explore how the theoretical framework established for discrete WFL and continuous WFL can be extended to the broader case of integrated WFL, which encompasses arbitrary WFs regardless of whether they are discrete or continuous.

Most of the notations used here are consistent with those in the main body of this paper. By contrast, since Lemma 4.1 assumes that the feature estimation model is probabilistic model, we define the feature estimation model as probabilistic model $q(\boldsymbol{x}^{\mathrm{w}}|\boldsymbol{x}^{\mathrm{o}})$ in this section. Thus, the discussion in this section can also be interpreted as a generalization of the analysis of discrete WFL and continuous WFL with respect to feature estimation models, which were previously restricted to deterministic models.

## A.1. Formulation of Integrated WFL

In this section, we formulate integrated WFL based on risk minimization. In integrated WFL, the objective is to improve the generalization performance of downstream tasks and restore the explainability lost due to the presence of WFs by accurately estimating the exact values of WFs using a probabilistic model $q$. To achieve this, we aim to learn a label prediction model $f$ and a feature estimation model $q$ that minimize the following two objectives. The first objective is the generalization error of $f$, evaluated by the following risk:

$$R_{l,q}(f) := \mathbb{E}_{p_*(\boldsymbol{x}^{\mathrm{o}},y)q(\boldsymbol{x}^{\mathrm{w}}|\boldsymbol{x}^{\mathrm{o}})}[l(f(\boldsymbol{X}),Y)]. \tag{A.16}$$

The second objective is the estimation error of $q$, evaluated by the following Hellinger divergence:

$$\mathbb{E}_{p_*(\boldsymbol{x}^{\mathrm{o}},y)}\big[D_{\mathrm{H}}^2(p_*(\boldsymbol{X}^{\mathrm{w}}|\boldsymbol{X}^{\mathrm{o}},Y), q(\boldsymbol{X}^{\mathrm{w}}|\boldsymbol{X}^{\mathrm{o}}))\big]. \tag{A.17}$$

Finally, the objective function for integrated WFL is defined as a linear combination of these two terms:

$$R_{l,\lambda}^{\mathrm{iWFL}}(f,q) := R_{l,q}(f) + \lambda\mathbb{E}_{p_*(\boldsymbol{x}^{\mathrm{o}},y)}\big[D_{\mathrm{H}}^2(p_*(\boldsymbol{X}^{\mathrm{w}}|\boldsymbol{X}^{\mathrm{o}},Y), q(\boldsymbol{X}^{\mathrm{w}}|\boldsymbol{X}^{\mathrm{o}}))\big], \ \ \lambda \in \mathbb{R}_+. \tag{A.18}$$

The objective function $R_{l,\lambda}^{\mathrm{iWFL}}$ is justified by the following theorem, the proof of which can be found in Appendix D.1.

**Theorem A.1.** *For any measurable $f \in \mathcal{F}$, probabilistic model $q(\boldsymbol{x}^{\mathrm{w}}|\boldsymbol{x}^{\mathrm{o}})$ and $l$ bounded by $U_l$, the following inequality holds:*

$$R_l(f) \leq R_{l,q}(f) + 2U_l(\mathbb{E}_{p_*(\boldsymbol{x}^{\mathrm{o}},y)}[D_{\mathrm{H}}^2(p_*(\boldsymbol{x}^{\mathrm{w}}|\boldsymbol{X}^{\mathrm{o}},Y), q(\boldsymbol{x}^{\mathrm{w}}|\boldsymbol{X}^{\mathrm{o}}))])^{\frac{1}{2}}. \tag{A.19}$$

Theorem A.1 demonstrates that minimizing the two terms defining $R_{l,\lambda}^{\mathrm{iWFL}}$ contributes to construction of $f$ that captures the true input-output relationship. This result validates the formulation of integrated WFL in a manner consistent with the discrete and continuous WFL frameworks, affirming the justification of the objective function under this formulation.

Next, we define a generalized class of learning algorithms for integrated WFL, analogous to LAC-dWFL (Sugiyama & Uchida, 2025) and LAC-cWFL, as follows:

**Definition A.2** (LAC-iWFL)**.** The *learning algorithm class for integrated WFL (LAC-iWFL)* refers to the class of learning algorithms that learn the feature estimation model $q$ and the label prediction model $f$ in integrated WFL through any combination of the following three steps: (i) Learning $q$ by directly or indirectly minimizing the second term on the RHS of Eq. (A.18). (ii) Learning $f$ with a fixed $q$ using $R_{l,q}$ as the objective function. (iii) Learning $q$ with a fixed $f$ using $R_{l,\lambda}^{\mathrm{iWFL}}$ as the objective function.

The next section analyze the generalization error of LAC-iWFL.

### A.2. Theoretical Analysis for Learning $f$ with fixed $q$

In this section, we analyze the generalization error when learning $f$ using a fixed $q$ as part of LAC-iWFL's step (ii). For theoretical analysis, we introduce some definitions. Similar to Section 4.2, we define the ordinary dataset $S = \{(\boldsymbol{x}_i, y_i)\}_{i=1,\ldots,n}$ and the weak dataset $\overline{S} = \{(\bar{\boldsymbol{x}}_i^{\mathrm{w}}, \boldsymbol{x}_i^{\mathrm{o}}, y_i)\}_{i=1,\ldots,n}$, where the $i$-th sample in $S$ and $\overline{S}$ correspond to the same instance for any $i \in [n]$. The empirical risks $\widehat{R}_l$ and $\widehat{R}_{l,\boldsymbol{g}}$ are computed by the sample average over $S$ and $\overline{S}$, respectively. For any $q$, the empirical risk minimizer obtained from LAC-iWFL's step (ii) is defined as follows:

$$f_{q,\overline{S}} := \arg\min_{f\in\mathcal{F}} \widehat{R}_{l,q}(f).$$

Based on Lemma 4.1, the generalization error bound for $f_{q,\overline{S}}$, learned through LAC-iWFL's step (ii) , is derived in the following theorem. The proof is provided in Appendix D.2.

**Theorem A.3.** *Suppose $S$ and $\overline{S}$ are ordinary and weak datasets consisting of $n$ samples, respectively. For any measurable $\boldsymbol{g} \in \mathcal{G}$, $L_l$-Lipschitz continuous $l$ bounded by $U_l < \infty$ and $\delta \in (0,1)$, the following inequality holds with probability at least $1 - \delta$:*

$$R_{l,q}(f_{q,\overline{S}}) - R_l(f_{\mathcal{F}})$$

$$\leq 4\left(L_l\mathfrak{R}_n^*(\mathcal{F}) + L_l\mathfrak{R}_n^q(\mathcal{F}) + U_l\sqrt{\frac{\log(4/\delta)}{2n}}\right)$$

$$+ \left\{2\left(R_l(f_{\mathcal{F}}) + 4L_l\mathfrak{R}_n^*(\mathcal{F}) + 2U_l\sqrt{\frac{\log(4/\delta)}{2n}}\right)^{\frac{1}{2}} + \left(2U_l\mathbb{E}_{p_*(\boldsymbol{x}^{\mathrm{o}},y)}\big[D_{\mathrm{H}}^2(p_*(\boldsymbol{X}^{\mathrm{w}}|\boldsymbol{X}^{\mathrm{o}},Y), q(\boldsymbol{X}^{\mathrm{w}}|\boldsymbol{X}^{\mathrm{o}}))\big]\right)^{\frac{1}{2}}\right\}$$

$$\times \left(2U_l\mathbb{E}_{p_*(\boldsymbol{x}^{\mathrm{o}},y)}\big[D_{\mathrm{H}}^2(p_*(\boldsymbol{X}^{\mathrm{w}}|\boldsymbol{X}^{\mathrm{o}},Y), q(\boldsymbol{X}^{\mathrm{w}}|\boldsymbol{X}^{\mathrm{o}}))\big]\right)^{\frac{1}{2}}.$$

$$(A.20)$$

*Here, $f_{\mathcal{F}} := \arg\min_{f\in\mathcal{F}} R_l(f)$ is the true risk minimizer in ordinary supervised learning. Also, $\mathfrak{R}_n^*(\mathcal{F})$ and $\mathfrak{R}_n^q(\mathcal{F})$ denote the Rademacher complexities calculated using $p_*(\boldsymbol{x}, y)$ and $p_*(\boldsymbol{x}^{\mathrm{o}}, y)q(\boldsymbol{x}^{\mathrm{w}}|\boldsymbol{x}^{\mathrm{o}})$, respectively.*

Theorem A.3 allows for the analysis of generalization error in learning $f$ while accounting for the estimation error of the probabilistic feature estimation model $q$ in scenarios involving mixed discrete and continuous WFs. This result elucidates the effect of $q$'s estimation error on the error bound for $f$. We assume that $\mathcal{F}$ is sufficiently expressive and that $R_l(f_{\mathcal{F}}) = 0$. Additionally, we assume that the orders of $\mathfrak{R}_n^*(\mathcal{F})$ and $\mathfrak{R}_n^q(\mathcal{F})$ are $\mathcal{O}_p(1/n^{1/2})$. Then, the order of the first term on the RHS of Eq. (A.20) is $\mathcal{O}_p(1/n^{1/2})$, while the order of the second term is $\mathcal{O}_p(1/n^{1/4})$. Therefore, Eq. (A.20) implies that larger the estimation error in $q$, as measured by the Hellinger distance, result in the slower-converging second term dominating the error bound, thereby slowing the rate of error reduction.

While Theorem A.3 clarifies the impact of $q$'s estimation error on learning $f$, constructing a parallel discussion analogous to Section 4.2 remains challenging. This difficulty arises because the Hellinger distance, used to measure $q$'s estimation error, evaluates the divergence between the conditional joint distributions $\boldsymbol{X}^{\mathrm{w}}|\boldsymbol{X}^{\mathrm{o}}, Y$ and $\boldsymbol{X}^{\mathrm{w}}|\boldsymbol{X}^{\mathrm{o}}$, without assessing estimation errors for individual WFs. Moreover, to the best of ours' knowledge, there are no theoretically guaranteed methods for learning $q$ from $\overline{\boldsymbol{X}}^{\mathrm{w}}$ as weak supervision that their theories can be combined with Theorem A.3, as in Theorem 4.3. For the reasons outlined above, it is currently challenging to conduct further theoretical analysis of integrated WFL. This limitation also applies to the learning of $q$ with fixed $f$ in LAC-iWFL's step (iii).

In contrast, the discussions above can also be extended to scenarios where $q$'s estimation error is measured using the Kullback-Leibler (KL) divergence $D_{\mathrm{KL}}(\cdot||\cdot)$. Specifically, integrated WFL can be formulated with the following objective function, allowing for similar discussions:

$$R_{l,\lambda,\mathrm{KL}}^{\mathrm{iWFL}}(f, q) := R_{l,q}(f) + \lambda D_{\mathrm{KL}}(p_*(\boldsymbol{X}^{\mathrm{w}}|\boldsymbol{X}^{\mathrm{o}}, Y)||q(\boldsymbol{X}^{\mathrm{w}}|\boldsymbol{X}^{\mathrm{o}})), \quad \lambda \in \mathbb{R}_+. \tag{A.21}$$

This is because, by utilizing the following inequality, we can derive equivalent inequalities corresponding to Theorem A.1 and Theorem A.3, which evaluates the estimation error of $q$ in terms of the KL divergence:

$$\mathbb{E}_{p_*(\boldsymbol{x}^{\mathrm{o}},y)}\big[D_{\mathrm{H}}^2(p_*(\boldsymbol{X}^{\mathrm{w}}|\boldsymbol{X}^{\mathrm{o}}, Y), q(\boldsymbol{X}^{\mathrm{w}}|\boldsymbol{X}^{\mathrm{o}}))\big] \leq \mathbb{E}_{p_*(\boldsymbol{x}^{\mathrm{o}},y)}\left[\int_{\mathcal{X}^{\mathrm{w}}} p_*(\boldsymbol{x}^{\mathrm{w}}|\boldsymbol{X}^{\mathrm{o}}, Y) \log\frac{p_*(\boldsymbol{x}^{\mathrm{w}}|\boldsymbol{X}^{\mathrm{o}}, Y)}{q(\boldsymbol{x}^{\mathrm{w}}|\boldsymbol{X}^{\mathrm{o}})} \mathrm{d}\boldsymbol{x}^{\mathrm{w}}\right]$$

$$= D_{\mathrm{KL}}(p_*(\boldsymbol{X}^{\mathrm{w}}|\boldsymbol{X}^{\mathrm{o}}, Y)||q(\boldsymbol{X}^{\mathrm{w}}|\boldsymbol{X}^{\mathrm{o}})). \tag{A.22}$$

This suggests that the integrated WFL framework is compatible with methods that learn $q$ as a maximum likelihood estimation problem. However, it remains challenging to express the error bound of $f$ in a form that depends on divergences individually evaluating the estimation error for each WF. Addressing these challenges in the theoretical analysis of integrated WFL will be an important direction for future research.

## B. Impute-then-Regress: the Case All Features Are Continuous and Can Be Missing

The results presented in the main body of this paper provide a unified theoretical framework for WFL that handles datasets where some features are continuous WFs, clarifying several theoretical properties. However, the analysis in Section 4 has a limitation: it cannot accommodate cases where all features are WFs. This limitation arises because the analysis in Section 4 assumes that the exact values of WFs are estimated using OFs. This issue is particularly significant in the context of ItR problems. In real-world scenarios, it is not uncommon for all features to be subject to missingness. Moreover, the latest theoretical studies on ItR assume scenarios where all features are continuous and may be missing (Le Morvan et al., 2021). Therefore, to conduct a comprehensive generalization error analysis for ItR from the perspective of continuous WFL, it is necessary to extend the analysis in Section 4 to include cases where "all features are continuous and may be missing."

In this section, we extend the analysis in Section 4 to conduct a finite-sample generalization error analysis for the case where "all features are continuous and may be missing." These results correspond in parallel with those presented in Section 4. The findings of this section, combined with the results of Section 4 and the theoretical analysis of discrete WFL (Sugiyama & Uchida, 2025), establish a unified generalization error analysis framework applicable to a broad range of ItR problem settings.

### B.1. Preliminary

To address cases where all features may be missing, we introduce definitions partially different from those in the Main Body, adhering to the definitions of the latest theoretical study on ItR (Le Morvan et al., 2021). Let $\boldsymbol{X} = (X_1, \ldots, X_d)$ denote the sequence of random variables representing the complete input features without missingness, and let $Y$ denote the random variable representing the downstream task label. Here, $d$ represents the dimensionality of the input features. The sets of possible values for $\boldsymbol{X}$ and $Y$ are denoted by $\mathcal{X} = \prod_{j=1}^{d} \mathcal{X}_j \subseteq \mathbb{R}^d$ and $Y$, respectively. Define $\overline{\boldsymbol{X}} = (\overline{X}_1, \ldots, \overline{X}_d)$ as the sequence of random variables representing the incomplete input features that may contain missing values. The set of possible values for $\overline{\boldsymbol{X}}$ is denoted by $\overline{\mathcal{X}} = \prod_{j=1}^{d}(\mathcal{X}_j \cup \{\text{NA}\})$, where NA indicates a missing value. Let $\boldsymbol{M} = (M_1, ..., M_d)$ represent the sequence of random variables indicating the missingness of features. The set of possible values for $\boldsymbol{M}$ is $\{0, 1\}^d$. For any $j \in [d]$, $M_j = 0$ indicates that $X_j$ is observed (i.e., $\overline{X}_j = X_j$), while $M_j = 1$ indicates that $X_j$ is missing (i.e., $\overline{X}_j = \text{NA}$). By definition, $\overline{\boldsymbol{X}}$ is uniquely determined by $\boldsymbol{X}$ and $\boldsymbol{M}$. Additionally, $\boldsymbol{M}$ can be uniquely determined from $\overline{\boldsymbol{X}}$. From this fact, we omit $\boldsymbol{M}$ when both $\overline{\boldsymbol{X}}$ and $\boldsymbol{M}$ are arguments. the realizations of $\boldsymbol{X}, Y, \boldsymbol{M}$, and $\overline{\boldsymbol{X}}$ are represented as $\boldsymbol{x}, y, \boldsymbol{m}$ and $\bar{\boldsymbol{x}}$, respectively. The true joint probability distribution of $\boldsymbol{X}, Y$ and $\boldsymbol{M}$ is denoted as $p_*(\boldsymbol{x}, y, \boldsymbol{m})$. All samples $(\boldsymbol{x}, y, \boldsymbol{m})$ are assumed to be independently drawn from $p_*(\boldsymbol{x}, y, \boldsymbol{m})$. For a given $\boldsymbol{m}$, let $\text{obs}(\boldsymbol{m}) \subseteq \{1, \ldots, d\}$ denote the set of indices corresponding to observed features, and $\text{mis}(\boldsymbol{m}) \subseteq \{1, \ldots, d\}$ denote the set of indices corresponding to missing features.

Define the *label prediction model* as $f \in \mathcal{F} : \mathcal{X} \to \mathcal{Y}$, where $\mathcal{F}$ is the hypothesis class for label prediction models. Define the *feature imputation models* $\boldsymbol{g} = (g_1, \ldots, g_d) \in \mathcal{G}_1 \times \cdots \times \mathcal{G}_d : \overline{\mathcal{X}} \times \{0, 1\}^d \to \mathcal{X}$ as follows:

$$g_j = \begin{cases} X_j & \text{if } M_j = 0 \\ \phi_j^{(M)}(\boldsymbol{X}_{\text{obs}(\boldsymbol{M})}) & \text{if } M_j = 1 \end{cases}, \tag{B.23}$$

where $\phi_j^{(\boldsymbol{M})} : \mathcal{X}_{\text{obs}(\boldsymbol{M})} \to \mathcal{X}_j$ and $\mathcal{G}_j$ represents the hypothesis class of feature imputation models for the $j$-th feature. The PDF associated with $\boldsymbol{g}$ is denoted by $q_{\boldsymbol{g}}(\boldsymbol{x}_{\text{mis}(\boldsymbol{m})} | \boldsymbol{x}_{\text{obs}(\boldsymbol{m})}, \boldsymbol{m})$.

### B.2. Formulation

we formalize ItR in the scenario where all features are continuous and potentially missing so that it corresponds in parallel to the discussion in Section 3.1. ItR aims to achieve two main goals: improving the generalization performance of downstream tasks and restoring the interpretability lost due to missing values. Consequently, we aim to learn $f$ and $\boldsymbol{g}$ that minimize the

following two risks. The first risk evaluates the generalization error of the label prediction model $f$:

$$R_{l,\boldsymbol{g}}(f) := \mathbb{E}_{p_*(y,\boldsymbol{m})q_{\boldsymbol{g}}(\boldsymbol{x}|\boldsymbol{m},y)}[l(f(\boldsymbol{X}),Y)] = \mathbb{E}_{p_*(\boldsymbol{x},\boldsymbol{m},y)}[l(f(\boldsymbol{g}(\overline{X})),Y)]. \tag{B.24}$$

The second risk evaluates the estimation error of the feature imputation models $\boldsymbol{g}$ under specific missing patterns:

$$R_{\mathrm{MSE},j}^{(\boldsymbol{m})}(g_j) := \mathbb{E}_{p_*(\boldsymbol{x}|\boldsymbol{m})}[l_{\mathrm{MSE}}(X_j - g_j(\overline{\boldsymbol{X}}))] = \mathbb{E}_{p_*(\boldsymbol{x}|\boldsymbol{m})}[l_{\mathrm{MSE}}(X_j - \phi_j^{(\boldsymbol{m})}(\boldsymbol{X}_{\mathrm{obs}(\boldsymbol{m})}))], \forall j \in [d], \tag{B.25}$$

where $l_{\mathrm{MSE}}(y,y') = (y - y')^2$ represents the mean squared error (MSE). Finally, the objective function for ItR with only continuous and potentially missing features is defined as a linear combination of these risks:

$$R_{l,\lambda}^{\mathrm{ItR}}(f,g) := R_{l,\boldsymbol{g}}(f) + \lambda \mathbb{E}_{p_*(\boldsymbol{m})}\left[ \sum_{j \in \mathrm{mis}(\boldsymbol{M})} R_{\mathrm{MSE},j}^{(\boldsymbol{M})}(g_j) \right], \quad \lambda \in \mathbb{R}_+. \tag{B.26}$$

Theorem 3.1, which corresponds to the formalization involving OFs and continuous WFs, can be generalized to the ItR framework. Consequently, the validity of the formalization with the objective function $R_{l,\lambda}^{\mathrm{ItR}}$ in the current setting is demonstrated as the following theorem. The proof is provided in Appendix E.1.

**Theorem B.1.** *Let $q_{\boldsymbol{g}}$ be the PDF of $\mathcal{N}(\boldsymbol{g}_{\mathrm{mis}(\boldsymbol{m})}(\boldsymbol{x}_{\mathrm{obs}(\boldsymbol{m})}), \sigma^2 I_{|\mathrm{mis}(\boldsymbol{m})| \times |\mathrm{mis}(\boldsymbol{m})|})$, where $\sigma \in \mathbb{R}_+$. For any measurable $f \in \mathcal{F}$, $\boldsymbol{g} \in \mathcal{G}$, and $l$ bounded by $U_l < \infty$, the following inequality holds:*

$$R_l(f) \leq R_{l,\boldsymbol{g}}(f) + U_l \left\{ \frac{1}{2}\left( C_\sigma' + \frac{1}{2\sigma^2}\mathbb{E}_{p_*(\boldsymbol{m})}\left[ \sum_{j \in \mathrm{mis}(\boldsymbol{M})} R_{\mathrm{MSE},j}^{(\boldsymbol{M})}(g_j) \right] \right) \right\}^{\frac{1}{2}}, \tag{B.27}$$

*where $C_\sigma' = \mathbb{E}_{p_*(\boldsymbol{x},y,\boldsymbol{m})}[p_*(\boldsymbol{X}_{\mathrm{mis}(\boldsymbol{M})}|\boldsymbol{X}_{\mathrm{obs}(\boldsymbol{M})}, Y, \boldsymbol{M})] + \mathbb{E}_{p_*(\boldsymbol{m})}[\frac{1}{2}\log(2\pi)^{|\mathrm{mis}(\boldsymbol{M})|}\sigma^{2|\mathrm{mis}(\boldsymbol{M})|}]$.*

According to the definition of $C_\sigma'$, when for any $j \in [d]$, $\mathcal{X}_j$ is a set of finite-precision decimals (in this case, $\mathbb{E}_{p_*(\boldsymbol{x},y,\boldsymbol{m})}[p_*(\boldsymbol{X}_{\mathrm{mis}(\boldsymbol{M})}|\boldsymbol{X}_{\mathrm{obs}(\boldsymbol{M})}, Y, \boldsymbol{M})] \leq 0$ holds), and $q_{\boldsymbol{g}}$ is defined with $\sigma^2 = (2\pi)^{-1}$, it follows that $C_\sigma' \leq 0$. Thus, under this mild assumption, $C_\sigma'$ can be disregarded.

Next, we define a general class of learning algorithms for ItR, analogous to LAC-cWFL, as follows:

**Definition B.2** (LAC-ItR). The *learning algorithm class for Impute then Regress (LAC-ItR)* refers to the set of algorithms for learning the feature imputation model $\boldsymbol{g}$ and the label prediction model $f$ in ItR using one or a combination of the following three steps:

(i) Learning $\boldsymbol{g}$ by directly or indirectly minimizing the second term on the RHS of Eq. (B.26).

(ii) Learning $f$ using $R_{l,\boldsymbol{g}}$ as the objective function while fixing $\boldsymbol{g}$.

(iii) Learning $\boldsymbol{g}$ using $R_{l,\lambda}^{\mathrm{ItR}}$ as the objective function while fixing $f$.

Analogous to LAC-cWFL, LAC-ItR encompasses a wide range of methods, including *sequential learning*, which involves steps (i) and (ii), and *iterative learning*, which alternates between steps (ii) and (iii) (Yoon et al., 2018; Mattei & Frellsen, 2019; Ipsen et al., 2021; Josse et al., 2024; Le Morvan et al., 2020a; 2021; Ipsen et al., 2022). In the following sections, we provide a generalization error analysis for this class of learning algorithms.

### B.3. Deriving Analysis Inequalities

To conduct a theoretical analysis for ItR involving continuous and potentially missing features, analogous to Section 4, it is necessary to derive results corresponding to Lemmas 4.1 and 4.2. In fact, the proofs of these lemmas can be extended to the problem setting of this section, leading to the following two lemmas. The first lemma assumes the use of an arbitrary probabilistic model for missing value imputation and corresponds to Lemma 4.1 in parallel. The proof is provided in Appendix E.2.

**Lemma B.3.** *For any measurable probabilistic model $q(\boldsymbol{x}_{\mathrm{mis}(\boldsymbol{m})}|\boldsymbol{x}_{\mathrm{obs}(\boldsymbol{m})}, \boldsymbol{m}))$, $f \in \mathcal{F}$ and $l$ bounded by $U_l < \infty$, the following inequality holds:*

$$
\begin{aligned}
&|R_l(f) - R_{l,\boldsymbol{g}}(f)| \\
&\leq \left( \sqrt{R_l(f)} + \sqrt{R_{l,\boldsymbol{g}}(f)} \right) \\
&\quad \times \left\{ 2U_l \mathbb{E}_{p_*(\boldsymbol{m})p_*(\boldsymbol{x}_{\mathrm{obs}(\boldsymbol{m})}, y|\boldsymbol{m})} \left[ \left\{ D_{\mathrm{H}}(p_*(\boldsymbol{X}_{\mathrm{mis}(\boldsymbol{M})}|\boldsymbol{X}_{\mathrm{obs}(\boldsymbol{M})}, Y, \boldsymbol{M}), q(\boldsymbol{X}_{\mathrm{mis}(\boldsymbol{M})}|\boldsymbol{X}_{\mathrm{obs}(\boldsymbol{M})}, \boldsymbol{M})) \right\}^2 \right] \right\}^{\frac{1}{2}}.
\end{aligned}
\tag{B.28}
$$

When the probabilistic model $q(\boldsymbol{x}_{\mathrm{mis}(\boldsymbol{m})}|\boldsymbol{x}_{\mathrm{obs}(\boldsymbol{m})}, \boldsymbol{m}))$ is defined as the PDF of $\mathcal{N}(\boldsymbol{g}_{\mathrm{mis}(\boldsymbol{m})}(\boldsymbol{x}_{\mathrm{obs}(\boldsymbol{m})}), \sigma^2 I_{|\mathrm{mis}(\boldsymbol{m})| \times |\mathrm{mis}(\boldsymbol{m})|})$, where $\boldsymbol{g}$ is a deterministic feature imputation model and $\sigma \in \mathbb{R}_+$, the second lemma holds. The proof is provided in Appendix E.3.

**Lemma B.4.** *Let $q_{\boldsymbol{g}}$ be the PDF of $\mathcal{N}(\boldsymbol{g}_{\mathrm{mis}(\boldsymbol{m})}(\boldsymbol{x}_{\mathrm{obs}(\boldsymbol{m})}), \sigma^2 I_{|\mathrm{mis}(\boldsymbol{m})| \times |\mathrm{mis}(\boldsymbol{m})|})$, where $\sigma \in \mathbb{R}_+$. For any measurable $f \in \mathcal{F}$, $\boldsymbol{g} \in \mathcal{G}$ and $l$ bounded by $U_l < \infty$, the following inequality holds:*

$$
|R_l(f) - R_{l,\boldsymbol{g}}(f)| \leq \left( \sqrt{R_l(f)} + \sqrt{R_{l,\boldsymbol{g}}(f)} \right) \left\{ 2U_l \left( C'_\sigma + \frac{1}{2\sigma^2} \mathbb{E}_{p_*(\boldsymbol{m})} \left[ \sum_{j \in \mathrm{mis}(\boldsymbol{M})} R^{(\boldsymbol{M})}_{\mathrm{MSE},j}(g_j)] \right] \right) \right\}^{\frac{1}{2}},
\tag{B.29}
$$

*where, $C'_\sigma = \mathbb{E}_{p_*(\boldsymbol{x}, y, \boldsymbol{m})}[p_*(\boldsymbol{X}_{\mathrm{mis}(\boldsymbol{M})}|\boldsymbol{X}_{\mathrm{obs}(\boldsymbol{M})}, Y, \boldsymbol{M})] + \mathbb{E}_{p_*(\boldsymbol{m})}[\frac{1}{2}\log(2\pi)^{|\mathrm{mis}(\boldsymbol{M})|}\sigma^{2|\mathrm{mis}(\boldsymbol{M})|}]$.*

Lemma B.4 correspond to Lemma 4.2. While Lemma B.3 is more broadly applicable than Lemma B.4, our analysis, similar to Section 4, focuses on the relationship between the estimation errors of $\boldsymbol{g}$ represented by MSE and $f$. Hence, we proceed by using Lemma B.4 with $q_{\boldsymbol{g}}$ set as the PDF of $\mathcal{N}(\boldsymbol{g}_{\mathrm{mis}(\boldsymbol{m})}(\boldsymbol{x}_{\mathrm{obs}(\boldsymbol{m})}), \sigma^2 I_{|\mathrm{mis}(\boldsymbol{m})| \times |\mathrm{mis}(\boldsymbol{m})|})$. This setting does not restrict the learning method for $\boldsymbol{g}$, preserving the generality of our analysis for deterministic models $\boldsymbol{g}$.

### B.4. Theoretical Analysis for Learning $f$ with fixed $\boldsymbol{g}$

In this section, we analyze the learning of $f$ by minimizing $R_{l,\boldsymbol{g}}$ in LAC-ItR's step (ii) with $\boldsymbol{g}$ fixed. We first assume that $n$ samples $\{(\boldsymbol{x}_i, y_i, \boldsymbol{m}_i)\}_{i=1}^n$ are independently drawn from $p_*(\boldsymbol{x}, y, \boldsymbol{m})$. The ordinary dataset and weak dataset are defined as $S := \{(\boldsymbol{x}_i, y_i)\}_{i=1}^n$ and $\overline{S} := \{(\bar{\boldsymbol{x}}_i, y_i, \boldsymbol{m}_i)\}_{i=1}^n$, respectively. Here, the $i$-th samples in $S$ and $\overline{S}$ correspond to the same instance, for any $i \in [n]$. Hereafter, $\widehat{R}_l$ and $\widehat{R}_{l,\boldsymbol{g}}$ denote the empirical risks that approximate $R_l$ and $R_{l,\boldsymbol{g}}$ by sample averages based on $S$ and $\overline{S}$, respectively. For any $\boldsymbol{g} \in \mathcal{G}$, the empirical risk minimizer obtained from LAC-ItR's step (ii) is defined as follows:

$$
f_{\boldsymbol{g}, \overline{S}} := \arg\min_{f \in \mathcal{F}} \widehat{R}_{l,\boldsymbol{g}}(f).
$$

From Lemma 4.2, the error bound for $f_{\boldsymbol{g}, \overline{S}}$ obtained using LAC-ItR's step (ii) is derived in the following theorem. The proof is provided in Appendix E.4.

**Theorem B.5.** *Let $q_{\boldsymbol{g}}$ be the PDF of $\mathcal{N}(\boldsymbol{g}_{\mathrm{mis}(\boldsymbol{m})}(\boldsymbol{x}_{\mathrm{obs}(\boldsymbol{m})}), \sigma^2 I_{|\mathrm{mis}(\boldsymbol{m})| \times |\mathrm{mis}(\boldsymbol{m})|})$, where $\sigma \in \mathbb{R}_+$. Let $S$ and $\overline{S}$ represent the ordinary dataset and weak dataset, respectively, consisting of $n$ samples. For any measurable $\boldsymbol{g} \in \mathcal{G}$, $L_l$-Lipschitz continuous $l$ bounded by $U_l < \infty$ and $\delta \in (0, 1)$, the following inequality holds with probability at least $1 - \delta$:*

$$
\begin{aligned}
&R_{l,\boldsymbol{g}}(f_{\boldsymbol{g}, \overline{S}}) - R_l(f_{\mathcal{F}}) \\
&\leq 4 \left( L_l \mathfrak{R}^*_n(\mathcal{F}) + L_l \mathfrak{R}^{\boldsymbol{g}}_n(\mathcal{F}) + U_l \sqrt{\frac{\log(4/\delta)}{2n}} \right) \\
&\quad + \left\{ 2 \left( R_l(f_{\mathcal{F}}) + 4L_l \mathfrak{R}^*_n(\mathcal{F}) + 2U_l \sqrt{\frac{\log(4/\delta)}{2n}} \right)^{\frac{1}{2}} + \left( 2U_l \left( C'_\sigma + \frac{1}{2\sigma^2} \mathbb{E}_{p_*(\boldsymbol{m})} \left[ \sum_{j \in \mathrm{mis}(\boldsymbol{M})} R^{(\boldsymbol{M})}_{\mathrm{MSE},j}(g_j)] \right] \right) \right)^{\frac{1}{2}} \right\} \\
&\qquad \times \left\{ 2U_l \left( C'_\sigma + \frac{1}{2\sigma^2} \mathbb{E}_{p_*(\boldsymbol{m})} \left[ \sum_{j \in \mathrm{mis}(\boldsymbol{M})} R^{(\boldsymbol{M})}_{\mathrm{MSE},j}(g_j)] \right] \right) \right\}^{\frac{1}{2}}.
\end{aligned}
\tag{B.30}
$$

*Here, $f_{\mathcal{F}} := \arg\min_{f \in \mathcal{F}} R_l(f)$ is the true risk minimizer in ordinary supervised learning, and $C'_\sigma =$*
$\mathbb{E}_{p_*(\boldsymbol{x},y,\boldsymbol{m})}[p_*(\boldsymbol{X}_{\mathrm{mis}(\boldsymbol{M})}|\boldsymbol{X}_{\mathrm{obs}(\boldsymbol{M})}, Y, \boldsymbol{M})] + \mathbb{E}_{p_*(\boldsymbol{m})}[\frac{1}{2}\log(2\pi)^{|\mathrm{mis}(\boldsymbol{M})|}\sigma^{2|\mathrm{mis}(\boldsymbol{M})|}].$

According to the definition of $C'_\sigma$, when for any $j \in [d]$, $\mathcal{X}_j$ is a set of finite-precision decimals (In this case, $\mathbb{E}_{p_*(\boldsymbol{x},y,\boldsymbol{m})}[p_*(\boldsymbol{X}_{\mathrm{mis}(\boldsymbol{M})}|\boldsymbol{X}_{\mathrm{obs}(\boldsymbol{M})}, Y, \boldsymbol{M})] \leq 0$ holds), and $q_{\boldsymbol{g}}$ is defined with $\sigma^2 = (2\pi)^{-1}$, it follows that $C'_\sigma \leq 0$. Thus, under this mild assumption, $C'_\sigma$ can be disregarded in discussions concerning error bounds.

Theorem B.5 parallels the result in Theorem 4.3 Therefore, it provides theoretical insights equivalent to those discussed in Section 4.2 for ItR involving only continuous missing features, elucidating the influence of $\boldsymbol{g}$ on the learning efficiency of $f$. This result further reveals conditions under which sequential learning in the current setting becomes consistent, as follows. The proof is shown in Appendix E.5.

**Theorem B.6.** *Suppose $\mathcal{X}_j$ is a set of finite-precision decimals for any $j \in [d]$. Assume the existence of true deterministic functions $g_j^* : \bar{\mathcal{X}} \times \{0,1\}^d \to \mathcal{X}_j, \forall j \in [d]$, such that $(g_1^*, \ldots, g_d^*) \in \mathcal{G}$ and $f^* : \mathcal{X} \to \mathcal{Y}$ such that $f^* \in \mathcal{F}$. Additionally, assume that $l$ is $L_l$-Lipschitz continuous, that $\mathfrak{R}_n^*(\mathcal{F})$ and $\mathfrak{R}_n^{\boldsymbol{g}}(\mathcal{F})$ asymptotically converge to 0 as $n \to \infty$, and that for all $j \in [d]$, the number of samples available for learning $g_j$ tends to infinity as $n \to \infty$. Under these conditions, using consistent methods for learning $\boldsymbol{g}$ and setting $\sigma^2 = (2\pi)^{-1}$ in $q_{\boldsymbol{g}}$, sequential learning in ItR becomes consistent, i.e., as $n \to \infty$, $R_{l,\boldsymbol{g}}(f_{\boldsymbol{g},\overline{S}}) \to R_l(f_{\mathcal{F}})$.*

## B.5. Theoretical Analysis for Learning $g$ with fixed $f$

This section analyzes the learning of $\boldsymbol{g}$ by minimizing $R_{l,\lambda}^{\mathrm{ItR}}(f, \boldsymbol{g})$ in LAC-ItR's step (iii) while $f$ is fixed. To conduct a similar analysis as in Section 4.3, we introduce the following definitions.

**Definition B.7.** Let $\hat{p}(\boldsymbol{m})$ denote the empirical distribution of $\boldsymbol{M}$ computed from a finite number of samples. For any $\boldsymbol{r} = (r_1, \ldots, r_d) \in \mathbb{R}_+^d$ and any $\overline{S}$, defined the following hypothesis set:

$$\mathcal{G}(\boldsymbol{r}, \overline{S}) = \mathcal{G}_1(r_1, \overline{S}_1) \times \cdots \times \mathcal{G}_d(r_d, \overline{S}_d),$$
$$\mathrm{where}, \mathcal{G}_j(r_j, \overline{S}_j) := \left\{ g_j | g_j \in \mathcal{G}_j \wedge \mathbb{E}_{\hat{p}(\boldsymbol{m}_{\backslash j})}\left[\widehat{R}_{\mathrm{MSE},j}^{(\boldsymbol{M}_{\backslash j}, M_j=1)}(g_j)\right] \leq r_j \right\},$$
$$\overline{S}_j := \{\bar{\boldsymbol{x}}_i | m_{ij} = 0, i \in [d]\}, \forall j \in [d].$$

Here, $\backslash j$ represents the index set excluding $j$, and $\widehat{R}_{\mathrm{MSE},j}^{(\boldsymbol{M}_{\backslash j}, M_j=1)}$ denotes the empirical risk approximating $R_{\mathrm{MSE},j}^{(\boldsymbol{M}_{\backslash j}, M_j=1)}$ by calculating the sample average over $\overline{S}_j$.

The learning of $\boldsymbol{g}$ under LAC-ItR's step (iii) is then expressed as follows:

$$\boldsymbol{g}_{f,\overline{S}}^{(\boldsymbol{r})} := [g_{\overline{S},d}^{(r_d)}, \ldots, g_{\overline{S},d}^{(r_d)}]$$
$$\mathrm{where} \quad g_{\overline{S},j}^{(r_j)} := \arg\min_{g_j \in \mathcal{G}_j(r_j, \overline{S}_j)} \mathbb{E}_{\hat{p}(\boldsymbol{m}_{\backslash j})}\left[\widehat{R}_{\mathrm{MSE},j}^{(\boldsymbol{M}_{\backslash j}, M_j=1)}(g_j)\right], \quad \forall j \in [d]. \tag{B.31}$$

Under these definitions, the following theorem holds for the case where the missing mechanism is Missing Completely At Random (MCAR). The proof is provided in Appendix E.6.

**Theorem B.8.** *Let $q_{\boldsymbol{g}}$ be the PDF of $\mathcal{N}(\boldsymbol{g}_{\mathrm{mis}(\boldsymbol{m})}(\boldsymbol{x}_{\mathrm{obs}(\boldsymbol{m})}), \sigma^2 I_{|\mathrm{mis}(\boldsymbol{m})| \times |\mathrm{mis}(\boldsymbol{m})|})$, where $\sigma \in \mathbb{R}_+$. Suppose $S$ and $\overline{S}$ represent the ordinary dataset and weak dataset, respectively, consisting of $n$ samples, and the missing mechanism is MCAR. For any measurable $f \in \mathcal{F}$, $l$ bounded by $U_l < \infty$ and $\delta \in (0,1)$, the following inequality holds with probability at least*

$1 - \delta$:

$$R_{l,f}(\boldsymbol{g}_{f,\overline{S}}^{(\boldsymbol{r})}) - R_l(f)$$

$$\leq 2\left(\mathfrak{R}_n^*(\widetilde{\mathcal{G}}_j(r_j, \overline{S}_j)) + U_l\sqrt{\frac{\log((d+1)/\delta)}{2n}}\right)$$

$$+ \left\{2\sqrt{R_l(f)} + \left(2U_l\left(C_\sigma' + \frac{1}{2\sigma^2}\left(\mathbb{E}_{p(\boldsymbol{m})}\left[\sum_{j\in\mathrm{mis}(\boldsymbol{M})} R_{\mathrm{MSE},j}^{(\boldsymbol{M})}\left(g_{\mathcal{G}(\boldsymbol{r},\overline{S}),j}\right)\right]\right.\right.\right.\right.$$

$$\left.\left.\left.\left. + 4\mathfrak{R}_n^*(\widetilde{\mathcal{G}}_j(r_j, \overline{S}_j)) + 2U_l\sqrt{\frac{\log((d+1)/\delta)}{2n}}\right)\right)\right)^{\frac{1}{2}}\right\}$$

$$\times \left\{2U_l\left(C_\sigma' + \frac{1}{2\sigma^2}\left(\mathbb{E}_{p(\boldsymbol{m})}\left[\sum_{j\in\mathrm{mis}(\boldsymbol{M})} R_{\mathrm{MSE},j}^{(\boldsymbol{M})}\left(g_{\mathcal{G}(\boldsymbol{r},\overline{S}),j}\right)\right] + 4\mathfrak{R}_n^*(\widetilde{\mathcal{G}}_j(r_j, \overline{S}_j)) + 2U_l\sqrt{\frac{\log((d+1)/\delta)}{2n}}\right)\right)\right\}^{\frac{1}{2}}.$$

$$(B.32)$$

*Here,* $\widetilde{\mathcal{G}}_j(r_j, \overline{S}_j) := \{(\boldsymbol{x}, \boldsymbol{m}) \mapsto l_{\mathrm{MSE}}(x_j, g_j(\bar{\boldsymbol{x}}, \boldsymbol{m})) : g_j \in \mathcal{G}_j(r_j, \overline{S}_j)\}$, $C_\sigma' = \mathbb{E}_{p_*(\boldsymbol{x},y,\boldsymbol{m})}[p_*(\boldsymbol{X}_{\mathrm{mis}(\boldsymbol{M})}|\boldsymbol{X}_{\mathrm{obs}(\boldsymbol{M})}, Y, \boldsymbol{M})] + \mathbb{E}_{p_*(\boldsymbol{m})}[\frac{1}{2}\log(2\pi)^{|\mathrm{mis}(\boldsymbol{M})|}\sigma^{2|\mathrm{mis}(\boldsymbol{M})|}]$, *and* $g_{\mathcal{G}(\boldsymbol{r},\overline{S}),j} := \arg\min_{g_j\in\mathcal{G}_j(r_j,\overline{S}_j)} \mathbb{E}_{p(\boldsymbol{m})}\left[\sum_{j\in\mathrm{mis}(\boldsymbol{M})} R_{\mathrm{MSE},j}^{(\boldsymbol{M})}(g_j)\right]$.

Theorem B.8 parallels Theorem 4.5. Hence, it provides theoretical insights equivalent to Section 4.3 regarding the influence of $f$ on the learning of $\boldsymbol{g}$ under MCAR assumptions for ItR settings with only continuous missing features. Additionally, Theorems B.5 and B.8 elucidate the conditions under which iterative learning achieves consistent when all features are continuous and potentially subject to missingness, as follows: The proof is provided in Appendix E.7.

**Theorem B.9.** *In addition to the conditions stated in Theorem B.6, suppose $f$ obtained in LAC-ItR's step (ii) be Lipschitz continuous, and the Rademacher complexities about $\boldsymbol{g}$ converge to $0$ asymptotically as $n \to \infty$. Under these conditions, the iterative learning in LAC-ItR's steps (ii) and (iii) achieves consistency.*

As in Section 4, for the sake of simplicity, the variance $\sigma^2$ of $q_{\boldsymbol{g}}$ is defined in this section as a constant independent of $\boldsymbol{x}^\circ$, but it is also possible to define this variance as a function $\sigma^2(\boldsymbol{x}^\circ)$ that depends on $\boldsymbol{x}^\circ$. In this case, Theorem B.1, Lemma B.4, Theorem B.5, and Theorem B.8 still hold, with each occurrence of the constant $\sigma^2$ in the corresponding upper bounds replaced by $\max_{\boldsymbol{x}^\circ} \sigma^2(\boldsymbol{x}^\circ)$.

# C. Proofs for Main Body

## C.1. Proof of Theorem 3.1

In the corresponding theorem for the formulation of discrete WFL (Theorem 3.1 in (Sugiyama & Uchida, 2025)) derives an upper bound on the 0-1 loss with respect to $g_j$ using the fact that the WFs are discrete and that $q_{\boldsymbol{g}}$ is defined as a deterministic probability distribution. In contrast, for continuous WFL, taking into account the continuity of WFs, we define $q_{\boldsymbol{g}}$ as a Gaussian distribution. By leveraging the definition, we derive the term including the MSE with respect to $g_j$ as being upper-bounded by $R_l(f) - R_{l,\boldsymbol{g}}(f)$.

*Proof of Theorem 3.1.* For any measurable $f \in \mathcal{F}$, $\boldsymbol{g} \in \mathcal{G}$, and $l$ bounded by $U_l < \infty$, the following inequality holds:

$$
\begin{aligned}
R_l(f) - R_{l,\boldsymbol{g}}(f) &= \mathbb{E}_{p_*(\boldsymbol{x},y)}[l(f(\boldsymbol{X}),Y)] - \mathbb{E}_{p_*(\boldsymbol{x}^\circ,y)q_{\boldsymbol{g}}(\boldsymbol{x}^{\mathrm{w}}|\boldsymbol{x}^\circ)}[l(f(\boldsymbol{X}),Y)] \\
&= \mathbb{E}_{p_*(\boldsymbol{x}^\circ,y)}\left[\int_{\mathcal{X}^{\mathrm{w}}} l(f(\boldsymbol{x}^{\mathrm{w}},\boldsymbol{X}^\circ),Y)\{p_*(\boldsymbol{x}^{\mathrm{w}}|\boldsymbol{X}^\circ,Y) - q_{\boldsymbol{g}}(\boldsymbol{x}^{\mathrm{w}}|\boldsymbol{X}^\circ)\}\mathrm{d}\boldsymbol{x}^{\mathrm{w}}\right] \\
&\leq U_l \mathbb{E}_{p_*(\boldsymbol{x}^\circ,y)}\left[\int_{\mathcal{X}^{\mathrm{w}}} \{p_*(\boldsymbol{x}^{\mathrm{w}}|\boldsymbol{X}^\circ,Y) - q_{\boldsymbol{g}}(\boldsymbol{x}^{\mathrm{w}}|\boldsymbol{X}^\circ)\}\mathrm{d}\boldsymbol{x}^{\mathrm{w}}\right] \\
&\leq U_l \mathbb{E}_{p_*(\boldsymbol{x}^\circ,y)}\left[\sqrt{\frac{1}{2}\int_{\mathcal{X}^{\mathrm{w}}} p_*(\boldsymbol{x}^{\mathrm{w}}|\boldsymbol{X}^\circ,Y)\log\frac{p_*(\boldsymbol{x}^{\mathrm{w}}|\boldsymbol{X}^\circ,Y)}{q_{\boldsymbol{g}}(\boldsymbol{x}^{\mathrm{w}}|\boldsymbol{X}^\circ)}\mathrm{d}\boldsymbol{x}^{\mathrm{w}}}\right] \\
&\leq U_l \left\{\frac{1}{2}\mathbb{E}_{p_*(\boldsymbol{x},y)}\left[\log\frac{p_*(\boldsymbol{X}^{\mathrm{w}}|\boldsymbol{X}^\circ,Y)}{q_{\boldsymbol{g}}(\boldsymbol{X}^{\mathrm{w}}|\boldsymbol{X}^\circ)}\right]\right\}^{\frac{1}{2}} \\
&= U_l \left\{\frac{1}{2}(\mathrm{LL}_{p_*} - \mathbb{E}_{p_*(\boldsymbol{x},y)}[\log q_{\boldsymbol{g}}(\boldsymbol{X}^{\mathrm{w}}|\boldsymbol{X}^\circ)])\right\}^{\frac{1}{2}}.
\end{aligned}
\tag{C.33}
$$

Here, $\mathrm{LL}_{p_*} = \mathbb{E}_{p_*(\boldsymbol{x},y)}[\log p_*(\boldsymbol{X}^{\mathrm{w}}|\boldsymbol{X}^\circ,Y)]$. In the first inequality, the condition $l(y,y') \leq U_l, \forall y, y' \in \mathcal{Y}$ is used. The second inequality employs Pinsker's inequality, while the third uses Jensen's inequality.

From the definition of $q_{\boldsymbol{g}}$, it follows that $-\mathbb{E}_{p_*(\boldsymbol{x},y)}[\log q_{\boldsymbol{g}}(\boldsymbol{X}^{\mathrm{w}}|\boldsymbol{X}^\circ)]$ can be reformulated as follows:

$$
\begin{aligned}
-\mathbb{E}_{p_*(\boldsymbol{x})}[\log q_{\boldsymbol{g}}(\boldsymbol{X}^{\mathrm{w}}|\boldsymbol{X}^\circ)] &= \log\sqrt{(2\pi)^{F^{\mathrm{w}}}\sigma^{2F^{\mathrm{w}}}} + \frac{1}{2\sigma^2}\mathbb{E}_{p_*(\boldsymbol{x})}[(\boldsymbol{X}^{\mathrm{w}} - \boldsymbol{g}(\boldsymbol{X}^\circ))^\top(\boldsymbol{x}^{\mathrm{w}} - \boldsymbol{g}(\boldsymbol{X}^\circ))] \\
&= \log\sqrt{(2\pi)^{F^{\mathrm{w}}}\sigma^{2F^{\mathrm{w}}}} + \frac{1}{2\sigma^2}\sum_{j\in[F^{\mathrm{w}}]}\mathbb{E}_{p_*(\boldsymbol{x})}[(X_j^{\mathrm{w}} - g_j(\boldsymbol{X}^\circ))^2]. \\
&= \log\sqrt{(2\pi)^{F^{\mathrm{w}}}\sigma^{2F^{\mathrm{w}}}} + \frac{1}{2\sigma^2}\sum_{j\in[F^{\mathrm{w}}]}R_{\mathrm{MSE},j}(g_j).
\end{aligned}
$$

Substituting this result into Eq. (C.33), we obtain:

$$
R_l(f) - R_{l,\boldsymbol{g}}(f) \leq U_l\left\{\frac{1}{2}\left(C_\sigma + \frac{1}{2\sigma^2}\sum_{j\in[F^{\mathrm{w}}]}R_{\mathrm{MSE},j}(g_j)\right)\right\}^{\frac{1}{2}}.
\tag{C.34}
$$

Hence, Theorem 3.1 is established.

$\square$

## C.2. Proof of Lemma 4.1

In the corresponding lemma for the theoretical analysis of discrete WFL (Lemma 4.1 in (Sugiyama & Uchida, 2025)) derives an upper bound on the 0-1 loss with respect to $g_j$ using the fact that the WFs are discrete and that $q_{\boldsymbol{g}}$ is defined as a deterministic probability distribution. In contrast, the analysis of continuous WFL cannot rely on these properties. To address this, we first derive Lemma 4.1, and by defining $q_{\boldsymbol{g}}$ as a Gaussian distribution, we subsequently derive Lemma 4.2, which plays a central role in the theoretical analysis of continuous WFL.

*Proof of Lemma 4.1.* The LHS of Eq. (4.9) can be upper-bounded as follows:

$$
\begin{aligned}
&|R_l(f) - R_{l,\boldsymbol{g}}(f)| \\
&= |\mathbb{E}_{p_*(\boldsymbol{x},y)}[l(f(\boldsymbol{X}),Y)] - \mathbb{E}_{p_*(\boldsymbol{x}^{\mathrm{o}},y)q_{\boldsymbol{g}}(\boldsymbol{x}^{\mathrm{w}}|\boldsymbol{x}^{\mathrm{o}})}[l(f(\boldsymbol{X}),Y)]| \\
&= \left| \mathbb{E}_{p_*(\boldsymbol{x}^{\mathrm{o}},y)}\left[ \int_{\mathcal{X}^{\mathrm{w}}} l(f(\boldsymbol{x}^{\mathrm{w}},\boldsymbol{X}^{\mathrm{o}}),Y)\{p_*(\boldsymbol{x}^{\mathrm{w}}|\boldsymbol{X}^{\mathrm{o}},Y) - q_{\boldsymbol{g}}(\boldsymbol{x}^{\mathrm{w}}|\boldsymbol{X}^{\mathrm{o}})\}\mathrm{d}\boldsymbol{x}^{\mathrm{w}} \right] \right| \\
&\leq \mathbb{E}_{p_*(\boldsymbol{x}^{\mathrm{o}},y)}\left[ \int_{\mathcal{X}^{\mathrm{w}}} l(f(\boldsymbol{x}^{\mathrm{w}},\boldsymbol{X}^{\mathrm{o}}),Y)|p_*(\boldsymbol{x}^{\mathrm{w}}|\boldsymbol{X}^{\mathrm{o}},Y) - q_{\boldsymbol{g}}(\boldsymbol{x}^{\mathrm{w}}|\boldsymbol{X}^{\mathrm{o}})|\mathrm{d}\boldsymbol{x}^{\mathrm{w}} \right] \\
&= \mathbb{E}_{p_*(\boldsymbol{x}^{\mathrm{o}},y)}\left[ \int_{\mathcal{X}^{\mathrm{w}}} l(f(\boldsymbol{x}^{\mathrm{w}},\boldsymbol{X}^{\mathrm{o}}),Y)\left( \sqrt{p_*(\boldsymbol{x}^{\mathrm{w}}|\boldsymbol{X}^{\mathrm{o}},Y)} + \sqrt{q_{\boldsymbol{g}}(\boldsymbol{x}^{\mathrm{w}}|\boldsymbol{X}^{\mathrm{o}})} \right) \right. \\
&\qquad\qquad \left. \cdot \left| \sqrt{p_*(\boldsymbol{x}^{\mathrm{w}}|\boldsymbol{X}^{\mathrm{o}},Y)} - \sqrt{q_{\boldsymbol{g}}(\boldsymbol{x}^{\mathrm{w}}|\boldsymbol{X}^{\mathrm{o}})} \right| \mathrm{d}\boldsymbol{x}^{\mathrm{w}} \right] \\
&= \mathbb{E}_{p_*(\boldsymbol{x}^{\mathrm{o}},y)}\left[ \int_{\mathcal{X}^{\mathrm{w}}} l(f(\boldsymbol{x}^{\mathrm{w}},\boldsymbol{X}^{\mathrm{o}}),Y)\sqrt{p_*(\boldsymbol{x}^{\mathrm{w}}|\boldsymbol{X}^{\mathrm{o}},Y)}\left| \sqrt{p_*(\boldsymbol{x}^{\mathrm{w}}|\boldsymbol{X}^{\mathrm{o}},Y)} - \sqrt{q_{\boldsymbol{g}}(\boldsymbol{x}^{\mathrm{w}}|\boldsymbol{X}^{\mathrm{o}})} \right| \mathrm{d}\boldsymbol{x}^{\mathrm{w}} \right] \\
&\quad + \mathbb{E}_{p_*(\boldsymbol{x}^{\mathrm{o}},y)}\left[ \int_{\mathcal{X}^{\mathrm{w}}} l(f(\boldsymbol{x}^{\mathrm{w}},\boldsymbol{X}^{\mathrm{o}}),Y)\sqrt{q_{\boldsymbol{g}}(\boldsymbol{x}^{\mathrm{w}}|\boldsymbol{X}^{\mathrm{o}})}\left| \sqrt{p_*(\boldsymbol{x}^{\mathrm{w}}|\boldsymbol{X}^{\mathrm{o}},Y)} - \sqrt{q_{\boldsymbol{g}}(\boldsymbol{x}^{\mathrm{w}}|\boldsymbol{X}^{\mathrm{o}})} \right| \mathrm{d}\boldsymbol{x}^{\mathrm{w}} \right] \\
&\leq \left\{ \underbrace{\mathbb{E}_{p_*(\boldsymbol{x},y)}[\{l(f(\boldsymbol{X}),Y)\}^2]}_{\text{(a1)}} \cdot \underbrace{\mathbb{E}_{p_*(\boldsymbol{x}^{\mathrm{o}},y)}\left[ \int_{\mathcal{X}^{\mathrm{w}}} \left| \sqrt{p_*(\boldsymbol{x}^{\mathrm{w}}|\boldsymbol{X}^{\mathrm{o}},Y)} - \sqrt{q_{\boldsymbol{g}}(\boldsymbol{x}^{\mathrm{w}}|\boldsymbol{X}^{\mathrm{o}})} \right|^2 \mathrm{d}\boldsymbol{x}^{\mathrm{w}} \right]}_{\text{(a2)}} \right\}^{\frac{1}{2}} \\
&\quad + \left\{ \underbrace{\mathbb{E}_{p_*(\boldsymbol{x}^{\mathrm{o}},y)q_{\boldsymbol{g}}(\boldsymbol{x}^{\mathrm{w}}|\boldsymbol{x}^{\mathrm{o}})}[\{l(f(\boldsymbol{X}),Y)\}^2]}_{\text{(a3)}} \cdot \underbrace{\mathbb{E}_{p_*(\boldsymbol{x}^{\mathrm{o}},y)}\left[ \int_{\mathcal{X}^{\mathrm{w}}} \left| \sqrt{p_*(\boldsymbol{x}^{\mathrm{w}}|\boldsymbol{X}^{\mathrm{o}},Y)} - \sqrt{q_{\boldsymbol{g}}(\boldsymbol{x}^{\mathrm{w}}|\boldsymbol{X}^{\mathrm{o}})} \right|^2 \mathrm{d}\boldsymbol{x}^{\mathrm{w}} \right]}_{\text{(a2)}} \right\}^{\frac{1}{2}}. \quad \text{(C.35)}
\end{aligned}
$$

Here, in the second inequality, Cauchy-Schwarz inequality is used since $l$, $p_*$ and $q_{\boldsymbol{g}}$ are measurable functions.

The terms (a1) and (a3) in Eq. (C.35) can be bounded as follows:

$$
\begin{aligned}
\text{(a1)} &= \mathbb{E}_{p_*(\boldsymbol{x},y)}[\{l(f(\boldsymbol{X}),Y)\}^2] = \mathbb{E}_{p_*(\boldsymbol{x},y)}[\{l(f(\boldsymbol{X}),Y)\}^2 - 0] \\
&\leq 2U_l \mathbb{E}_{p_*(\boldsymbol{x},y)}[l(f(\boldsymbol{X}),Y)] = 2U_l R_l(f).
\end{aligned}
\quad \text{(C.36)}
$$

$$
\begin{aligned}
\text{(a3)} &= \mathbb{E}_{p_*(\boldsymbol{x}^{\mathrm{o}},y)q_{\boldsymbol{g}}(\boldsymbol{x}^{\mathrm{w}}|\boldsymbol{x}^{\mathrm{o}})}[\{l(f(\boldsymbol{X}),Y)\}^2] = \mathbb{E}_{p_*(\boldsymbol{x}^{\mathrm{o}},y)q_{\boldsymbol{g}}(\boldsymbol{x}^{\mathrm{w}}|\boldsymbol{x}^{\mathrm{o}})}[\{l(f(\boldsymbol{X}),Y)\}^2 - 0] \\
&\leq 2U_l \mathbb{E}_{p_*(\boldsymbol{x}^{\mathrm{o}},y)q_{\boldsymbol{g}}(\boldsymbol{x}^{\mathrm{w}}|\boldsymbol{x}^{\mathrm{o}})}[l(f(\boldsymbol{X}),Y)] = 2U_l R_{l,\boldsymbol{g}}(f).
\end{aligned}
\quad \text{(C.37)}
$$

Here in terms (a1) and (a3), the fact that the function $x \mapsto x^2$ is $2U_l$-Lipschitz continuous on the interval $[0, U_l]$ was utilized. Applying the above inequalities to Eq. (C.35), Lemma 4.1 can be showed:

$$
|R_l(f) - R_{l,\boldsymbol{g}}(f)| \leq \left( \sqrt{R_l(f)} + \sqrt{R_{l,\boldsymbol{g}}(f)} \right)\left\{ 2U_l \mathbb{E}_{p_*(\boldsymbol{x}^{\mathrm{o}},y)}\left[ D_{\mathrm{H}}^2(p_*(\boldsymbol{X}^{\mathrm{w}}|\boldsymbol{X}^{\mathrm{o}},Y), q_{\boldsymbol{g}}(\boldsymbol{X}^{\mathrm{w}}|\boldsymbol{X}^{\mathrm{o}})) \right] \right\}^{\frac{1}{2}}. \quad \text{(C.38)}
$$

Here, $D_{\mathrm{H}}$ represents the Hellinger distance.

$\square$

## C.3. Proof of Lemma 4.2

*Proof of Lemma 4.2.* The term $\mathbb{E}_{p_*(\boldsymbol{x}^{\mathrm{o}},y)}\left[ D_{\mathrm{H}}^2(p_*(\boldsymbol{X}^{\mathrm{w}}|\boldsymbol{X}^{\mathrm{o}},Y), q_{\boldsymbol{g}}(\boldsymbol{X}^{\mathrm{w}}|\boldsymbol{X}^{\mathrm{o}})) \right]$ in Eq. (4.9) can be upper-bounded as follows:

$$
\begin{aligned}
\mathbb{E}_{p_*(\boldsymbol{x}^{\mathrm{o}},y)}\left[ D_{\mathrm{H}}^2(p_*(\boldsymbol{X}^{\mathrm{w}}|\boldsymbol{X}^{\mathrm{o}},Y), q_{\boldsymbol{g}}(\boldsymbol{X}^{\mathrm{w}}|\boldsymbol{X}^{\mathrm{o}})) \right] &\leq \mathbb{E}_{p_*(\boldsymbol{x}^{\mathrm{o}},y)}\left[ \int_{\mathcal{X}^{\mathrm{w}}} p_*(\boldsymbol{x}^{\mathrm{w}}|\boldsymbol{X}^{\mathrm{o}},Y)\log\frac{p_*(\boldsymbol{x}^{\mathrm{w}}|\boldsymbol{X}^{\mathrm{o}},Y)}{q_{\boldsymbol{g}}(\boldsymbol{x}^{\mathrm{w}}|\boldsymbol{X}^{\mathrm{o}})}\mathrm{d}\boldsymbol{x}^{\mathrm{w}} \right] \\
&= \mathrm{LL}_{p_*} - \mathbb{E}_{p_*(\boldsymbol{x})}[\log q_{\boldsymbol{g}}(\boldsymbol{X}^{\mathrm{w}}|\boldsymbol{X}^{\mathrm{o}})]
\end{aligned}
$$

$$\text{(C.39)}$$

Here, $\mathrm{LL}_{p_*} = \mathbb{E}_{p_*(\boldsymbol{x},y)}[\log p_*(\boldsymbol{X}^{\mathrm{w}}|\boldsymbol{X}^{\mathrm{o}}, Y)]$. Defining $q_{\boldsymbol{g}}$ by the PDF of $\mathcal{N}(\boldsymbol{g}(\boldsymbol{x}^{\mathrm{o}}), \sigma^2 I_{F^{\mathrm{w}} \times F^{\mathrm{w}}})$ where $\sigma^2 \in \mathbb{R}_+$, the term $-\mathbb{E}_{p_*(\boldsymbol{x})}[\log q_{\boldsymbol{g}}(\boldsymbol{X}^{\mathrm{w}}|\boldsymbol{X}^{\mathrm{o}})]$ can rewritten as follows:

$$
\begin{aligned}
-\mathbb{E}_{p_*(\boldsymbol{x})}[\log q_{\boldsymbol{g}}(\boldsymbol{X}^{\mathrm{w}}|\boldsymbol{X}^{\mathrm{o}})] &= \log \sqrt{(2\pi)^{F^{\mathrm{w}}} \sigma^{2F^{\mathrm{w}}}} + \frac{1}{2\sigma^2} \mathbb{E}_{p_*(\boldsymbol{x})}[(\boldsymbol{X}^{\mathrm{w}} - \boldsymbol{g}(\boldsymbol{X}^{\mathrm{o}}))^\top (\boldsymbol{X}^{\mathrm{w}} - \boldsymbol{g}(\boldsymbol{X}^{\mathrm{o}}))] \\
&= \log \sqrt{(2\pi)^{F^{\mathrm{w}}} \sigma^{2F^{\mathrm{w}}}} + \frac{1}{2\sigma^2} \sum_{j \in [F^{\mathrm{w}}]} \mathbb{E}_{p_*(\boldsymbol{x})}[(X_j^{\mathrm{w}} - g_j(\boldsymbol{X}^{\mathrm{o}}))^2] \\
&= \log \sqrt{(2\pi)^{F^{\mathrm{w}}} \sigma^{2F^{\mathrm{w}}}} + \frac{1}{2\sigma^2} \sum_{j \in [F^{\mathrm{w}}]} R_{\mathrm{MSE},j}(g_j).
\end{aligned}
$$

Substituting this result into Eq. (C.39), we obtain:

$$
\mathbb{E}_{p_*(\boldsymbol{x}^{\mathrm{o}}, y)}\left[\{D_{\mathrm{H}}(p_*(\boldsymbol{X}^{\mathrm{w}}|\boldsymbol{X}^{\mathrm{o}}, Y), q_{\boldsymbol{g}}(\boldsymbol{X}^{\mathrm{w}}|\boldsymbol{X}^{\mathrm{o}}))\}^2\right] \leq \mathrm{LL}_{p_*} + Z(\sigma^2) + \frac{1}{2\sigma^2} \sum_{j \in [F^{\mathrm{w}}]} R_{\mathrm{MSE},j}(g_j). \tag{C.40}
$$

Here, $Z(\sigma^2) := \log \sqrt{(2\pi)^{F^{\mathrm{w}}} \sigma^{2F^{\mathrm{w}}}}$ Applying the above result to Eq. (4.9), we establish Eq. (4.10) in Lemma 4.2:

$$
|R_l(f) - R_{l,\boldsymbol{g}}(f)| \leq \left(\sqrt{R_l(f)} + \sqrt{R_{l,\boldsymbol{g}}(f)}\right)\left\{2U_l\left(\mathrm{LL}_{p_*} + Z(\sigma^2) + \frac{1}{2\sigma^2} \sum_{j \in [F^{\mathrm{w}}]} R_{\mathrm{MSE},j}(g_j)\right)\right\}^{\frac{1}{2}}. \tag{C.41}
$$

$\square$

## C.4. Proof of Theorem 4.3

Using Lemma 4.2, we establish the following lemma:

**Lemma C.1.** *Let $q_{\boldsymbol{g}}$ be the PDF of $\mathcal{N}(\boldsymbol{g}(\boldsymbol{x}^{\mathrm{o}}), \sigma^2 I_{F^{\mathrm{w}} \times F^{\mathrm{w}}})$, where $\sigma^2 \in \mathbb{R}_+$. For any measurable $f \in \mathcal{F}$, $\boldsymbol{g} \in \mathcal{G}$ and $l$ bounded by $U_l < \infty$, the following inequality holds:*

$$
\begin{aligned}
&|R_l(f) - R_{l,\boldsymbol{g}}(f)| \\
&\leq \left\{2\sqrt{R_l(f)} + \left(2U_l\left(C_\sigma + \frac{1}{2\sigma^2}\sum_{j\in[F^{\mathrm{w}}]} R_{\mathrm{MSE},j}(g_j)\right)\right)^{\frac{1}{2}}\right\}\left\{2U_l\left(C_\sigma + \frac{1}{2\sigma^2}\sum_{j\in[F^{\mathrm{w}}]} R_{\mathrm{MSE},j}(g_j)\right)\right\}^{\frac{1}{2}}.
\end{aligned} \tag{C.42}
$$

*Here, $C_\sigma = \mathbb{E}_{p_*(\boldsymbol{x},y)}[\log p_*(\boldsymbol{X}^{\mathrm{w}}|\boldsymbol{X}^{\mathrm{o}}, Y)] + \log \sqrt{(2\pi)^{F^{\mathrm{w}}} \sigma^{2F^{\mathrm{w}}}}$.*

*Proof of Lemma C.1.* From Eq. (C.41), for any measurable $f \in \mathcal{F}$, $\boldsymbol{g} \in \mathcal{G}$ and $l$ bounded by $U_l < \infty$, the following holds:

$$
\left|\sqrt{R_l(f)} - \sqrt{R_{l,\boldsymbol{g}}(f)}\right| \leq \left\{2U_l\left(\mathrm{LL}_{p_*} + Z(\sigma^2) + \frac{1}{2\sigma^2} \sum_{j \in [F^{\mathrm{w}}]} R_{\mathrm{MSE},j}(g_j)\right)\right\}^{\frac{1}{2}}. \tag{C.43}
$$

Thus, $\sqrt{R_{l,\boldsymbol{g}}(f)}$ can be upper-bounded as follows:

$$
\begin{aligned}
\sqrt{R_{l,\boldsymbol{g}}(f)} &= \sqrt{R_l(f)} + \left|\sqrt{R_l(f)} - \sqrt{R_{l,\boldsymbol{g}}(f)}\right| \\
&\leq \sqrt{R_l(f)} + \left\{2U_l\left(\mathrm{LL}_{p_*} + Z(\sigma^2) + \frac{1}{2\sigma^2} \sum_{j \in [F^{\mathrm{w}}]} R_{\mathrm{MSE},j}(g_j)\right)\right\}^{\frac{1}{2}}.
\end{aligned} \tag{C.44}
$$

Applying Eq. (C.44) to Eq. (4.10) in Lemma 4.2, we conclude that Lemma C.1 holds:

$$|R_l(f) - R_{l,\boldsymbol{g}}(f)|$$

$$\leq \left\{ 2\sqrt{R_l(f)} + \left( 2U_l\left( \mathrm{LL}_{p_*} + Z(\sigma^2) + \frac{1}{2\sigma^2} \sum_{j \in [F^{\mathrm{w}}]} R_{\mathrm{MSE},j}(g_j) \right) \right)^{\frac{1}{2}} \right\}$$

$$\times \left\{ 2U_l\left( \mathrm{LL}_{p_*} + Z(\sigma^2) + \frac{1}{2\sigma^2} \sum_{j \in [F^{\mathrm{w}}]} R_{\mathrm{MSE},j}(g_j) \right) \right\}^{\frac{1}{2}}. \tag{C.45}$$

$$\square$$

Using Lemma C.1, Theorem 4.3 is demonstrated as follows:

*Proof of Theorem 4.3.* In the ordinary supervised learning setting described in Section 2.1, we define the empirical risk minimizer as follows:

$$f_S := \arg\min_{f \in \mathcal{F}} \widehat{R}_l(f).$$

The LHS of Eq. (4.11) can be decomposed as follows:

$$R_{l,\boldsymbol{g}}(f_{\boldsymbol{g},\overline{S}}) - R_l(f_{\mathcal{F}}) = \underbrace{R_{l,\boldsymbol{g}}(f_{\boldsymbol{g},\overline{S}}) - \widehat{R}_{l,\boldsymbol{g}}(f_{\boldsymbol{g},\overline{S}})}_{\text{(a1)}} + \underbrace{\widehat{R}_{l,\boldsymbol{g}}(f_{\boldsymbol{g},\overline{S}}) - R_{l,\boldsymbol{g}}(f_S)}_{\text{(a2)}}$$

$$+ \underbrace{R_{l,\boldsymbol{g}}(f_S) - R_l(f_S)}_{\text{(a3)}} + \underbrace{R_l(f_S) - R_l(f_{\mathcal{F}})}_{\text{(a4)}}. \tag{C.46}$$

The terms (a1) and (a2) in Eq. (C.46) can be upper-bounded as follows:

$$\text{(a1)} \leq \max_{f \in \mathcal{F}} |R_{l,\boldsymbol{g}}(f) - \widehat{R}_{l,\boldsymbol{g}}(f)|,$$

$$\text{(a2)} \leq \widehat{R}_{l,\boldsymbol{g}}(f_S) - R_{l,\boldsymbol{g}}(f_S) \leq \max_{f \in \mathcal{F}} |R_{l,\boldsymbol{g}}(f) - \widehat{R}_{l,\boldsymbol{g}}(f)|.$$

The term (a3) in Eq. (C.46) can be upper-bounded using Lemma C.1 as follows:

$$\text{(a3)} \leq \left\{ 2\sqrt{R_l(f_S)} + \left( 2U_l\left( C_\sigma + \frac{1}{2\sigma^2} \sum_{j \in [F^{\mathrm{w}}]} R_{\mathrm{MSE},j}(g_j) \right) \right)^{\frac{1}{2}} \right\} \left\{ 2U_l\left( C_\sigma + \frac{1}{2\sigma^2} \sum_{j \in [F^{\mathrm{w}}]} R_{\mathrm{MSE},j}(g_j) \right) \right\}^{\frac{1}{2}}. \tag{C.47}$$

Here, $R_l(f_S)$ can be further bounded as follows:

$$\begin{aligned} R_l(f_S) &= R_l(f_S) - \widehat{R}_l(f_S) + \widehat{R}_l(f_S) - R_l(f_{\mathcal{F}}) + R_l(f_{\mathcal{F}}) \\ &\leq R_l(f_S) - \widehat{R}_l(f_S) + \widehat{R}_l(f_{\mathcal{F}}) - R_l(f_{\mathcal{F}}) + R_l(f_{\mathcal{F}}) \\ &\leq R_l(f_{\mathcal{F}}) + 2\max_{f \in \mathcal{F}} |R_l(f) - \widehat{R}_l(f)|. \end{aligned} \tag{C.48}$$

Applying the above inequality to Eq. (C.47), the term (a3) in Eq. (C.46) can be bounded as follows:

$$\text{(a3)} \leq \left\{ 2\left( R_l(f_{\mathcal{F}}) + 2\max_{f \in \mathcal{F}} |R_l(f) - \widehat{R}_l(f)| \right)^{\frac{1}{2}} + \left( 2U_l\left( C_\sigma + \frac{1}{2\sigma^2} \sum_{j \in [F^{\mathrm{w}}]} R_{\mathrm{MSE},j}(g_j) \right) \right)^{\frac{1}{2}} \right\}$$

$$\times \left\{ 2U_l\left( C_\sigma + \frac{1}{2\sigma^2} \sum_{j \in [F^{\mathrm{w}}]} R_{\mathrm{MSE},j}(g_j) \right) \right\}^{\frac{1}{2}}. \tag{C.49}$$

Similarly, the term (a4) in Eq. (C.46) can be bounded as follows:

$$R_l(f_S) - R_l(f_\mathcal{F}) \leq 2 \max_{f \in \mathcal{F}} |R_l(f) - \widehat{R}_l(f)|.$$

Combining the results above, the LHS of Eq. (C.46) can be upper-bounded as follows:

$$
\begin{aligned}
&R_{l,\boldsymbol{g}}(f_{\boldsymbol{g},\overline{S}}) - R_l(f_\mathcal{F}) \\
&\leq 2 \max_{f \in \mathcal{F}} |R_{l,\boldsymbol{g}}(f) - \widehat{R}_{l,\boldsymbol{g}}(f)| + 2 \max_{f \in \mathcal{F}} |R_l(f) - \widehat{R}_l(f)| \\
&\quad + \left\{ 2\left(R_l(f_\mathcal{F}) + 2\max_{f \in \mathcal{F}} |R_l(f) - \widehat{R}_l(f)|\right)^{\frac{1}{2}} + \left(2U_l\left(C_\sigma + \frac{1}{2\sigma^2}\sum_{j \in [F^{\mathrm{w}}]} R_{\mathrm{MSE},j}(g_j)\right)\right)^{\frac{1}{2}} \right\} \\
&\qquad\qquad \times \left\{ 2U_l\left(C_\sigma + \frac{1}{2\sigma^2}\sum_{j \in [F^{\mathrm{w}}]} R_{\mathrm{MSE},j}(g_j)\right) \right\}^{\frac{1}{2}}.
\end{aligned}
\tag{C.50}
$$

Using the uniform law of large numbers (Mohri et al., 2018), for any $\delta \in (0,1)$, the following inequality holds with probability at least $1 - \delta/2$:

$$\max_{f \in \mathcal{F}} |R_{l,\boldsymbol{g}}(f) - \widehat{R}_{l,\boldsymbol{g}}(f)| \leq 2\mathfrak{R}_n^{\boldsymbol{g}}(\widetilde{\mathcal{F}}_l) + U_l\sqrt{\frac{\log(4/\delta)}{2n}},$$

$$\max_{f \in \mathcal{F}} |R_l(f) - \widehat{R}_l(f)| \leq 2\mathfrak{R}_n^*(\widetilde{\mathcal{F}}_l) + U_l\sqrt{\frac{\log(4/\delta)}{2n}}.$$

Here, $\widetilde{\mathcal{F}}_l := \{(\boldsymbol{x}, y) \mapsto l(f(\boldsymbol{x}), y) : f \in \mathcal{F}\}$. Additionally, from the assumption that $l$ is $L_l$-Lipschitz continuous and Lemma 26.9 in (Shalev-Shwartz & Ben-David, 2014), it follows that $\mathfrak{R}_n^*(\widetilde{\mathcal{F}}_l) \leq L_l\mathfrak{R}_n^*(\mathcal{F})$ and $\mathfrak{R}_n^{\boldsymbol{g}}(\widetilde{\mathcal{F}}_l) \leq L_l\mathfrak{R}_n^{\boldsymbol{g}}(\mathcal{F})$. Thus, for any $\delta \in (0,1)$, the following inequality holds with probability at least $1 - \delta$:

$$
\begin{aligned}
&R_{l,\boldsymbol{g}}(f_{\boldsymbol{g},\overline{S}}) - R_l(f_\mathcal{F}) \\
&\leq 4\left(L_l\mathfrak{R}_n^*(\mathcal{F}) + L_l\mathfrak{R}_n^{\boldsymbol{g}}(\mathcal{F}) + U_l\sqrt{\frac{\log(4/\delta)}{2n}}\right) \\
&\quad + \left\{ 2\left(R_l(f_\mathcal{F}) + 4L_l\mathfrak{R}_n^*(\mathcal{F}) + 2U_l\sqrt{\frac{\log(4/\delta)}{2n}}\right)^{\frac{1}{2}} + \left(2U_l\left(C_\sigma + \frac{1}{2\sigma^2}\sum_{j \in [F^{\mathrm{w}}]} R_{\mathrm{MSE},j}(g_j)\right)\right)^{\frac{1}{2}} \right\} \\
&\qquad\qquad \times \left\{ 2U_l\left(C_\sigma + \frac{1}{2\sigma^2}\sum_{j \in [F^{\mathrm{w}}]} R_{\mathrm{MSE},j}(g_j)\right) \right\}^{\frac{1}{2}}.
\end{aligned}
\tag{C.51}
$$

$\square$

## C.5. Proof of Theorem 4.4

*Proof of Theorem 4.4.* Firstly, since $\mathcal{X}_j^{\mathrm{w}}$ is a set of finite-precision decimals, for any $j \in [F^{\mathrm{w}}]$, and $\sigma^2 = (2\pi)^{-1}$, it follows that $C_\sigma \leq 0$.

The remainder of this proof follows a structure similar to that of the proof for discrete WFL (Sugiyama & Uchida, 2025).

By assumption, there exist true deterministic functions $g_j^* : \mathcal{X}^{\mathrm{o}} \to \mathcal{X}_j^{\mathrm{w}}$ for any $j \in [F^{\mathrm{w}}]$, and $(g_1^*, \ldots, g_{F^{\mathrm{w}}}^*) \in \mathcal{G}$. Therefore, when $\boldsymbol{g}_{\overline{S}} = (g_{\overline{S},1}, \ldots, g_{\overline{S},F^{\mathrm{w}}})$ is obtained by the methods that achieve consistency (Mohri et al., 2018; Cheng et al., 2023), the following holds:

$$n \to \infty, \quad R_{\mathrm{MSE},j}(g_{\overline{S},j}) \to 0, \quad \forall j \in [F^{\mathrm{w}}]. \tag{C.52}$$

By assumption, there exists a true deterministic function $f^* : \mathcal{X} \to \mathcal{Y}$ for label prediction, and $f^* \in \mathcal{F}$. Hence, the following holds:

$$R_l(f_{\mathcal{F}}) = 0. \tag{C.53}$$

Thus, if $\mathfrak{R}_n^*(\mathcal{F})$ and $\mathfrak{R}_n^g(\mathcal{F})$ are monotonically decreasing with respect to $n$ and converge to 0, the error bound established in Theorem 4.3 converges to 0 as $n \to \infty$.

$\square$

## C.6. Proof of Theorem 4.5

For a weak dataset $\overline{S}$ and a positive real-valued vector $\boldsymbol{r}$, define the following feature estimation models:

$$\boldsymbol{g}_{\overline{S}}^{(\boldsymbol{r})} := (g_{\overline{S},1}^{(r_1)}, \ldots, g_{\overline{S},F^{\mathrm{w}}}^{(r_{F^{\mathrm{w}}})}), \tag{C.54}$$

$$\text{where } g_{\overline{S},j}^{(r_j)} := \operatorname*{arg\,min}_{g_j \in \mathcal{G}(r_j, \overline{S})} \widehat{R}_{\mathrm{MSE},j}(g_j), \ \ \forall j \in [F^{\mathrm{w}}]. \tag{C.55}$$

Using Lemma C.1, we establish Theorem 4.5 as follows.

*Proof.* Proof of Theorem 4.5

The LHS of Eq. (4.13) can be reformulated as follows:

$$R_{l,f}(\boldsymbol{g}_{f,\overline{S}}^{(\boldsymbol{r})}) - R_l(f) = \underbrace{R_{l,f}(\boldsymbol{g}_{f,\overline{S}}^{(\boldsymbol{r})}) - \widehat{R}_{l,f}(\boldsymbol{g}_{f,\overline{S}}^{(\boldsymbol{r})})}_{(a1)} + \underbrace{\widehat{R}_{l,f}(\boldsymbol{g}_{f,\overline{S}}^{(\boldsymbol{r})}) - R_{l,f}(\boldsymbol{g}_{\overline{S}}^{(\boldsymbol{r})})}_{(a2)} + \underbrace{R_{l,f}(\boldsymbol{g}_{\overline{S}}^{(\boldsymbol{r})}) - R_l(f)}_{(a3)}. \tag{C.56}$$

The terms (a1) and (a2) in Eq. (C.56) can be upper-bounded as follows:

$$(a1) \leq \max_{\boldsymbol{g} \in \mathcal{G}(\boldsymbol{r}, \overline{S})} |R_{l,f}(\boldsymbol{g}) - \widehat{R}_{l,f}(\boldsymbol{g})|, \tag{C.57}$$

$$(a2) \leq \widehat{R}_{l,f}(\boldsymbol{g}_{\overline{S}}^{(\boldsymbol{r})}) - R_{l,f}(\boldsymbol{g}_{\overline{S}}^{(\boldsymbol{r})}) \leq \max_{\boldsymbol{g} \in \mathcal{G}(\boldsymbol{r}, \overline{S})} |R_{l,f}(\boldsymbol{g}) - \widehat{R}_{l,f}(\boldsymbol{g})|. \tag{C.58}$$

The term (a3) in Eq.(C.56) can be upper-bounded using Lemma C.1 as follows:

$$
\begin{aligned}
(a3) &\leq |R_{l,f}(\boldsymbol{g}_{\overline{S}}^{(\boldsymbol{r})}) - R_l(f)| \\
&\leq \left\{ 2\sqrt{R_l(f)} + \left( 2U_l \left( C_\sigma + \frac{1}{2\sigma^2} \sum_{j \in [F^{\mathrm{w}}]} R_{\mathrm{MSE},j}(g_{\overline{S},j}^{(r_j)}) \right) \right)^{\frac{1}{2}} \right\} \left\{ 2U_l \left( C_\sigma + \frac{1}{2\sigma^2} \sum_{j \in [F^{\mathrm{w}}]} R_{\mathrm{MSE},j}(g_{\overline{S},j}^{(r_j)}) \right) \right\}^{\frac{1}{2}}.
\end{aligned}
\tag{C.59}
$$

Therefore, by applying Eqs. (C.57), (C.58), and (C.59) to Eq. (C.56), we obtain:

$$
\begin{aligned}
&R_{l,f}(\boldsymbol{g}_{f,\overline{S}}^{(\boldsymbol{r})}) - R_l(f) \\
&\leq 2 \max_{\boldsymbol{g} \in \mathcal{G}(\boldsymbol{r}, \overline{S})} |R_{l,f}(\boldsymbol{g}) - \widehat{R}_{l,f}(\boldsymbol{g})| \\
&\quad + \left\{ 2\sqrt{R_l(f)} + \left( 2U_l \left( C_\sigma + \frac{1}{2\sigma^2} \sum_{j \in [F^{\mathrm{w}}]} R_{\mathrm{MSE},j}(g_{\overline{S},j}^{(r_j)}) \right) \right)^{\frac{1}{2}} \right\} \left\{ 2U_l \left( C_\sigma + \frac{1}{2\sigma^2} \sum_{j \in [F^{\mathrm{w}}]} R_{\mathrm{MSE},j}(g_{\overline{S},j}^{(r_j)}) \right) \right\}^{\frac{1}{2}}.
\end{aligned}
\tag{C.60}
$$

From the uniform law of large numbers (Mohri et al., 2018), for any $\delta \in (0, 1)$, the following holds with a probability of at least $1 - \delta$:

$$\max_{\boldsymbol{g} \in \mathcal{G}(\boldsymbol{r}, \overline{S})} |R_{l,f}(\boldsymbol{g}) - \widehat{R}_{l,f}(\boldsymbol{g})| \leq 2\mathfrak{R}_n^*(\widetilde{\mathcal{G}}_{l,f}(\boldsymbol{r}, \overline{S})) + U_l \sqrt{\frac{\log(2/\delta)}{2n}}. \tag{C.61}$$

Furthermore, by applying Eq. (C.61) to Eq. (C.60), we obtain that, for any $\delta \in (0, 1)$, with probability at least $1 - \delta$, Eq. (4.13) holds:

$$
\begin{aligned}
&R_{l,f}(\boldsymbol{g}_{f,\overline{S}}^{(\boldsymbol{r})}) - R_l(f) \\
&\leq 4\mathfrak{R}_n^*(\widetilde{\mathcal{G}}_{l,f}(\boldsymbol{r}, \overline{S})) + 2U_l\sqrt{\frac{\log(2/\delta)}{2n}} \\
&+ \left\{ 2\sqrt{R_l(f)} + \left( 2U_l\left( C_\sigma + \frac{1}{2\sigma^2}\sum_{j\in[F^{\mathrm{w}}]} R_{\mathrm{MSE},j}(g_{\overline{S},j}^{(r_j)}) \right) \right)^{\frac{1}{2}} \right\} \left\{ 2U_l\left( C_\sigma + \frac{1}{2\sigma^2}\sum_{j\in[F^{\mathrm{w}}]} R_{\mathrm{MSE},j}(g_{\overline{S},j}^{(r_j)}) \right) \right\}^{\frac{1}{2}}.
\end{aligned}
\tag{C.62}
$$

$\square$

## C.7. Proof of Eq. 4.14

*Proof of Eq. 4.14.* Since, by the assumption about $\mathcal{G}_j(r_j, \overline{S}_j)$ and the fact that $g_{\overline{S},j}^{(r_j)} \in \mathcal{G}_j(r_j, \overline{S}_j)$, the following holds:

$$
\widehat{R}_{\mathrm{MSE},j}(g_{\overline{S},j}^{(r_j)}) = \min_{g_j \in \mathcal{G}_j} \widehat{R}_{\mathrm{MSE},j}(g_j).
\tag{C.63}
$$

Moreover, by the assumption about $\mathcal{G}_j$, the following holds:

$$
R_{\mathrm{MSE},j}(g_{\mathcal{G}_j}) = 0,
\tag{C.64}
$$

where $g_{\mathcal{G}_j} := \arg\min_{g_j \in \mathcal{G}_j} R_{\mathrm{MSE},j}(g_j)$. In this case, the LHS of Eq. 4.14 can be upper bounded as follows:

$$
\begin{aligned}
R_{\mathrm{MSE},j}(g_{\overline{S},j}^{(r_j)}) &= R_{\mathrm{MSE},j}(g_{\overline{S},j}^{(r_j)}) - \widehat{R}_{\mathrm{MSE},j}(g_{\overline{S},j}^{(r_j)}) + \widehat{R}_{\mathrm{MSE},j}(g_{\overline{S},j}^{(r_j)}) - R_{\mathrm{MSE},j}(g_{\mathcal{G}_j}) \\
&\leq R_{\mathrm{MSE},j}(g_{\overline{S},j}^{(r_j)}) - \widehat{R}_{\mathrm{MSE},j}(g_{\overline{S},j}^{(r_j)}) + \widehat{R}_{\mathrm{MSE},j}(g_{\mathcal{G}_j}) - R_{\mathrm{MSE},j}(g_{\mathcal{G}_j}) \\
&\leq 2\max_{g_j \in \mathcal{G}_j} |R_{\mathrm{MSE},j}(g_j) - \widehat{R}_{\mathrm{MSE},j}(g_j)|.
\end{aligned}
\tag{C.65}
$$

By applying Theorem 11.3 in (Mohri et al., 2018) to the RHS of the last inequality above, we obtain that, for any $\delta \in (0, 1)$, with probability at least $1 - \delta$, the following holds:

$$
R_{\mathrm{MSE},j}(g_{\overline{S},j}^{(r_j)}) \leq 8U_{l_{\mathrm{MSE}}}\mathfrak{R}_{n_j'}^*(\mathcal{G}_j) + 2U_{l_{\mathrm{MSE}}}\sqrt{\frac{\log(1/\delta)}{2n_j'}}.
\tag{C.66}
$$

$\square$

## C.8. Proof of Theorem 4.6

*Proof of Theorem 4.6.* Firstly, since $\mathcal{X}_j^{\mathrm{w}}$ is a set of finite-precision decimals, for any $j \in [F^{\mathrm{w}}]$, and $\sigma^2 = (2\pi)^{-1}$, it follows that $C_\sigma \leq 0$.

The remainder of this proof follows a structure similar to that of the proof for discrete WFL (Sugiyama & Uchida, 2025). From assumption, there exist true deterministic functions $g_j^* : \mathcal{X}^{\mathrm{o}} \to \mathcal{X}_j^{\mathrm{w}}$ for any $j \in [F^{\mathrm{w}}]$, and $(g_1^*, \ldots, g_{F^{\mathrm{w}}}^*) \in \mathcal{G}$. In this case, for any $\boldsymbol{r}$ and $\overline{S}$, it holds that $\boldsymbol{g}^* \in \mathcal{G}(\boldsymbol{r}, \overline{S})$. Hence, the following holds:

$$
R_{\mathrm{MSE},j}(g_{\mathcal{G}(\boldsymbol{r},\overline{S}),j}) = 0, \quad \forall j \in [F^{\mathrm{w}}].
\tag{C.67}
$$

For any $j \in [F^{\mathrm{w}}]$, define $\mathcal{G}_j(r_j, \overline{S}_j)$ as the set of hypotheses that satisfy the following two conditions: (i) each element is a solution obtained by methods that are guaranteed to achieve consistency (Mohri et al., 2018; Cheng et al., 2023), and (ii) its empirical risk is at most $r_j$. As $n$ increases and $r_j \to 0$, the assumptions on $\overline{R}_{l_j}$ and the theoretical guarantees of these learning methods for $g_j$ imply the following:

$$
R_{\mathrm{MSE},j}(g_j) \to 0, \quad \forall g_j \in \mathcal{G}_j(r_j, \overline{S}_j), \quad \forall j \in [F^{\mathrm{w}}].
\tag{C.68}
$$

Additionally, by assumption, there exists a true deterministic function $f^* : \mathcal{X} \to \mathcal{Y}$ for label prediction, and $f^* \in \mathcal{F}$. Therefore, the following holds:

$$R_l(f_{\mathcal{F}}) = 0. \tag{C.69}$$

Thus, under the conditions of Theorem 4.4, the following holds for $f_{\boldsymbol{g},\overline{S}}$ obtained through LAC-dWFL's step (ii):

$$n \to \infty, \ \ R_{l,\boldsymbol{g}}(f_{\boldsymbol{g},\overline{S}}) \to 0. \tag{C.70}$$

Furthermore, using Theorem 3.1 with $C_\sigma \leq 0$ and $\sigma^2 = (2\pi)^{-1}$, and additionally letting $r_j \to 0$ as $n$ increases for any $j \in [F^{\mathrm{w}}]$, the following holds:

$$n \to \infty, \ \ R_l(f_{\boldsymbol{g},\overline{S}}) \to 0, \ \ \forall \boldsymbol{g} \in \mathcal{G}(\boldsymbol{r}, \overline{S}). \tag{C.71}$$

Since $l$ is $L_l$-Lipschitz continuous and $f_{\boldsymbol{g},\overline{S}}$ is $L_f$-Lipschitz continuous, the following holds (Shalev-Shwartz & Ben-David, 2014):

$$\mathfrak{R}_n^*(\tilde{\mathcal{G}}_{l,f}(\boldsymbol{r}, \boldsymbol{S})) \leq L_l L_f \mathfrak{R}_n^*(\mathcal{G}(\boldsymbol{r}, \boldsymbol{S})). \tag{C.72}$$

Consequently, if $\mathfrak{R}_n^*(\mathcal{G}(\boldsymbol{r}, \boldsymbol{S}))$ and $\mathfrak{R}_n^*(\mathcal{G}_j(r_j, \overline{S}_j))$ for any $j \in [F^{\mathrm{w}}]$ are monotonically decreasing with respect to $n$ and converge to 0, the error bound established in Theorem 4.5 converges to 0 as $n \to \infty$.

$\square$

# D. Proofs for Appendix A

## D.1. Proof of Theorem A.1

*Proof of Theorem A.1.* For any $f \in \mathcal{F}$, $q$ and $l$ bounded by $U_l$, $R_l(f) - R_{l,q}(f)$ can be upper-bounded as follows:

$$
\begin{aligned}
&R_l(f) - R_{l,q}(f) \\
&= \mathbb{E}_{p_*(\boldsymbol{x},y)}[l(f(\boldsymbol{X}), Y)] - \mathbb{E}_{p_*(\boldsymbol{x}^\circ,y)q(\boldsymbol{x}^{\mathrm{w}}|\boldsymbol{x}^\circ)}[l(f(\boldsymbol{X}), Y)] \\
&\leq \mathbb{E}_{p_*(\boldsymbol{x}^\circ,y)}\left[\int_{\mathcal{X}^{\mathrm{w}}} l(f(\boldsymbol{x}^{\mathrm{w}}, \boldsymbol{X}^\circ), Y)(p_*(\boldsymbol{x}^{\mathrm{w}}|\boldsymbol{X}^\circ, Y) - q(\boldsymbol{x}^{\mathrm{w}}|\boldsymbol{X}^\circ))\mathrm{d}\boldsymbol{x}^{\mathrm{w}}\right] \\
&= \mathbb{E}_{p_*(\boldsymbol{x}^\circ,y)}\left[\int_{\mathcal{X}^{\mathrm{w}}} l(f(\boldsymbol{x}^{\mathrm{w}}, \boldsymbol{X}^\circ), Y)\left(\sqrt{p_*(\boldsymbol{x}^{\mathrm{w}}|\boldsymbol{X}^\circ, Y)} + \sqrt{q(\boldsymbol{x}^{\mathrm{w}}|\boldsymbol{X}^\circ)}\right) \right. \\
&\qquad\qquad\qquad \left. \times \left(\sqrt{p_*(\boldsymbol{x}^{\mathrm{w}}|\boldsymbol{X}^\circ, Y)} - \sqrt{q(\boldsymbol{x}^{\mathrm{w}}|\boldsymbol{X}^\circ)}\right)\mathrm{d}\boldsymbol{x}^{\mathrm{w}}\right] \\
&\leq \left\{\mathbb{E}_{p_*(\boldsymbol{x},y)}[(l(f(\boldsymbol{X}), Y))^2] \times \mathbb{E}_{p_*(\boldsymbol{x}^\circ,y)}\left[\int_{\mathcal{X}^{\mathrm{w}}}\left(\sqrt{p_*(\boldsymbol{x}^{\mathrm{w}}|\boldsymbol{X}^\circ, Y)} - \sqrt{q(\boldsymbol{x}^{\mathrm{w}}|\boldsymbol{X}^\circ)}\right)^2\mathrm{d}\boldsymbol{x}^{\mathrm{w}}\right]\right\}^{\frac{1}{2}} \\
&\quad + \left\{\mathbb{E}_{p_*(\boldsymbol{x}^\circ,y)q(\boldsymbol{x}^{\mathrm{w}}|\boldsymbol{x}^\circ)}[(l(f(\boldsymbol{X}), Y))^2] \times \mathbb{E}_{p_*(\boldsymbol{x}^\circ,y)}\left[\int_{\mathcal{X}^{\mathrm{w}}}\left(\sqrt{p_*(\boldsymbol{x}^{\mathrm{w}}|\boldsymbol{X}^\circ, Y)} - \sqrt{q(\boldsymbol{x}^{\mathrm{w}}|\boldsymbol{X}^\circ)}\right)^2\mathrm{d}\boldsymbol{x}^{\mathrm{w}}\right]\right\}^{\frac{1}{2}} \\
&\leq \left(\sqrt{R_l(f)} + \sqrt{R_{l,q}(f)}\right)\left\{2U_l\mathbb{E}_{p_*(\boldsymbol{x}^\circ,y)}[D_{\mathrm{H}}^2(p_*(\boldsymbol{x}^{\mathrm{w}}|\boldsymbol{X}^\circ, Y), q(\boldsymbol{x}^{\mathrm{w}}|\boldsymbol{X}^\circ))]\right\}^{\frac{1}{2}} \\
&\leq 2U_l(\mathbb{E}_{p_*(\boldsymbol{x}^\circ,y)}[D_{\mathrm{H}}^2(p_*(\boldsymbol{x}^{\mathrm{w}}|\boldsymbol{X}^\circ, Y), q(\boldsymbol{x}^{\mathrm{w}}|\boldsymbol{X}^\circ))])^{\frac{1}{2}}. \tag{D.73}
\end{aligned}
$$

Since $l$, $p_*$, $f$ and $q$ are all measurable functions, the second inequality uses the Cauchy-Schwarz inequality. The third inequality uses the fact that $l$ is upper bounded by $U_l$. Hence, Theorem A.1 is showed.

$\square$

## D.2. Proof of Theorem A.3

Using Lemma 4.1, we establish the following lemma:

**Lemma D.1.** *For any measurable $f \in \mathcal{F}$, $q$ and $l$ bounded by $U_l$, the following inequality holds:*

$$|R_l(f) - R_{l,q}(f)| \leq \left\{ 2\sqrt{R_l(f)} + \left( 2U_l \mathbb{E}_{p_*(\boldsymbol{x}^\circ, y)} \left[ D_H^2(p_*(\boldsymbol{X}^w | \boldsymbol{X}^\circ, Y), q(\boldsymbol{X}^w | \boldsymbol{X}^\circ)) \right] \right)^{\frac{1}{2}} \right\}$$

$$\times \left\{ 2U_l \mathbb{E}_{p_*(\boldsymbol{x}^\circ, y)} \left[ D_H^2(p_*(\boldsymbol{X}^w | \boldsymbol{X}^\circ, Y), q(\boldsymbol{X}^w | \boldsymbol{X}^\circ)) \right] \right\}^{\frac{1}{2}}. \tag{D.74}$$

*Proof of Lemma D.1.* Using Lemma 4.1, for any measurable $f \in \mathcal{F}$, $q$ and $l$ bounded by $U_l < \infty$, the following inequality holds:

$$\left| \sqrt{R_l(f)} - \sqrt{R_{l,q}(f)} \right| \leq \left\{ 2U_l \mathbb{E}_{p_*(\boldsymbol{x}^\circ, y)} \left[ D_H^2(p_*(\boldsymbol{X}^w | \boldsymbol{X}^\circ, Y), q(\boldsymbol{X}^w | \boldsymbol{X}^\circ)) \right] \right\}^{\frac{1}{2}}.$$

From this inequality, $\sqrt{R_{l,g}(f)}$ can be upper-bounded as follows:

$$\sqrt{R_{l,q}(f)} \leq \sqrt{R_l(f)} + \left| \sqrt{R_l(f)} - \sqrt{R_{l,q}(f)} \right|$$

$$\leq \sqrt{R_l(f)} + \left\{ 2U_l \mathbb{E}_{p_*(\boldsymbol{x}^\circ, y)} \left[ D_H^2(p_*(\boldsymbol{X}^w | \boldsymbol{X}^\circ, Y), q(\boldsymbol{X}^w | \boldsymbol{X}^\circ)) \right] \right\}^{\frac{1}{2}}. \tag{D.75}$$

Applying the above result to Eq. (4.9) in Lemma 4.1 establishes Lemma (D.1):

$$|R_l(f) - R_{l,q}(f)| \leq \left\{ 2\sqrt{R_l(f)} + \left( 2U_l \mathbb{E}_{p_*(\boldsymbol{x}^\circ, y)} \left[ \{ D_H^2(p_*(\boldsymbol{X}^w | \boldsymbol{X}^\circ, Y), q(\boldsymbol{X}^w | \boldsymbol{X}^\circ)) \right] \right)^{\frac{1}{2}} \right\} \tag{D.76}$$

$$\times \left\{ 2U_l \mathbb{E}_{p_*(\boldsymbol{x}^\circ, y)} \left[ D_H^2(p_*(\boldsymbol{X}^w | \boldsymbol{X}^\circ, Y), q(\boldsymbol{X}^w | \boldsymbol{X}^\circ)) \right] \right\}^{\frac{1}{2}}. \tag{D.77}$$

$\square$

Using Lemma D.1, we now prove Theorem A.3 as follows.

*Proofs of Theorem A.3.* The empirical risk minimizer in the ordinary supervised learning setting is defined as follows:

$$f_S := \arg\min_{f \in \mathcal{F}} \widehat{R}_l(f).$$

The LHS of Eq. (A.20) can be decomposed as follows:

$$R_{l,q}(f_{q,\overline{S}}) - R_l(f_{\mathcal{F}}) = \underbrace{R_{l,q}(f_{q,\overline{S}}) - \widehat{R}_{l,q}(f_{q,\overline{S}})}_{(a1)} + \underbrace{\widehat{R}_{l,q}(f_{q,\overline{S}}) - R_{l,q}(f_S)}_{(a2)}$$

$$+ \underbrace{R_{l,q}(f_S) - R_l(f_S)}_{(a3)} + \underbrace{R_l(f_S) - R_l(f_{\mathcal{F}})}_{(a4)}. \tag{D.78}$$

The terms (a1) and (a2) in Eq. (D.78) can be upper-bounded as follows:

$$(a1) \leq \max_{f \in \mathcal{F}} |R_{l,q}(f) - \widehat{R}_{l,q}(f)|,$$

$$(a2) \leq \widehat{R}_{l,q}(f_S) - R_{l,q}(f_S) \leq \max_{f \in \mathcal{F}} |R_{l,q}(f) - \widehat{R}_{l,q}(f)|.$$

The term (a3) in Eq. (D.78) can be upper-bounded using Lemma D.1 as follows:

$$(a3) \leq \left\{ 2\sqrt{R_l(f_S)} + \left( 2U_l \mathbb{E}_{p_*(\boldsymbol{x}^\circ, y)} \left[ D_H^2(p_*(\boldsymbol{X}^w | \boldsymbol{X}^\circ, Y), q(\boldsymbol{X}^w | \boldsymbol{X}^\circ)) \right] \right)^{\frac{1}{2}} \right\}$$

$$\times \left\{ 2U_l \mathbb{E}_{p_*(\boldsymbol{x}^\circ, y)} \left[ D_H^2(p_*(\boldsymbol{X}^w | \boldsymbol{X}^\circ, Y), q(\boldsymbol{X}^w | \boldsymbol{X}^\circ)) \right] \right\}^{\frac{1}{2}}. \tag{D.79}$$

Here, $R_l(f_S)$ can be upper-bounded as follows:

$$\begin{aligned}
R_l(f_S) &= R_l(f_S) - \widehat{R}_l(f_S) + \widehat{R}_l(f_S) - R_l(f_{\mathcal{F}}) + R_l(f_{\mathcal{F}}) \\
&\leq R_l(f_S) - \widehat{R}_l(f_S) + \widehat{R}_l(f_{\mathcal{F}}) - R_l(f_{\mathcal{F}}) + R_l(f_{\mathcal{F}}) \\
&\leq R_l(f_{\mathcal{F}}) + 2 \max_{f \in \mathcal{F}} |R_l(f) - \widehat{R}_l(f)|.
\end{aligned} \tag{D.80}$$

Applying the above inequality to Eq. (D.79), the term (a3) in Eq. (D.78) can be bounded as follows:

$$\begin{aligned}
\text{(a3)} &\leq \left\{ 2 \left( R_l(f_{\mathcal{F}}) + 2 \max_{f \in \mathcal{F}} |R_l(f) - \widehat{R}_l(f)| \right)^{\frac{1}{2}} + \left( 2U_l \mathbb{E}_{p_*(\boldsymbol{x}^{\circ}, y)} \left[ D_{\mathrm{H}}^2(p_*(\boldsymbol{X}^{\mathrm{w}}|\boldsymbol{X}^{\circ}, Y), q(\boldsymbol{X}^{\mathrm{w}}|\boldsymbol{X}^{\circ})) \right] \right)^{\frac{1}{2}} \right\} \\
&\quad \times \left\{ 2U_l \mathbb{E}_{p_*(\boldsymbol{x}^{\circ}, y)} \left[ D_{\mathrm{H}}^2(p_*(\boldsymbol{X}^{\mathrm{w}}|\boldsymbol{X}^{\circ}, Y), q(\boldsymbol{X}^{\mathrm{w}}|\boldsymbol{X}^{\circ})) \right] \right\}^{\frac{1}{2}}.
\end{aligned} \tag{D.81}$$

Similarly, the term (a4) in Eq. (D.78) can be upper-bounded as follows:

$$R_l(f_S) - R_l(f_{\mathcal{F}}) \leq 2 \max_{f \in \mathcal{F}} |R_l(f) - \widehat{R}_l(f)|.$$

Combining the above results, the LHS of Eq. (D.78) can be upper-bounded as follows:

$$\begin{aligned}
&R_{l,q}(f_{q,\overline{S}}) - R_l(f_{\mathcal{F}}) \\
&\leq 2 \max_{f \in \mathcal{F}} |R_{l,q}(f) - \widehat{R}_{l,q}(f)| + 2 \max_{f \in \mathcal{F}} |R_l(f) - \widehat{R}_l(f)| \\
&\quad + \left\{ 2 \left( R_l(f_{\mathcal{F}}) + 2 \max_{f \in \mathcal{F}} |R_l(f) - \widehat{R}_l(f)| \right)^{\frac{1}{2}} + \left( 2U_l \mathbb{E}_{p_*(\boldsymbol{x}^{\circ}, y)} \left[ D_{\mathrm{H}}^2(p_*(\boldsymbol{X}^{\mathrm{w}}|\boldsymbol{X}^{\circ}, Y), q(\boldsymbol{X}^{\mathrm{w}}|\boldsymbol{X}^{\circ})) \right] \right)^{\frac{1}{2}} \right\} \\
&\quad \times \left\{ 2U_l \mathbb{E}_{p_*(\boldsymbol{x}^{\circ}, y)} \left[ D_{\mathrm{H}}^2(p_*(\boldsymbol{X}^{\mathrm{w}}|\boldsymbol{X}^{\circ}, Y), q(\boldsymbol{X}^{\mathrm{w}}|\boldsymbol{X}^{\circ})) \right] \right\}^{\frac{1}{2}}.
\end{aligned} \tag{D.82}$$

By the uniform law of large numbers (Mohri et al., 2018), for any $\delta \in (0, 1)$, the following holds with probability at least $1 - \delta/2$:

$$\begin{aligned}
\max_{f \in \mathcal{F}} |R_{l,q}(f) - \widehat{R}_{l,q}(f)| &\leq 2\mathfrak{R}_n^q(\widetilde{\mathcal{F}}_l) + U_l \sqrt{\frac{\log(4/\delta)}{2n}}, \\
\max_{f \in \mathcal{F}} |R_l(f) - \widehat{R}_l(f)| &\leq 2\mathfrak{R}_n^*(\widetilde{\mathcal{F}}_l) + U_l \sqrt{\frac{\log(4/\delta)}{2n}}.
\end{aligned}$$

Here, $\widetilde{\mathcal{F}}_l := \{(\boldsymbol{x}, y) \mapsto l(f(\boldsymbol{x}), y) : f \in \mathcal{F}\}$. From the assumption of $l$ and Lemma 26.9 in (Shalev-Shwartz & Ben-David, 2014), it follows that $\mathfrak{R}_n^*(\widetilde{\mathcal{F}}_l) \leq L_l \mathfrak{R}_n^*(\mathcal{F})$ and $\mathfrak{R}_n^g(\widetilde{\mathcal{F}}_l) \leq L_l \mathfrak{R}_n^g(\mathcal{F})$. Thus, for any $\delta \in (0, 1)$, the following inequality holds with probability at least $1 - \delta$:

$$\begin{aligned}
&R_{l,q}(f_{q,\overline{S}}) - R_l(f_{\mathcal{F}}) \\
&\leq 4 \left( L_l \mathfrak{R}_n^*(\mathcal{F}) + L_l \mathfrak{R}_n^q(\mathcal{F}) + U_l \sqrt{\frac{\log(4/\delta)}{2n}} \right) \\
&\quad + \left\{ 2 \left( R_l(f_{\mathcal{F}}) + 4L_l \mathfrak{R}_n^*(\mathcal{F}) + 2U_l \sqrt{\frac{\log(4/\delta)}{2n}} \right)^{\frac{1}{2}} + \left( 2U_l \mathbb{E}_{p_*(\boldsymbol{x}^{\circ}, y)} \left[ D_{\mathrm{H}}^2(p_*(\boldsymbol{X}^{\mathrm{w}}|\boldsymbol{X}^{\circ}, Y), q(\boldsymbol{X}^{\mathrm{w}}|\boldsymbol{X}^{\circ})) \right] \right)^{\frac{1}{2}} \right\} \\
&\quad \times \left( 2U_l \mathbb{E}_{p_*(\boldsymbol{x}^{\circ}, y)} \left[ D_{\mathrm{H}}^2(p_*(\boldsymbol{X}^{\mathrm{w}}|\boldsymbol{X}^{\circ}, Y), q(\boldsymbol{X}^{\mathrm{w}}|\boldsymbol{X}^{\circ})) \right] \right)^{\frac{1}{2}}.
\end{aligned} \tag{D.83}$$

This concludes the proof of Theorem A.3.

$\square$

# E. Proofs for Appendix B

## E.1. Proof of Theorem B.1

*Proof of Theorem B.1.* The LHS of Eq. (B.27) can be upper-bounded as follows:

$$
\begin{aligned}
&R_l(f) - R_{l,\boldsymbol{g}}(f) \\
&= \mathbb{E}_{p_*(\boldsymbol{x},y)}[l(f(\boldsymbol{X}),Y)] - \mathbb{E}_{p_*(y,\boldsymbol{m})q_{\boldsymbol{g}}(\boldsymbol{x}|y,\boldsymbol{m})}[l(f(\boldsymbol{X}),Y)] \\
&= \mathbb{E}_{p_*(\boldsymbol{m})p_*(\boldsymbol{x},y|\boldsymbol{m})}[l(f(\boldsymbol{X}),Y)] - \mathbb{E}_{p_*(y,\boldsymbol{m})q_{\boldsymbol{g}}(\boldsymbol{x}_{\mathrm{mis}(\boldsymbol{m})}|\boldsymbol{x}_{\mathrm{obs}(\boldsymbol{m})},\boldsymbol{m})p_*(\boldsymbol{x}_{\mathrm{obs}(\boldsymbol{m})}|\boldsymbol{m},y)}[l(f(\boldsymbol{X}),Y)] \\
&= \mathbb{E}_{p_*(\boldsymbol{m})}\big[\mathbb{E}_{p_*(\boldsymbol{x},y|\boldsymbol{M})}[l(f(\boldsymbol{X}),Y)] - \mathbb{E}_{p_*(\boldsymbol{x}_{\mathrm{obs}(\boldsymbol{M})},y|\boldsymbol{M})q_{\boldsymbol{g}}(\boldsymbol{x}_{\mathrm{mis}(\boldsymbol{M})}|\boldsymbol{x}_{\mathrm{obs}(\boldsymbol{M})},\boldsymbol{M})}[l(f(\boldsymbol{X}),Y)]\big] \\
&\leq \mathbb{E}_{p_*(\boldsymbol{m})}\bigg[\mathbb{E}_{p_*(\boldsymbol{x}_{\mathrm{obs}(\boldsymbol{M})},y|\boldsymbol{M})}\bigg[\int_{\mathcal{X}_{\mathrm{mis}(\boldsymbol{M})}} l(f(\boldsymbol{X}_{\mathrm{obs}(\boldsymbol{M})},\boldsymbol{x}_{\mathrm{obs}(\boldsymbol{M})}),Y) \\
&\qquad\qquad\qquad\qquad \times \Big|p_*(\boldsymbol{x}_{\mathrm{mis}(\boldsymbol{M})}|\boldsymbol{X}_{\mathrm{obs}(\boldsymbol{M})},Y,\boldsymbol{M}) - q_{\boldsymbol{g}}(\boldsymbol{x}_{\mathrm{mis}(\boldsymbol{M})}|\boldsymbol{X}_{\mathrm{obs}(\boldsymbol{M})},\boldsymbol{M})\Big|\mathrm{d}\boldsymbol{x}_{\mathrm{mis}(\boldsymbol{M})}\bigg]\bigg] \\
&\leq U_l\mathbb{E}_{p_*(\boldsymbol{m})}\bigg[\mathbb{E}_{p_*(\boldsymbol{x}_{\mathrm{obs}(\boldsymbol{M})},y|\boldsymbol{M})}\bigg[\int_{\mathcal{X}_{\mathrm{mis}(\boldsymbol{M})}} \\
&\qquad\qquad\qquad \Big|p_*(\boldsymbol{x}_{\mathrm{mis}(\boldsymbol{M})}|\boldsymbol{X}_{\mathrm{obs}(\boldsymbol{M})},Y,\boldsymbol{M}) - q_{\boldsymbol{g}}(\boldsymbol{x}_{\mathrm{mis}(\boldsymbol{M})}|\boldsymbol{X}_{\mathrm{obs}(\boldsymbol{M})},\boldsymbol{M})\Big|\mathrm{d}\boldsymbol{x}_{\mathrm{mis}(\boldsymbol{M})}\bigg]\bigg] \\
&\leq U_l\mathbb{E}_{p_*(\boldsymbol{m})}\bigg[\mathbb{E}_{p_*(\boldsymbol{x}_{\mathrm{obs}(\boldsymbol{M})},y|\boldsymbol{M})}\bigg[\frac{1}{2}\bigg\{\int_{\mathcal{X}_{\mathrm{mis}(\boldsymbol{M})}} p_*(\boldsymbol{x}_{\mathrm{mis}(\boldsymbol{M})}|\boldsymbol{X}_{\mathrm{obs}(\boldsymbol{M})},Y,\boldsymbol{M}) \\
&\qquad\qquad\qquad\qquad \times \log\frac{p_*(\boldsymbol{x}_{\mathrm{mis}(\boldsymbol{M})}|\boldsymbol{X}_{\mathrm{obs}(\boldsymbol{M})},Y,\boldsymbol{M})}{q_{\boldsymbol{g}}(\boldsymbol{x}_{\mathrm{mis}(\boldsymbol{M})}|\boldsymbol{X}_{\mathrm{obs}(\boldsymbol{M})},\boldsymbol{M})}\mathrm{d}\boldsymbol{x}_{\mathrm{mis}(\boldsymbol{M})}\bigg\}^{\frac{1}{2}}\bigg]\bigg] \\
&\leq U_l\bigg\{\frac{1}{2}\mathbb{E}_{p_*(\boldsymbol{x},y,\boldsymbol{m})}\bigg[\log\frac{p_*(\boldsymbol{X}_{\mathrm{mis}(\boldsymbol{M})}|\boldsymbol{X}_{\mathrm{obs}(\boldsymbol{M})},Y,\boldsymbol{M})}{q_{\boldsymbol{g}}(\boldsymbol{X}_{\mathrm{mis}(\boldsymbol{M})}|\boldsymbol{X}_{\mathrm{obs}(\boldsymbol{M})},\boldsymbol{M})}\bigg]\bigg\}^{\frac{1}{2}} \\
&= U_l\bigg\{\frac{1}{2}\bigg(\mathrm{LL}'_{p_*} + \mathbb{E}_{p_*(\boldsymbol{x},\boldsymbol{m})}\Big[-\log q_{\boldsymbol{g}}(\boldsymbol{X}_{\mathrm{mis}(\boldsymbol{M})}|\boldsymbol{X}_{\mathrm{obs}(\boldsymbol{M})},\boldsymbol{M})\Big]\bigg)\bigg\}^{\frac{1}{2}}.
\end{aligned}
\tag{E.84}
$$

Here, $\mathrm{LL}'_{p_*} = \mathbb{E}_{p_*(\boldsymbol{x},y,\boldsymbol{m})}[p_*(\boldsymbol{X}_{\mathrm{mis}(\boldsymbol{M})}|\boldsymbol{X}_{\mathrm{obs}(\boldsymbol{M})},Y,\boldsymbol{M})]$. The second inequality uses $l(y,y') \leq U_l, \forall y,y' \in \mathcal{Y}$. The third inequality uses Pinsker's inequality, and the fourth inequality uses Jensen's inequality. Furthermore, the following holds:

$$
\begin{aligned}
&-\mathbb{E}_{p_*(\boldsymbol{x},y,\boldsymbol{m})}[\log q_{\boldsymbol{g}}(\boldsymbol{X}_{\mathrm{mis}(\boldsymbol{M})}|\boldsymbol{X}_{\mathrm{obs}(\boldsymbol{M})},\boldsymbol{M})] \\
&= \frac{1}{2}\log(2\pi)^{|\mathrm{mis}(\boldsymbol{M})|}\sigma^{2|\mathrm{mis}(\boldsymbol{M})|} \\
&\quad + \mathbb{E}_{p_*(\boldsymbol{x},y,\boldsymbol{m})}\bigg[\frac{1}{2\sigma^2}(\boldsymbol{X}_{\mathrm{mis}(\boldsymbol{M})} - \boldsymbol{g}_{\mathrm{mis}(\boldsymbol{M})}(\boldsymbol{X}_{\mathrm{obs}(\boldsymbol{M})}))^\top(\boldsymbol{X}_{\mathrm{mis}(\boldsymbol{M})} - \boldsymbol{g}_{\mathrm{mis}(\boldsymbol{M})}(\boldsymbol{X}_{\mathrm{obs}(\boldsymbol{M})}))\bigg] \\
&= \mathbb{E}_{p_*(\boldsymbol{m})}\bigg[\frac{1}{2}\log(2\pi)^{|\mathrm{mis}(\boldsymbol{M})|}\sigma^{2|\mathrm{mis}(\boldsymbol{M})|}\bigg] + \mathbb{E}_{p_*(\boldsymbol{m})}\bigg[\frac{1}{2\sigma^2}\sum_{j\in\mathrm{mis}(\boldsymbol{M})}\mathbb{E}_{p_*(\boldsymbol{x}|\boldsymbol{M})}[(X_j - g_j(\widetilde{X}))^2]\bigg] \\
&= \mathbb{E}_{p_*(\boldsymbol{m})}\bigg[\frac{1}{2}\log(2\pi)^{|\mathrm{mis}(\boldsymbol{M})|}\sigma^{2|\mathrm{mis}(\boldsymbol{M})|}\bigg] + \mathbb{E}_{p_*(\boldsymbol{m})}\bigg[\frac{1}{2\sigma^2}\sum_{j\in\mathrm{mis}(\boldsymbol{M})}R^{(\boldsymbol{M})}_{\mathrm{MSE},j}(g_j)\bigg].
\end{aligned}
\tag{E.85}
$$

Substituting this result into Eq. (E.84) establishes Eq. (B.27) in Theorem B.1:

$$
R_l(f) - R_{l,\boldsymbol{g}}(f) \leq U_l\bigg\{\frac{1}{2}\bigg(C'_\sigma + \frac{1}{2\sigma^2}\mathbb{E}_{p_*(\boldsymbol{m})}\bigg[\sum_{j\in\mathrm{mis}(\boldsymbol{M})}R^{(\boldsymbol{M})}_{\mathrm{MSE},j}(g_j)\bigg]\bigg)\bigg\}^{\frac{1}{2}}.
\tag{E.86}
$$

Here, $C'_\sigma = \mathbb{E}_{p_*(\boldsymbol{x},y,\boldsymbol{m})}[p_*(\boldsymbol{X}_{\mathrm{mis}(\boldsymbol{M})}|\boldsymbol{X}_{\mathrm{obs}(\boldsymbol{M})},Y,\boldsymbol{M})] + \mathbb{E}_{p_*(\boldsymbol{m})}[\frac{1}{2}\log(2\pi)^{|\mathrm{mis}(\boldsymbol{M})|}\sigma^{2|\mathrm{mis}(\boldsymbol{M})|}]$ □

### E.2. Proof of Lemma B.3

*Proof of Lemma B.3.* The LHS of Eq. (B.28) can be bounded from above as follows:

$$
\begin{aligned}
&|R_l(f) - R_{l,\boldsymbol{g}}(f)| \\
&= \left| \mathbb{E}_{p_*(\boldsymbol{x},y)}[l(f(\boldsymbol{X}),Y)] - \mathbb{E}_{p_*(y,\boldsymbol{m})q_{\boldsymbol{g}}(\boldsymbol{x}|y,\boldsymbol{m})}[l(f(\boldsymbol{X}),Y)] \right| \\
&= \left| \mathbb{E}_{p_*(\boldsymbol{m})p_*(\boldsymbol{x},y|\boldsymbol{m})}[l(f(\boldsymbol{X}),Y)] - \mathbb{E}_{p_*(y,\boldsymbol{m})q_{\boldsymbol{g}}(\boldsymbol{x}_{\mathrm{mis}(\boldsymbol{m})}|\boldsymbol{x}_{\mathrm{obs}(\boldsymbol{m})},\boldsymbol{m})p_*(\boldsymbol{x}_{\mathrm{obs}(\boldsymbol{m})}|\boldsymbol{m},y)}[l(f(\boldsymbol{X}),Y)] \right| \\
&= \left| \mathbb{E}_{p_*(\boldsymbol{m})}\left[ \mathbb{E}_{p_*(\boldsymbol{x},y|\boldsymbol{M})}[l(f(\boldsymbol{X}),Y)] - \mathbb{E}_{p_*(\boldsymbol{x}_{\mathrm{obs}(\boldsymbol{M})},y|\boldsymbol{M})q_{\boldsymbol{g}}(\boldsymbol{x}_{\mathrm{mis}(\boldsymbol{M})}|\boldsymbol{x}_{\mathrm{obs}(\boldsymbol{M})},\boldsymbol{M})}[l(f(\boldsymbol{X}),Y)] \right] \right| \\
&\leq \mathbb{E}_{p_*(\boldsymbol{m})}\Bigg[ \mathbb{E}_{p_*(\boldsymbol{x}_{\mathrm{obs}(\boldsymbol{M})},y|\boldsymbol{M})}\Bigg[ \int_{\mathcal{X}_{\mathrm{mis}(\boldsymbol{M})}} l(f(\boldsymbol{X}_{\mathrm{obs}(\boldsymbol{M})},\boldsymbol{x}_{\mathrm{obs}(\boldsymbol{M})}),Y) \\
&\qquad\qquad \times \left| p_*(\boldsymbol{x}_{\mathrm{mis}(\boldsymbol{M})}|\boldsymbol{X}_{\mathrm{obs}(\boldsymbol{M})},Y,\boldsymbol{M}) - q_{\boldsymbol{g}}(\boldsymbol{x}_{\mathrm{mis}(\boldsymbol{M})}|\boldsymbol{X}_{\mathrm{obs}(\boldsymbol{M})},\boldsymbol{M}) \right| \mathrm{d}\boldsymbol{x}_{\mathrm{mis}(\boldsymbol{M})} \Bigg] \Bigg] \\
&= \mathbb{E}_{p_*(\boldsymbol{m})}\Bigg[ \mathbb{E}_{p_*(\boldsymbol{x}_{\mathrm{obs}(\boldsymbol{M})},y|\boldsymbol{M})}\Bigg[ \int_{\mathcal{X}_{\mathrm{mis}(\boldsymbol{M})}} l(f(\boldsymbol{X}_{\mathrm{obs}(\boldsymbol{M})},\boldsymbol{x}_{\mathrm{obs}(\boldsymbol{M})}),Y) \\
&\qquad\qquad \times \left( \sqrt{p_*(\boldsymbol{x}_{\mathrm{mis}(\boldsymbol{M})}|\boldsymbol{X}_{\mathrm{obs}(\boldsymbol{M})},Y,\boldsymbol{M})} + \sqrt{q_{\boldsymbol{g}}(\boldsymbol{x}_{\mathrm{mis}(\boldsymbol{M})}|\boldsymbol{X}_{\mathrm{obs}(\boldsymbol{M})},\boldsymbol{M})} \right) \\
&\qquad\qquad \times \left| \sqrt{p_*(\boldsymbol{x}_{\mathrm{mis}(\boldsymbol{M})}|\boldsymbol{X}_{\mathrm{obs}(\boldsymbol{M})},Y,\boldsymbol{M})} - \sqrt{q_{\boldsymbol{g}}(\boldsymbol{x}_{\mathrm{mis}(\boldsymbol{M})}|\boldsymbol{X}_{\mathrm{obs}(\boldsymbol{M})},\boldsymbol{M})} \right| \mathrm{d}\boldsymbol{x}_{\mathrm{mis}(\boldsymbol{M})} \Bigg] \Bigg] \\
&= \mathbb{E}_{p_*(\boldsymbol{m})}\Bigg[ \mathbb{E}_{p_*(\boldsymbol{x}_{\mathrm{obs}(\boldsymbol{M})},y|\boldsymbol{M})}\Bigg[ \int_{\mathcal{X}_{\mathrm{mis}(\boldsymbol{M})}} l(f(\boldsymbol{X}_{\mathrm{obs}(\boldsymbol{M})},\boldsymbol{x}_{\mathrm{obs}(\boldsymbol{M})}),Y)\sqrt{p_*(\boldsymbol{x}_{\mathrm{mis}(\boldsymbol{M})}|\boldsymbol{X}_{\mathrm{obs}(\boldsymbol{M})},Y,\boldsymbol{M})} \\
&\qquad\qquad \times \left| \sqrt{p_*(\boldsymbol{x}_{\mathrm{mis}(\boldsymbol{M})}|\boldsymbol{X}_{\mathrm{obs}(\boldsymbol{M})},Y,\boldsymbol{M})} - \sqrt{q_{\boldsymbol{g}}(\boldsymbol{x}_{\mathrm{mis}(\boldsymbol{M})}|\boldsymbol{X}_{\mathrm{obs}(\boldsymbol{M})},\boldsymbol{M})} \right| \mathrm{d}\boldsymbol{x}_{\mathrm{mis}(\boldsymbol{M})} \Bigg] \Bigg] \\
&\quad + \mathbb{E}_{p_*(\boldsymbol{m})}\Bigg[ \mathbb{E}_{p_*(\boldsymbol{x}_{\mathrm{obs}(\boldsymbol{M})},y|\boldsymbol{M})}\Bigg[ \int_{\mathcal{X}_{\mathrm{mis}(\boldsymbol{M})}} l(f(\boldsymbol{X}_{\mathrm{obs}(\boldsymbol{M})},\boldsymbol{x}_{\mathrm{obs}(\boldsymbol{M})}),Y)\sqrt{q_{\boldsymbol{g}}(\boldsymbol{x}_{\mathrm{mis}(\boldsymbol{M})}|\boldsymbol{X}_{\mathrm{obs}(\boldsymbol{M})},\boldsymbol{M})} \\
&\qquad\qquad \times \left| \sqrt{p_*(\boldsymbol{x}_{\mathrm{mis}(\boldsymbol{M})}|\boldsymbol{X}_{\mathrm{obs}(\boldsymbol{M})},Y,\boldsymbol{M})} - \sqrt{q_{\boldsymbol{g}}(\boldsymbol{x}_{\mathrm{mis}(\boldsymbol{M})}|\boldsymbol{X}_{\mathrm{obs}(\boldsymbol{M})},\boldsymbol{M})} \right| \mathrm{d}\boldsymbol{x}_{\mathrm{mis}(\boldsymbol{M})} \Bigg] \Bigg] \\
&\leq \Bigg\{ \underbrace{\mathbb{E}_{p_*(\boldsymbol{m})p_*(\boldsymbol{x},y|\boldsymbol{m})}[\{l(f(\boldsymbol{X}),Y)\}^2]}_{(a1)} \\
&\qquad \times \underbrace{\mathbb{E}_{p_*(\boldsymbol{m})p_*(\boldsymbol{x}_{\mathrm{obs}(\boldsymbol{m})},y|\boldsymbol{m})}\left[ D_{\mathrm{H}}^2(p_*(\boldsymbol{X}_{\mathrm{mis}(\boldsymbol{M})}|\boldsymbol{X}_{\mathrm{obs}(\boldsymbol{M})},Y,\boldsymbol{M}),q_{\boldsymbol{g}}(\boldsymbol{X}_{\mathrm{mis}(\boldsymbol{M})}|\boldsymbol{X}_{\mathrm{obs}(\boldsymbol{M})},\boldsymbol{M})) \right]}_{(a2)} \Bigg\}^{\frac{1}{2}} \\
&\quad + \Bigg\{ \underbrace{\mathbb{E}_{p_*(\boldsymbol{m})p_*(\boldsymbol{x}_{\mathrm{obs}(\boldsymbol{m})},y|\boldsymbol{m})q_{\boldsymbol{g}}(\boldsymbol{x}_{\mathrm{mis}(\boldsymbol{m})}|\boldsymbol{x}_{\mathrm{obs}(\boldsymbol{m})},\boldsymbol{m})}[\{l(f(\boldsymbol{X}),Y)\}^2]}_{(a3)} \\
&\qquad \times \underbrace{\mathbb{E}_{p_*(\boldsymbol{m})p_*(\boldsymbol{x}_{\mathrm{obs}(\boldsymbol{m})},y|\boldsymbol{m})}\left[ D_{\mathrm{H}}^2(p_*(\boldsymbol{X}_{\mathrm{mis}(\boldsymbol{M})}|\boldsymbol{X}_{\mathrm{obs}(\boldsymbol{M})},Y,\boldsymbol{M}),q_{\boldsymbol{g}}(\boldsymbol{X}_{\mathrm{mis}(\boldsymbol{M})}|\boldsymbol{X}_{\mathrm{obs}(\boldsymbol{M})},\boldsymbol{M})) \right]}_{(a2)} \Bigg\}^{\frac{1}{2}}
\end{aligned}
$$
(E.87)

Here, the second inequality uses the Cauchy-Schwarz inequality, since $l$, $f$, $p_*$, and $q_{\boldsymbol{g}}$ are all measurable functions.

The terms (a1) and (a3) in Eq. (E.87) can be bounded as follows:

$$
\begin{aligned}
(a1) &= \mathbb{E}_{p_*(\boldsymbol{x},y)}[l(f(\boldsymbol{X}),Y)^2 - 0] \\
&\leq 2U_l \mathbb{E}_{p_*(\boldsymbol{x},y)}[l(f(\boldsymbol{X}),Y)] = 2U_l R_l(f),
\end{aligned}
$$
(E.88)

$$(\text{a3}) = \mathbb{E}_{p_*(\boldsymbol{m})p_*(\boldsymbol{x}_{\text{obs}(\boldsymbol{m})},y|\boldsymbol{m})q_{\boldsymbol{g}}(\boldsymbol{x}_{\text{mis}(\boldsymbol{m})}|\boldsymbol{x}_{\text{obs}(\boldsymbol{m})},\boldsymbol{m})}[\{l(f(\boldsymbol{X}),Y)\}^2 - 0]$$

$$\leq 2U_l\mathbb{E}_{p_*(\boldsymbol{m})p_*(\boldsymbol{x}_{\text{obs}(\boldsymbol{m})},y|\boldsymbol{m})q_{\boldsymbol{g}}(\boldsymbol{x}_{\text{mis}(\boldsymbol{m})}|\boldsymbol{x}_{\text{obs}(\boldsymbol{m})},\boldsymbol{m})}[l(f(\boldsymbol{X}),Y)] = 2U_lR_{l,\boldsymbol{g}}(f). \tag{E.89}$$

In the above, it is used that the function $x \mapsto x^2$ is $2U_l$-Lipschitz continuous when its domain is restricted to $[0, U_l]$. Applying these results to Eq. (E.87), we have demonstrated Lemma B.3:

$$|R_l(f) - R_{l,\boldsymbol{g}}(f)|$$
$$\leq \left(\sqrt{R_l(f)} + \sqrt{R_{l,\boldsymbol{g}}(f)}\right)$$
$$\times \left\{2U_l\mathbb{E}_{p_*(\boldsymbol{m})p_*(\boldsymbol{x}_{\text{obs}(\boldsymbol{m})},y|\boldsymbol{m})}\left[D_{\text{H}}^2(p_*(\boldsymbol{X}_{\text{mis}(\boldsymbol{M})}|\boldsymbol{X}_{\text{obs}(\boldsymbol{M})},Y,\boldsymbol{M}),q_{\boldsymbol{g}}(\boldsymbol{X}_{\text{mis}(\boldsymbol{M})}|\boldsymbol{X}_{\text{obs}(\boldsymbol{M})},\boldsymbol{M}))\right]\right\}^{\frac{1}{2}}.$$

$\square$

### E.3. Proof of Lemma B.4

*Proof of Lemma B.4.* The term $\mathbb{E}_{p_*(\boldsymbol{m})p_*(\boldsymbol{x}_{\text{obs}(\boldsymbol{m})},y|\boldsymbol{m})}\left[D_{\text{H}}^2(p_*(\boldsymbol{X}_{\text{mis}(\boldsymbol{M})}|\boldsymbol{X}_{\text{obs}(\boldsymbol{M})},Y,\boldsymbol{M}),q_{\boldsymbol{g}}(\boldsymbol{X}_{\text{mis}(\boldsymbol{M})}|\boldsymbol{X}_{\text{obs}(\boldsymbol{M})},\boldsymbol{M}))\right]$ in Eq. (B.28) can be bounded as follows:

$$\mathbb{E}_{p_*(\boldsymbol{m})p_*(\boldsymbol{x}_{\text{obs}(\boldsymbol{m})},y|\boldsymbol{m})}\left[D_{\text{H}}^2(p_*(\boldsymbol{X}_{\text{mis}(\boldsymbol{M})}|\boldsymbol{X}_{\text{obs}(\boldsymbol{M})},Y,\boldsymbol{M}),q_{\boldsymbol{g}}(\boldsymbol{X}_{\text{mis}(\boldsymbol{M})}|\boldsymbol{X}_{\text{obs}(\boldsymbol{M})},\boldsymbol{M}))\right]$$

$$\leq \mathbb{E}_{p_*(\boldsymbol{m})p_*(\boldsymbol{x}_{\text{obs}(\boldsymbol{m})},y|\boldsymbol{m})}\Bigg[$$

$$\int_{\mathcal{X}_{\text{mis}(\boldsymbol{M})}} p_*(\boldsymbol{x}_{\text{mis}(\boldsymbol{M})}|\boldsymbol{X}_{\text{obs}(\boldsymbol{M})},Y,\boldsymbol{M})\log\frac{p_*(\boldsymbol{X}_{\text{mis}(\boldsymbol{M})}|\boldsymbol{X}_{\text{obs}(\boldsymbol{M})},Y,\boldsymbol{M})}{q_{\boldsymbol{g}}(\boldsymbol{X}_{\text{mis}(\boldsymbol{M})}|\boldsymbol{X}_{\text{obs}(\boldsymbol{M})},\boldsymbol{M})}d\boldsymbol{x}_{\text{mis}(\boldsymbol{M})}\Bigg]$$

$$= \text{LL}'_{p_*} - \mathbb{E}_{p_*(\boldsymbol{x},y,\boldsymbol{m})}[\log q_{\boldsymbol{g}}(\boldsymbol{X}_{\text{mis}(\boldsymbol{M})}|\boldsymbol{X}_{\text{obs}(\boldsymbol{M})},\boldsymbol{M})]. \tag{E.90}$$

Here, $\text{LL}'_{p_*} = \mathbb{E}_{p_*(\boldsymbol{x},y,\boldsymbol{m})}[p_*(\boldsymbol{X}_{\text{mis}(\boldsymbol{M})}|\boldsymbol{X}_{\text{obs}(\boldsymbol{M})},Y,\boldsymbol{M})]$. Defining $q_{\boldsymbol{g}}$ by the PDF of $\mathcal{N}(\boldsymbol{g}_{\text{mis}(\boldsymbol{m})}(\boldsymbol{x}_{\text{obs}(\boldsymbol{m})}),\sigma^2 I_{|\text{mis}(\boldsymbol{m})|\times|\text{mis}(\boldsymbol{m})|})$ where $\sigma \in \mathbb{R}_+$, The term $-\mathbb{E}_{p_*(\boldsymbol{x},y,\boldsymbol{m})}[\log q_{\boldsymbol{g}}(\boldsymbol{X}_{\text{mis}(\boldsymbol{M})}|\boldsymbol{X}_{\text{obs}(\boldsymbol{M})},\boldsymbol{M})]$ can be rewritten as follows:

$$-\mathbb{E}_{p_*(\boldsymbol{x},y,\boldsymbol{m})}[\log q_{\boldsymbol{g}}(\boldsymbol{X}_{\text{mis}(\boldsymbol{M})}|\boldsymbol{X}_{\text{obs}(\boldsymbol{M})},\boldsymbol{M})]$$

$$= \frac{1}{2}\log(2\pi)^{|\text{mis}(\boldsymbol{M})|}\sigma^{2|\text{mis}(\boldsymbol{M})|}$$

$$+ \mathbb{E}_{p_*(\boldsymbol{x},y,\boldsymbol{m})}\left[\frac{1}{2\sigma^2}(\boldsymbol{X}_{\text{mis}(\boldsymbol{M})} - \boldsymbol{g}_{\text{mis}(\boldsymbol{M})}(\boldsymbol{X}_{\text{obs}(\boldsymbol{M})}))^{\top}(\boldsymbol{X}_{\text{mis}(\boldsymbol{M})} - \boldsymbol{g}_{\text{mis}(\boldsymbol{M})}(\boldsymbol{X}_{\text{obs}(\boldsymbol{M})}))\right]$$

$$= \mathbb{E}_{p_*(\boldsymbol{m})}\left[\frac{1}{2}\log(2\pi)^{|\text{mis}(\boldsymbol{M})|}\sigma^{2|\text{mis}(\boldsymbol{M})|}\right] + \mathbb{E}_{p_*(\boldsymbol{m})}\left[\frac{1}{2\sigma^2}\sum_{j\in\text{mis}(\boldsymbol{M})}\mathbb{E}_{p_*(\boldsymbol{x}|\boldsymbol{M})}[(X_j - g_j(\widetilde{X}))^2]\right]$$

$$= \mathbb{E}_{p_*(\boldsymbol{m})}\left[\frac{1}{2}\log(2\pi)^{|\text{mis}(\boldsymbol{M})|}\sigma^{2|\text{mis}(\boldsymbol{M})|}\right] + \mathbb{E}_{p_*(\boldsymbol{m})}\left[\frac{1}{2\sigma^2}\sum_{j\in\text{mis}(\boldsymbol{M})}R_{\text{MSE},j}^{(\boldsymbol{M})}(g_j)\right]. \tag{E.91}$$

Substituting the above into Eq. (E.90), we obtain:

$$\mathbb{E}_{p_*(\boldsymbol{m})p_*(\boldsymbol{x}_{\text{obs}(\boldsymbol{m})},y|\boldsymbol{m})}\left[\left\{D_{\text{H}}(p_*(\boldsymbol{X}_{\text{mis}(\boldsymbol{M})}|\boldsymbol{X}_{\text{obs}(\boldsymbol{M})},Y,\boldsymbol{M}),q_{\boldsymbol{g}}(\boldsymbol{X}_{\text{mis}(\boldsymbol{M})}|\boldsymbol{X}_{\text{obs}(\boldsymbol{M})},\boldsymbol{M}))\right\}^2\right]$$

$$\leq \text{LL}'_{p_*} + Z'(\sigma^2) + \mathbb{E}_{p_*(\boldsymbol{m})}\left[\frac{1}{2\sigma^2}\sum_{j\in\text{mis}(\boldsymbol{M})}R_{\text{MSE},j}^{(\boldsymbol{M})}(g_j)\right]. \tag{E.92}$$

Here, $Z'(\sigma^2) := \mathbb{E}_{p_*(\boldsymbol{m})}\left[\frac{1}{2}\log(2\pi)^{|\text{mis}(\boldsymbol{M})|}\sigma^{2|\text{mis}(\boldsymbol{M})|}\right]$.

Thus, applying the above results to Eq. (B.28) completes the proof of Lemma B.4 as stated in Equation (B.29).

$$
\begin{aligned}
&|R_l(f) - R_{l,\boldsymbol{g}}(f)| \\
&\leq \left( \sqrt{R_l(f)} + \sqrt{R_{l,\boldsymbol{g}}(f)} \right) \left\{ 2U_l \left( \mathrm{LL}'_{p_*} + Z'(\sigma^2) + \mathbb{E}_{p_*(\boldsymbol{m})} \left[ \frac{1}{2\sigma^2} \sum_{j \in \mathrm{mis}(\boldsymbol{M})} R^{(\boldsymbol{M})}_{\mathrm{MSE},j}(g_j) \right] \right) \right\}^{\frac{1}{2}}.
\end{aligned}
\tag{E.93}
$$

$\square$

### E.4. Proof of Theorem B.5

First, we use Lemma B.4 to prove the following lemma:

**Lemma E.1.** *Let $q_{\boldsymbol{g}}$ be the PDF of $\mathcal{N}(\boldsymbol{g}_{\mathrm{mis}(\boldsymbol{m})}(\boldsymbol{x}_{\mathrm{obs}(\boldsymbol{m})}), \sigma^2 I_{|\mathrm{mis}(\boldsymbol{m})| \times |\mathrm{mis}(\boldsymbol{m})|})$, where $\sigma \in \mathbb{R}_+$. For any measurable $f \in \mathcal{F}, \boldsymbol{g} \in \mathcal{G}$ and $l$ bounded above by $U_l < \infty$, the following inequality holds:*

$$
\begin{aligned}
|R_l(f) - R_{l,\boldsymbol{g}}(f)| &\leq \left\{ 2\sqrt{R_l(f)} + \left( 2U_l \left( C'_\sigma + \frac{1}{2\sigma^2} \mathbb{E}_{p_*(\boldsymbol{m})} \left[ \sum_{j \in \mathrm{mis}(\boldsymbol{M})} R^{(\boldsymbol{M})}_{\mathrm{MSE},j}(g_j) \right] \right) \right)^{\frac{1}{2}} \right\} \\
&\quad \times \left\{ 2U_l \left( C'_\sigma + \frac{1}{2\sigma^2} \mathbb{E}_{p_*(\boldsymbol{m})} \left[ \sum_{j \in \mathrm{mis}(\boldsymbol{M})} R^{(\boldsymbol{M})}_{\mathrm{MSE},j}(g_j) \right] \right) \right\}^{\frac{1}{2}},
\end{aligned}
\tag{E.94}
$$

*where $C'_\sigma = \mathbb{E}_{p_*(\boldsymbol{x},y,\boldsymbol{m})}[p_*(\boldsymbol{X}_{\mathrm{mis}(\boldsymbol{M})}|\boldsymbol{X}_{\mathrm{obs}(\boldsymbol{M})}, Y, \boldsymbol{M})] + \mathbb{E}_{p_*(\boldsymbol{m})}[\frac{1}{2}\log(2\pi)^{|\mathrm{mis}(\boldsymbol{M})|}\sigma^{2|\mathrm{mis}(\boldsymbol{M})|}]$.*

*Proof of Lemma E.1.* From Lemma B.4, for any measurable $f \in \mathcal{F}, \boldsymbol{g} \in \mathcal{G}$ and $l$ bounded above by $U_l < \infty$, the following holds:

$$
\left| \sqrt{R_l(f)} - \sqrt{R_{l,\boldsymbol{g}}(f)} \right| \leq \left\{ 2U_l \left( \mathrm{LL}'_{p_*} + Z'(\sigma^2) + \mathbb{E}_{p_*(\boldsymbol{m})} \left[ \frac{1}{2\sigma^2} \sum_{j \in \mathrm{mis}(\boldsymbol{M})} R^{(\boldsymbol{M})}_{\mathrm{MSE},j}(g_j) \right] \right) \right\}^{\frac{1}{2}}.
$$

Using the above result, $\sqrt{R_{l,\boldsymbol{g}}(f)}$ can be bounded as follows:

$$
\begin{aligned}
\sqrt{R_{l,\boldsymbol{g}}(f)} &\leq \sqrt{R_l(f)} + \left| \sqrt{R_l(f)} - \sqrt{R_{l,\boldsymbol{g}}(f)} \right| \\
&\leq \sqrt{R_l(f)} + \left\{ 2U_l \left( \mathrm{LL}'_{p_*} + Z'(\sigma^2) + \mathbb{E}_{p_*(\boldsymbol{m})} \left[ \frac{1}{2\sigma^2} \sum_{j \in \mathrm{mis}(\boldsymbol{M})} R^{(\boldsymbol{M})}_{\mathrm{MSE},j}(g_j) \right] \right) \right\}^{\frac{1}{2}}.
\end{aligned}
\tag{E.95}
$$

Substituting this inequality into Eq. (B.29) completes the proof of Lemma E.1 as given in Eq. (E.94).

$$
\begin{aligned}
|R_l(f) - R_{l,\boldsymbol{g}}(f)| &\leq \left\{ 2\sqrt{R_l(f)} + \left( 2U_l \left( \mathrm{LL}'_{p_*} + Z'(\sigma^2) + \frac{1}{2\sigma^2} \mathbb{E}_{p_*(\boldsymbol{m})} \left[ \sum_{j \in \mathrm{mis}(\boldsymbol{M})} R^{(\boldsymbol{M})}_{\mathrm{MSE},j}(g_j) \right] \right) \right)^{\frac{1}{2}} \right\} \\
&\quad \times \left\{ 2U_l \left( \mathrm{LL}'_{p_*} + Z'(\sigma^2) + \frac{1}{2\sigma^2} \mathbb{E}_{p_*(\boldsymbol{m})} \left[ \sum_{j \in \mathrm{mis}(\boldsymbol{M})} R^{(\boldsymbol{M})}_{\mathrm{MSE},j}(g_j) \right] \right) \right\}^{\frac{1}{2}}.
\end{aligned}
\tag{E.96}
$$

$\square$

Using Lemma E.1, we now prove Theorem B.5.

*Proof of Theorem B.5.* Define the empirical risk minimizer under the ordinary supervised learning setting described in Section 2.1, as follows:

$$
f_S := \arg\min_{f \in \mathcal{F}} \widehat{R}_l(f).
\tag{E.97}
$$

The LHS of Eq. (B.30) can be rewritten as follows:

$$R_{l,\boldsymbol{g}}(f_{\boldsymbol{g},\overline{S}}) - R_l(f_{\mathcal{F}}) = \underbrace{R_{l,\boldsymbol{g}}(f_{\boldsymbol{g},\overline{S}}) - \widehat{R}_{l,\boldsymbol{g}}(f_{\boldsymbol{g},\overline{S}})}_{\text{(a1)}} + \underbrace{\widehat{R}_{l,\boldsymbol{g}}(f_{\boldsymbol{g},\overline{S}}) - R_{l,\boldsymbol{g}}(f_S)}_{\text{(a2)}}$$
$$+ \underbrace{R_{l,\boldsymbol{g}}(f_S) - R_l(f_S)}_{\text{(a3)}} + \underbrace{R_l(f_S) - R_l(f_{\mathcal{F}})}_{\text{(a4)}}. \tag{E.98}$$

The terms (a1) and (a2) in Eq. (E.98) can be upper-bounded as:

$$\text{(a1)} \leq \max_{f \in \mathcal{F}} |R_{l,\boldsymbol{g}}(f) - \widehat{R}_{l,\boldsymbol{g}}(f)|,$$
$$\text{(a2)} \leq \widehat{R}_{l,\boldsymbol{g}}(f_S) - R_{l,\boldsymbol{g}}(f_S) \leq \max_{f \in \mathcal{F}} |R_{l,\boldsymbol{g}}(f) - \widehat{R}_{l,\boldsymbol{g}}(f)|. \tag{E.99}$$

The term (a3) in Eq. (E.98) can be upper-bounded using Lemma E.1 as follows:

$$\text{(a3)} \leq \left\{ 2\sqrt{R_l(f_S)} + \left( 2U_l\left( C'_\sigma + \frac{1}{2\sigma^2}\mathbb{E}_{p_*(\boldsymbol{m})}\left[ \sum_{j \in \text{mis}(\boldsymbol{M})} R^{(\boldsymbol{M})}_{\text{MSE},j}(g_j)] \right] \right) \right)^{\frac{1}{2}} \right\} \tag{E.100}$$

$$\times \left\{ 2U_l\left( C'_\sigma + \frac{1}{2\sigma^2}\mathbb{E}_{p_*(\boldsymbol{m})}\left[ \sum_{j \in \text{mis}(\boldsymbol{M})} R^{(\boldsymbol{M})}_{\text{MSE},j}(g_j)] \right] \right) \right\}^{\frac{1}{2}}. \tag{E.101}$$

Here, $\widetilde{\mathcal{F}}_l := \{(\boldsymbol{x},y) \mapsto l(f(\boldsymbol{x}),y) : f \in \mathcal{F}\}$, $C'_\sigma = \mathbb{E}_{p_*(\boldsymbol{x},y,\boldsymbol{m})}[p_*(\boldsymbol{X}_{\text{mis}(\boldsymbol{M})}|\boldsymbol{X}_{\text{obs}(\boldsymbol{M})},Y,\boldsymbol{M})] + \mathbb{E}_{p_*(\boldsymbol{m})}[\frac{1}{2}\log(2\pi)^{|\text{mis}(\boldsymbol{M})|}\sigma^{2|\text{mis}(\boldsymbol{M})|}]$. Using these, $R_l(f_S)$ can be upper-bounded as follows:

$$R_l(f_S) = R_l(f_S) - \widehat{R}_l(f_S) + \widehat{R}_l(f_S) - R_l(f_{\mathcal{F}}) + R_l(f_{\mathcal{F}})$$
$$= R_l(f_S) - \widehat{R}_l(f_S) + \widehat{R}_l(f_{\mathcal{F}}) - R_l(f_{\mathcal{F}}) + R_l(f_{\mathcal{F}})$$
$$\leq R_l(f_{\mathcal{F}}) + 2\max_{f \in \mathcal{F}} |R_l(f) - \widehat{R}_l(f)|. \tag{E.102}$$

Hence, the term (a3) in Eq. (E.98) can also be upper-bounded as:

$$\text{(a3)} \leq \left\{ 2\left( R_l(f_{\mathcal{F}}) + 2\max_{f \in \mathcal{F}}|R_l(f) - \widehat{R}_l(f)| \right)^{\frac{1}{2}} + \left( 2U_l\left( C'_\sigma + \frac{1}{2\sigma^2}\mathbb{E}_{p_*(\boldsymbol{m})}\left[ \sum_{j \in \text{mis}(\boldsymbol{M})} R^{(\boldsymbol{M})}_{\text{MSE},j}(g_j)] \right] \right) \right)^{\frac{1}{2}} \right\} \tag{E.103}$$

$$\times \left\{ 2U_l\left( C'_\sigma + \frac{1}{2\sigma^2}\mathbb{E}_{p_*(\boldsymbol{m})}\left[ \sum_{j \in \text{mis}(\boldsymbol{M})} R^{(\boldsymbol{M})}_{\text{MSE},j}(g_j)] \right] \right) \right\}^{\frac{1}{2}}. \tag{E.104}$$

Similarly, the term (a4) in Eq. (E.98) can be upper-bounded as:

$$R_l(f_S) - R_l(f_{\mathcal{F}}) \leq 2\max_{f \in \mathcal{F}} |R_l(f) - \widehat{R}_l(f)|.$$

From the above, the LHS of Eq. (B.30) can be upper-bounded as:

$$R_{l,\boldsymbol{g}}(f_{\boldsymbol{g},\overline{S}}) - R_l(f_{\mathcal{F}})$$
$$\leq 2\max_{f \in \mathcal{F}} |R_{l,\boldsymbol{g}}(f) - \widehat{R}_{l,\boldsymbol{g}}(f)| + 2\max_{f \in \mathcal{F}} |R_l(f) - \widehat{R}_l(f)|$$
$$+ \left\{ 2\left( R_l(f_{\mathcal{F}}) + 2\max_{f \in \mathcal{F}}|R_l(f) - \widehat{R}_l(f)| \right)^{\frac{1}{2}} + \left( 2U_l\left( C'_\sigma + \frac{1}{2\sigma^2}\mathbb{E}_{p_*(\boldsymbol{m})}\left[ \sum_{j \in \text{mis}(\boldsymbol{M})} R^{(\boldsymbol{M})}_{\text{MSE},j}(g_j)] \right] \right) \right)^{\frac{1}{2}} \right\} \tag{E.105}$$

$$\times \left\{ 2U_l\left( C'_\sigma + \frac{1}{2\sigma^2}\mathbb{E}_{p_*(\boldsymbol{m})}\left[ \sum_{j \in \text{mis}(\boldsymbol{M})} R^{(\boldsymbol{M})}_{\text{MSE},j}(g_j)] \right] \right) \right\}^{\frac{1}{2}}.$$

By the uniform law of large numbers (Mohri et al., 2018), for any $\delta \in (0, 1)$, the following holds with probability at least $1 - \delta/2$:

$$\max_{f \in \mathcal{F}} |R_{l,\boldsymbol{g}}(f) - \widehat{R}_{l,\boldsymbol{g}}(f)| \leq 2\mathfrak{R}_n^{\boldsymbol{g}}(\widetilde{\mathcal{F}}_l) + U_l\sqrt{\frac{\log(4/\delta)}{2n}},$$

$$\max_{f \in \mathcal{F}} |R_l(f) - \widehat{R}_l(f)| \leq 2\mathfrak{R}_n^*(\widetilde{\mathcal{F}}_l) + U_l\sqrt{\frac{\log(4/\delta)}{2n}}.$$

Here, $\widetilde{\mathcal{F}}_l := \{(\boldsymbol{x}, y) \mapsto l(f(\boldsymbol{x}), y) : f \in \mathcal{F}\}$.

From the assumption that $l$ and Lemma 26.9 in (Shalev-Shwartz & Ben-David, 2014), it follows that $\mathfrak{R}_n^{\boldsymbol{g}}(\widetilde{\mathcal{F}}_l) \leq L_l\mathfrak{R}_n^{\boldsymbol{g}}(\mathcal{F})$ and $\mathfrak{R}_n^*(\widetilde{\mathcal{F}}_l) \leq L_l\mathfrak{R}_n^*(\mathcal{F})$. Thus, for any $\delta \in (0, 1)$, the following holds with probability at least $1 - \delta$:

$$R_{l,\boldsymbol{g}}(f_{\boldsymbol{g},\overline{S}}) - R_l(f_{\mathcal{F}})$$

$$\leq 4\left(L_l\mathfrak{R}_n^*(\mathcal{F}) + L_l\mathfrak{R}_n^{\boldsymbol{g}}(\mathcal{F}) + U_l\sqrt{\frac{\log(4/\delta)}{2n}}\right)$$

$$+ \left\{2\left(R_l(f_{\mathcal{F}}) + 4L_l\mathfrak{R}_n^*(\mathcal{F}) + 2U_l\sqrt{\frac{\log(4/\delta)}{2n}}\right)^{\frac{1}{2}} + \left(2U_l\left(C'_\sigma + \frac{1}{2\sigma^2}\mathbb{E}_{p_*(\boldsymbol{m})}\left[\sum_{j \in \text{mis}(\boldsymbol{M})} R_{\text{MSE},j}^{(\boldsymbol{M})}(g_j)]\right]\right)\right)^{\frac{1}{2}}\right\}$$

(E.106)

$$\times \left\{2U_l\left(C'_\sigma + \frac{1}{2\sigma^2}\mathbb{E}_{p_*(\boldsymbol{m})}\left[\sum_{j \in \text{mis}(\boldsymbol{M})} R_{\text{MSE},j}^{(\boldsymbol{M})}(g_j)]\right]\right)\right\}^{\frac{1}{2}}.$$

$\square$

### E.5. Proof of Theorem B.6

*Proof of Theorem B.6.* Firstly, since $\mathcal{X}_j$ is a set of finite-precision decimals, for any $j \in [d]$, and $\sigma^2 = (2\pi)^{-1}$, it follows that $C'_\sigma \leq 0$. The remainder of this proof follows a structure similar to that of the proof for discrete WFL (Sugiyama & Uchida, 2025). By assumption, there exist true deterministic functions $g_j^* : \overline{\mathcal{X}} \times \{0, 1\}^d \to \mathcal{X}_j$ for any $j \in [d]$, and $(g_1^*, \ldots, g_d^*) \in \mathcal{G}$. Therefore, when $\boldsymbol{g}_{\overline{S}} = (g_{\overline{S},1}, \ldots, g_{\overline{S},d})$ is obtained by the methods that achieve consistency (Mohri et al., 2018), the following holds:

$$n \to \infty, \quad \mathbb{E}_{p_*(\boldsymbol{m})}\left[\sum_{j \in \text{mis}(\boldsymbol{M})} R_{\text{MSE},j}^{(\boldsymbol{M})}(g_{\overline{S},j})]\right] \to 0, \quad \forall j \in [d]. \tag{E.107}$$

By assumption, there exists a true deterministic function $f^* : \mathcal{X} \to \mathcal{Y}$ for label prediction, and $f^* \in \mathcal{F}$. Hence, the following holds:

$$R_l(f_{\mathcal{F}}) = 0. \tag{E.108}$$

Thus, if $\mathfrak{R}_n^*(\mathcal{F})$ and $\mathfrak{R}_n^{\boldsymbol{g}}(\mathcal{F})$ are monotonically decreasing with respect to $n$ and converge to 0, the error bound established in Theorem B.5 converges to 0 as $n \to \infty$.

$\square$

### E.6. Proof of Theorem B.8

Using Lemma E.1, we first prove the following Theorem E.2. Theorem E.2 holds for any missing mechanism.

**Theorem E.2.** *Let $q_{\boldsymbol{g}}$ be the PDF of $\mathcal{N}(\boldsymbol{g}_{\text{mis}(\boldsymbol{m})}(\boldsymbol{x}_{\text{obs}(\boldsymbol{m})}), \sigma^2 I_{|\text{mis}(\boldsymbol{m})| \times |\text{mis}(\boldsymbol{m})|})$. Let $S$ and $\overline{S}$ represent ordinary and weak datasets with $n$ samples, respectively. For any measurable $\boldsymbol{g} \in \mathcal{G}$, $l$ bounded by $U_l < \infty$ and $\delta \in (0, 1)$, the following*

*inequality holds with probability at least $1 - \delta$:*

$$
R_{l,f}(\boldsymbol{g}_{f,\overline{S}}^{(\boldsymbol{r})}) - R_l(f)
$$

$$
\leq 2\left( \mathfrak{R}_n^*(\widetilde{\mathcal{G}}_j(r_j, \overline{S}_j)) + U_l\sqrt{\frac{\log(1/\delta)}{2n}} \right)
$$

$$
+ 2\Bigg\{ 2\pi U_l \Bigg( R_l(f)
$$

$$
+ U_l\Bigg(\frac{\pi}{2}\Bigg(\max_{g_j \in \mathcal{G}_j(r_j,\overline{S}_j)} \Bigg| \mathbb{E}_{p_*(\boldsymbol{m})}\Bigg[ \sum_{j\in\mathrm{mis}(\boldsymbol{M})} R_{\mathrm{MSE},j}^{(\boldsymbol{M})}\Big(g_{\overline{S},j}^{(r_j)}\Big)\Bigg] - \mathbb{E}_{\hat{p}(\boldsymbol{m})}\Bigg[ \sum_{j\in\mathrm{mis}(\boldsymbol{M})} \widehat{R}_{\mathrm{MSE},j}^{(\boldsymbol{M})}\Big(g_{\overline{S},j}^{(r_j)}\Big)\Bigg] \Bigg| \tag{E.109}
$$

$$
+ \sum_{j\in[d]} r_j \Bigg)\Bigg)^{\frac{1}{2}}\Bigg)
$$

$$
\times \Bigg( \max_{g_j \in \mathcal{G}_j(r_j,\overline{S}_j)} \Bigg| \mathbb{E}_{p_*(\boldsymbol{m})}\Bigg[ \sum_{j\in\mathrm{mis}(\boldsymbol{M})} R_{\mathrm{MSE},j}^{(\boldsymbol{M})}\Big(g_{\overline{S},j}^{(r_j)}\Big)\Bigg] - \mathbb{E}_{\hat{p}(\boldsymbol{m})}\Bigg[ \sum_{j\in\mathrm{mis}(\boldsymbol{M})} \widehat{R}_{\mathrm{MSE},j}^{(\boldsymbol{M})}\Big(g_{\overline{S},j}^{(r_j)}\Big)\Bigg] \Bigg|
$$

$$
+ \sum_{j\in[d]} r_j \Bigg)\Bigg)^{\frac{1}{2}}\Bigg)\Bigg\}^{\frac{1}{2}}.
$$

*Here,* $\widetilde{\mathcal{G}}_j(r_j, \overline{S}_j) := \{(\boldsymbol{x}^{\mathrm{o}}, x_j^{\mathrm{w}}) \mapsto l_{\mathrm{MSE}}(g_j(\boldsymbol{x}^{\mathrm{o}}), x_j^{\mathrm{w}}) : g_j \in \mathcal{G}_j(r_j, \overline{S}_j)\},$ $\forall j \in [F^{\mathrm{w}}],$ $C_\sigma' = \mathbb{E}_{p_*(\boldsymbol{x},y,\boldsymbol{m})}[p_*(\boldsymbol{X}_{\mathrm{mis}(\boldsymbol{M})}|\boldsymbol{X}_{\mathrm{obs}(\boldsymbol{M})}, Y, \boldsymbol{M})] + \mathbb{E}_{p_*(\boldsymbol{m})}[\frac{1}{2}\log(2\pi)^{|\mathrm{mis}(\boldsymbol{M})|}\sigma^{2|\mathrm{mis}(\boldsymbol{M})|}].$

*Proof of Theorem E.2.* The LHS of Eq. (E.109) can be rewritten as:

$$
R_{l,f}(\boldsymbol{g}_{f,\overline{S}}^{(\boldsymbol{r})}) - R_l(f) = \underbrace{R_{l,f}(\boldsymbol{g}_{f,\overline{S}}^{(\boldsymbol{r})}) - \widehat{R}_{l,f}(\boldsymbol{g}_{f,\overline{S}}^{(\boldsymbol{r})})}_{(a1)} + \underbrace{\widehat{R}_{l,f}(\boldsymbol{g}_{f,\overline{S}}^{(\boldsymbol{r})}) - R_{l,f}(\boldsymbol{g}_{\overline{S}}^{(\boldsymbol{r})})}_{(a2)} + \underbrace{R_{l,f}(\boldsymbol{g}_{\overline{S}}^{(\boldsymbol{r})}) - R_l(f)}_{(a3)}. \tag{E.110}
$$

The terms (a1) and (a2) in Eq. (E.110) can be upper-bounded as follows:

$$
(a1) \leq \max_{\boldsymbol{g}\in\mathcal{G}(\boldsymbol{r},\overline{S})} |R_{l,f}(\boldsymbol{g}) - \widehat{R}_{l,f}(\boldsymbol{g})|, \tag{E.111}
$$

$$
(a2) \leq \widehat{R}_{l,f}(\boldsymbol{g}_{\overline{S}}^{(\boldsymbol{r})}) - R_{l,f}(\boldsymbol{g}_{\overline{S}}^{(\boldsymbol{r})}) \leq \max_{\boldsymbol{g}\in\mathcal{G}(\boldsymbol{r},\overline{S})} |R_{l,f}(\boldsymbol{g}) - \widehat{R}_{l,f}(\boldsymbol{g})|. \tag{E.112}
$$

The term (a3) in Eq. (E.110) can be upper-bounded using Lemma E.1 as follows:

$$
(a3) \leq |R_{l,f}(\boldsymbol{g}_{\overline{S}}^{(\boldsymbol{r})}) - R_l(f)|
$$

$$
\leq \left\{ 2\sqrt{R_l(f)} + \left( 2U_l\left( C_\sigma' + \frac{1}{2\sigma^2}\mathbb{E}_{p_*(\boldsymbol{m})}\Bigg[ \sum_{j\in\mathrm{mis}(\boldsymbol{M})} R_{\mathrm{MSE},j}^{(\boldsymbol{M})}\Big(g_{\overline{S},j}^{(r_j)}\Big)\Bigg] \right)\right)^{\frac{1}{2}} \right\} \tag{E.113}
$$

$$
\times \left\{ 2U_l\left( C_\sigma' + \frac{1}{2\sigma^2}\mathbb{E}_{p_*(\boldsymbol{m})}\Bigg[ \sum_{j\in\mathrm{mis}(\boldsymbol{M})} R_{\mathrm{MSE},j}^{(\boldsymbol{M})}\Big(g_{\overline{S},j}^{(r_j)}\Big)\Bigg] \right)\right\}^{\frac{1}{2}}.
$$

By definition, the following holds:

$$
\mathbb{E}_{p_*(\boldsymbol{m})}\left[\sum_{j\in\mathrm{mis}(\boldsymbol{M})} R_{\mathrm{MSE},j}^{(\boldsymbol{M})}\left(g_{\overline{S},j}^{(r_j)}\right)\right]
$$

$$
= \mathbb{E}_{p_*(\boldsymbol{m})}\left[\sum_{j\in\mathrm{mis}(\boldsymbol{M})} R_{\mathrm{MSE},j}^{(\boldsymbol{M})}\left(g_{\overline{S},j}^{(r_j)}\right)\right] - \mathbb{E}_{\hat{p}(\boldsymbol{m})}\left[\sum_{j\in\mathrm{mis}(\boldsymbol{M})} \widehat{R}_{\mathrm{MSE},j}^{(\boldsymbol{M})}\left(g_{\overline{S},j}^{(r_j)}\right)\right]
$$

$$
+ \mathbb{E}_{\hat{p}(\boldsymbol{m})}\left[\sum_{j\in\mathrm{mis}(\boldsymbol{M})} \widehat{R}_{\mathrm{MSE},j}^{(\boldsymbol{M})}\left(g_{\overline{S},j}^{(r_j)}\right)\right] - \mathbb{E}_{p(\boldsymbol{m})}\left[\sum_{j\in\mathrm{mis}(\boldsymbol{M})} R_{\mathrm{MSE},j}^{(\boldsymbol{M})}\left(g_{\mathcal{G}(\boldsymbol{r},\overline{S}),j}\right)\right]
$$

$$
+ \mathbb{E}_{p(\boldsymbol{m})}\left[\sum_{j\in\mathrm{mis}(\boldsymbol{M})} R_{\mathrm{MSE},j}^{(\boldsymbol{M})}\left(g_{\mathcal{G}(\boldsymbol{r},\overline{S}),j}\right)\right]
$$

$$
\leq \mathbb{E}_{p_*(\boldsymbol{m})}\left[\sum_{j\in\mathrm{mis}(\boldsymbol{M})} R_{\mathrm{MSE},j}^{(\boldsymbol{M})}\left(g_{\overline{S},j}^{(r_j)}\right)\right] - \mathbb{E}_{\hat{p}(\boldsymbol{m})}\left[\sum_{j\in\mathrm{mis}(\boldsymbol{M})} \widehat{R}_{\mathrm{MSE},j}^{(\boldsymbol{M})}\left(g_{\overline{S},j}^{(r_j)}\right)\right]
$$

$$
+ \mathbb{E}_{\hat{p}(\boldsymbol{m})}\left[\sum_{j\in\mathrm{mis}(\boldsymbol{M})} \widehat{R}_{\mathrm{MSE},j}^{(\boldsymbol{M})}\left(g_{\mathcal{G}(\boldsymbol{r},\overline{S}),j}\right)\right] - \mathbb{E}_{p(\boldsymbol{m})}\left[\sum_{j\in\mathrm{mis}(\boldsymbol{M})} R_{\mathrm{MSE},j}^{(\boldsymbol{M})}\left(g_{\mathcal{G}(\boldsymbol{r},\overline{S}),j}\right)\right]
$$

$$
+ \mathbb{E}_{p(\boldsymbol{m})}\left[\sum_{j\in\mathrm{mis}(\boldsymbol{M})} R_{\mathrm{MSE},j}^{(\boldsymbol{M})}\left(g_{\mathcal{G}(\boldsymbol{r},\overline{S}),j}\right)\right]
$$

$$
\leq 2 \max_{g_j\in\mathcal{G}_j(r_j,S_j)} \left|\mathbb{E}_{p_*(\boldsymbol{m})}\left[\sum_{j\in\mathrm{mis}(\boldsymbol{M})} R_{\mathrm{MSE},j}^{(\boldsymbol{M})}\left(g_{\overline{S},j}^{(r_j)}\right)\right] - \mathbb{E}_{\hat{p}(\boldsymbol{m})}\left[\sum_{j\in\mathrm{mis}(\boldsymbol{M})} \widehat{R}_{\mathrm{MSE},j}^{(\boldsymbol{M})}\left(g_{\overline{S},j}^{(r_j)}\right)\right]\right|
$$

$$
+ \mathbb{E}_{p(\boldsymbol{m})}\left[\sum_{j\in\mathrm{mis}(\boldsymbol{M})} R_{\mathrm{MSE},j}^{(\boldsymbol{M})}\left(g_{\mathcal{G}(\boldsymbol{r},\overline{S}),j}\right)\right]
$$

$$\tag{E.114}$$

Thus, the term (a3) in Eq. (E.110) can be further upper-bounded as:

$$
\text{(a3)} \leq \left\{ 2\sqrt{R_l(f)} + \left( 2U_l\left( C'_\sigma + \frac{1}{2\sigma^2}\left( \mathbb{E}_{p(\boldsymbol{m})}\left[\sum_{j\in\mathrm{mis}(\boldsymbol{M})} R_{\mathrm{MSE},j}^{(\boldsymbol{M})}\left(g_{\mathcal{G}(\boldsymbol{r},\overline{S}),j}\right)\right] \right.\right.\right.\right.
$$

$$
\left.\left.\left.\left. + 2\max_{g_j\in\mathcal{G}_j(r_j,\overline{S}_j)} \left|\mathbb{E}_{p_*(\boldsymbol{m})}\left[\sum_{j\in\mathrm{mis}(\boldsymbol{M})} R_{\mathrm{MSE},j}^{(\boldsymbol{M})}\left(g_{\overline{S},j}^{(r_j)}\right)\right] - \mathbb{E}_{\hat{p}(\boldsymbol{m})}\left[\sum_{j\in\mathrm{mis}(\boldsymbol{M})} \widehat{R}_{\mathrm{MSE},j}^{(\boldsymbol{M})}\left(g_{\overline{S},j}^{(r_j)}\right)\right]\right| \right)\right)\right)^{\frac{1}{2}} \right\}
$$

$$
\times \left\{ 2U_l\left( C'_\sigma + \frac{1}{2\sigma^2}\left( \mathbb{E}_{p(\boldsymbol{m})}\left[\sum_{j\in\mathrm{mis}(\boldsymbol{M})} R_{\mathrm{MSE},j}^{(\boldsymbol{M})}\left(g_{\mathcal{G}(\boldsymbol{r},\overline{S}),j}\right)\right] \right.\right.\right.
$$

$$
\left.\left.\left. + 2\max_{g_j\in\mathcal{G}_j(r_j,\overline{S}_j)} \left|\mathbb{E}_{p_*(\boldsymbol{m})}\left[\sum_{j\in\mathrm{mis}(\boldsymbol{M})} R_{\mathrm{MSE},j}^{(\boldsymbol{M})}\left(g_{\overline{S},j}^{(r_j)}\right)\right] - \mathbb{E}_{\hat{p}(\boldsymbol{m})}\left[\sum_{j\in\mathrm{mis}(\boldsymbol{M})} \widehat{R}_{\mathrm{MSE},j}^{(\boldsymbol{M})}\left(g_{\overline{S},j}^{(r_j)}\right)\right]\right| \right)\right)\right\}^{\frac{1}{2}}.
$$

$$\tag{E.115}$$

Applying the above results to Eq. (E.110), we have:

$$
\begin{aligned}
&R_{l,f}(\boldsymbol{g}_{f,\overline{S}}^{(\boldsymbol{r})}) - R_l(f) \\
&\leq 2 \max_{\boldsymbol{g}\in\mathcal{G}(\boldsymbol{r},\overline{S})} |R_{l,f}(\boldsymbol{g}) - \widehat{R}_{l,f}(\boldsymbol{g})| \\
&+ \left\{ 2\sqrt{R_l(f)} + \left( 2U_l\left( C'_\sigma + \frac{1}{2\sigma^2}\left( \mathbb{E}_{p(\boldsymbol{m})}\left[ \sum_{j\in\mathrm{mis}(\boldsymbol{M})} R^{(\boldsymbol{M})}_{\mathrm{MSE},j}\left( g_{\mathcal{G}(\boldsymbol{r},\overline{S}),j} \right) \right] \right. \right. \right. \right. \\
&\qquad\qquad \left. \left. \left. \left. + 2\max_{g_j\in\mathcal{G}_j(r_j,\overline{S}_j)} \left| \mathbb{E}_{p_*(\boldsymbol{m})}\left[ \sum_{j\in\mathrm{mis}(\boldsymbol{M})} R^{(\boldsymbol{M})}_{\mathrm{MSE},j}\left( g^{(r_j)}_{\overline{S},j} \right) \right] - \mathbb{E}_{\hat{p}(\boldsymbol{m})}\left[ \sum_{j\in\mathrm{mis}(\boldsymbol{M})} \widehat{R}^{(\boldsymbol{M})}_{\mathrm{MSE},j}\left( g^{(r_j)}_{\overline{S},j} \right) \right] \right| \right) \right) \right)^{\frac{1}{2}} \right\} \\
&\times \left\{ 2U_l\left( C'_\sigma + \frac{1}{2\sigma^2}\left( \mathbb{E}_{p(\boldsymbol{m})}\left[ \sum_{j\in\mathrm{mis}(\boldsymbol{M})} R^{(\boldsymbol{M})}_{\mathrm{MSE},j}\left( g_{\mathcal{G}(\boldsymbol{r},\overline{S}),j} \right) \right] \right. \right. \right. \\
&\qquad\qquad \left. \left. \left. + 2\max_{g_j\in\mathcal{G}_j(r_j,\overline{S}_j)} \left| \mathbb{E}_{p_*(\boldsymbol{m})}\left[ \sum_{j\in\mathrm{mis}(\boldsymbol{M})} R^{(\boldsymbol{M})}_{\mathrm{MSE},j}\left( g^{(r_j)}_{\overline{S},j} \right) \right] - \mathbb{E}_{\hat{p}(\boldsymbol{m})}\left[ \sum_{j\in\mathrm{mis}(\boldsymbol{M})} \widehat{R}^{(\boldsymbol{M})}_{\mathrm{MSE},j}\left( g^{(r_j)}_{\overline{S},j} \right) \right] \right| \right) \right) \right\}^{\frac{1}{2}}.
\end{aligned}
$$
(E.116)

Using the uniform law of large numbers (Mohri et al., 2018), for any $\delta \in (0,1)$, the following inequality holds with probability at least $1 - \delta$:

$$
\max_{\boldsymbol{g}\in\mathcal{G}(\boldsymbol{r},\overline{S})} |R_{l,f}(\boldsymbol{g}) - \widehat{R}_{l,f}(\boldsymbol{g})| \leq 2\mathfrak{R}^*_n(\widetilde{\mathcal{G}}_{l,f}(\boldsymbol{r},\overline{S})) + U_l\sqrt{\frac{\log(2/\delta)}{2n}}.
$$
(E.117)

Hence, for any $\delta \in (0,1)$, Eq. (E.109) holds with probability at least $1 - \delta$:

$$
\begin{aligned}
&R_{l,f}(\boldsymbol{g}_{f,\overline{S}}^{(\boldsymbol{r})}) - R_l(f) \\
&\leq 2\left( \mathfrak{R}^*_n(\widetilde{\mathcal{G}}_j(r_j,\overline{S}_j)) + U_l\sqrt{\frac{\log(1/\delta)}{2n}} \right) \\
&+ \left\{ 2\sqrt{R_l(f)} + \left( 2U_l\left( C'_\sigma + \frac{1}{2\sigma^2}\left( \mathbb{E}_{p(\boldsymbol{m})}\left[ \sum_{j\in\mathrm{mis}(\boldsymbol{M})} R^{(\boldsymbol{M})}_{\mathrm{MSE},j}\left( g_{\mathcal{G}(\boldsymbol{r},\overline{S}),j} \right) \right] \right. \right. \right. \right. \\
&\qquad\qquad \left. \left. \left. \left. + 2\max_{g_j\in\mathcal{G}_j(r_j,\overline{S}_j)} \left| \mathbb{E}_{p_*(\boldsymbol{m})}\left[ \sum_{j\in\mathrm{mis}(\boldsymbol{M})} R^{(\boldsymbol{M})}_{\mathrm{MSE},j}\left( g^{(r_j)}_{\overline{S},j} \right) \right] - \mathbb{E}_{\hat{p}(\boldsymbol{m})}\left[ \sum_{j\in\mathrm{mis}(\boldsymbol{M})} \widehat{R}^{(\boldsymbol{M})}_{\mathrm{MSE},j}\left( g^{(r_j)}_{\overline{S},j} \right) \right] \right| \right) \right) \right)^{\frac{1}{2}} \right\} \\
&\times \left\{ 2U_l\left( C'_\sigma + \frac{1}{2\sigma^2}\left( \mathbb{E}_{p(\boldsymbol{m})}\left[ \sum_{j\in\mathrm{mis}(\boldsymbol{M})} R^{(\boldsymbol{M})}_{\mathrm{MSE},j}\left( g_{\mathcal{G}(\boldsymbol{r},\overline{S}),j} \right) \right] \right. \right. \right. \\
&\qquad\qquad \left. \left. \left. + 2\max_{g_j\in\mathcal{G}_j(r_j,\overline{S}_j)} \left| \mathbb{E}_{p_*(\boldsymbol{m})}\left[ \sum_{j\in\mathrm{mis}(\boldsymbol{M})} R^{(\boldsymbol{M})}_{\mathrm{MSE},j}\left( g^{(r_j)}_{\overline{S},j} \right) \right] - \mathbb{E}_{\hat{p}(\boldsymbol{m})}\left[ \sum_{j\in\mathrm{mis}(\boldsymbol{M})} \widehat{R}^{(\boldsymbol{M})}_{\mathrm{MSE},j}\left( g^{(r_j)}_{\overline{S},j} \right) \right] \right| \right) \right) \right\}^{\frac{1}{2}}.
\end{aligned}
$$
(E.118)

$\square$

By adding the MCAR constraint to Theorem E.2, we prove Theorem B.8 as follows:

*Proof of Theorem B.8.* First, we have:

$$
\left| \mathbb{E}_{p_*(\boldsymbol{m})}\left[ \sum_{j \in \mathrm{mis}(\boldsymbol{M})} R_{\mathrm{MSE},j}^{(\boldsymbol{M})}\left( g_{\overline{S},j}^{(r_j)} \right) \right] - \mathbb{E}_{\hat{p}(\boldsymbol{m})}\left[ \sum_{j \in \mathrm{mis}(\boldsymbol{M})} \widehat{R}_{\mathrm{MSE},j}^{(\boldsymbol{M})}\left( g_{\overline{S},j}^{(r_j)} \right) \right] \right|
$$

$$
\leq \sum_{j \in \mathrm{mis}(\boldsymbol{M})} \left| \mathbb{E}_{p_*(\boldsymbol{m})}\left[ R_{\mathrm{MSE},j}^{(\boldsymbol{M})}\left( g_{\overline{S},j}^{(r_j)} \right) \right] - \mathbb{E}_{\hat{p}(\boldsymbol{m})}\left[ \widehat{R}_{\mathrm{MSE},j}^{(\boldsymbol{M})}\left( g_{\overline{S},j}^{(r_j)} \right) \right] \right|. \tag{E.119}
$$

Here, we define $\widetilde{\mathcal{G}}_j(r_j, \overline{S}_j) := \{(\boldsymbol{x}, \boldsymbol{m}) \mapsto l_{\mathrm{MSE}}(x_j, g_j(\bar{\boldsymbol{x}}, \boldsymbol{m})) : g_j \in \mathcal{G}_j(r_j, \overline{S}_j)\}$. Under the MCAR assumption, Theorem 11.3 of (Mohri et al., 2018) implies that for any $j \in [d]$ and any $\delta \in (0, 1)$, the following holds with probability at least $1 - \delta/(d+1)$:

$$
\left| \mathbb{E}_{p_*(\boldsymbol{m})}\left[ R_{\mathrm{MSE},j}^{(\boldsymbol{M})}\left( g_{\overline{S},j}^{(r_j)} \right) \right] - \mathbb{E}_{\hat{p}(\boldsymbol{m})}\left[ \widehat{R}_{\mathrm{MSE},j}^{(\boldsymbol{M})}\left( g_{\overline{S},j}^{(r_j)} \right) \right] \right| \leq 2\mathfrak{R}_n^*(\widetilde{\mathcal{G}}_j(r_j, \overline{S}_j)) + U_l \sqrt{\frac{\log((d+1)/\delta)}{2n}}. \tag{E.120}
$$

Thus, for any $\delta \in (0, 1)$, Eq. (B.32) holds with probability at least $1 - \delta$:

$$
R_{l,f}(\boldsymbol{g}_{f,\overline{S}}^{(\boldsymbol{r})}) - R_l(f)
$$

$$
\leq 2\left( \mathfrak{R}_n^*(\widetilde{\mathcal{G}}_j(r_j, \overline{S}_j)) + U_l \sqrt{\frac{\log((d+1)/\delta)}{2n}} \right)
$$

$$
+ \left\{ 2\sqrt{R_l(f)} + \left( 2U_l\left( C_\sigma' + \frac{1}{2\sigma^2}\left( \mathbb{E}_{p(\boldsymbol{m})}\left[ \sum_{j \in \mathrm{mis}(\boldsymbol{M})} R_{\mathrm{MSE},j}^{(\boldsymbol{M})}\left( g_{\mathcal{G}(\boldsymbol{r},\overline{S}),j} \right) \right] \right.\right.\right.\right.
$$

$$
\left.\left.\left.\left. + 4\mathfrak{R}_n^*(\widetilde{\mathcal{G}}_j(r_j, \overline{S}_j)) + 2U_l\sqrt{\frac{\log((d+1)/\delta)}{2n}} \right) \right) \right)^{\frac{1}{2}} \right\}
$$

$$
\times \left\{ 2U_l\left( C_\sigma' + \frac{1}{2\sigma^2}\left( \mathbb{E}_{p(\boldsymbol{m})}\left[ \sum_{j \in \mathrm{mis}(\boldsymbol{M})} R_{\mathrm{MSE},j}^{(\boldsymbol{M})}\left( g_{\mathcal{G}(\boldsymbol{r},\overline{S}),j} \right) \right] + 4\mathfrak{R}_n^*(\widetilde{\mathcal{G}}_j(r_j, \overline{S}_j)) + 2U_l\sqrt{\frac{\log((d+1)/\delta)}{2n}} \right) \right) \right\}^{\frac{1}{2}}. \tag{E.121}
$$

$\square$

### E.7. Proof of Theorem B.9

*Proof of Theorem B.9.* Firstly, since $\mathcal{X}_j$ is a set of finite-precision decimals, for any $j \in [d]$, and $\sigma^2 = (2\pi)^{-1}$, it follows that $C_\sigma' \leq 0$. The remainder of this proof follows a structure similar to that of the proof for discrete WFL (Sugiyama & Uchida, 2025). From assumption, there exist true deterministic functions $g_j^* : \overline{\mathcal{X}} \times \{0,1\}^d \to \mathcal{X}_j$ for any $j \in [d]$, and $(g_1^*, \ldots, g_d^*) \in \mathcal{G}$. In this case, for any $\boldsymbol{r}$ and $\overline{S}$, it holds that $\boldsymbol{g}^* \in \mathcal{G}(\boldsymbol{r}, \overline{S})$. Hence, the following holds:

$$
R_{\mathrm{MSE},j}^{(\boldsymbol{m})}\left( g_{\mathcal{G}(\boldsymbol{r},\overline{S}),j} \right) = 0, \ \ \forall j \in [d], \forall \boldsymbol{m} \in \{0,1\}^d. \tag{E.122}
$$

For any $j \in [d]$, define $\mathcal{G}_j(r_j, \overline{S}_j)$ as the set of hypotheses that satisfy the following two conditions: (i) each element is a solution obtained by methods that are guaranteed to achieve consistency (Mohri et al., 2018), and (ii) its empirical risk is at most $r_j$. As $n$ increases and $r_j \to 0$, the theoretical guarantees of these learning methods for $g_j$ imply the following:

$$
R_{\mathrm{MSE},j}^{(\boldsymbol{m})}(g_j) \to 0, \ \ \forall g_j \in \mathcal{G}_j(r_j, \overline{S}_j), \ \ \forall j \in [d], \forall \boldsymbol{m} \in \{0,1\}^d. \tag{E.123}
$$

Additionally, by assumption, there exists a true deterministic function $f^* : \mathcal{X} \to \mathcal{Y}$ for label prediction, and $f^* \in \mathcal{F}$. Therefore, the following holds:

$$
R_l(f_{\mathcal{F}}) = 0. \tag{E.124}
$$

Thus, under the conditions of Theorem B.6, the following holds for $f_{\boldsymbol{g},\overline{S}}$ obtained through LAC-dWFL's step (ii):

$$n \to \infty, \;\; R_{l,\boldsymbol{g}}(f_{\boldsymbol{g},\overline{S}}) \to 0. \tag{E.125}$$

Furthermore, using Theorem B.1 with $C'_\sigma \leq 0$ and $\sigma^2 = (2\pi)^{-1}$, and additionally letting $r_j \to 0$ as $n$ increases for any $j \in [d]$, the following holds:

$$n \to \infty, \;\; R_l(f_{\boldsymbol{g},\overline{S}}) \to 0, \;\; \forall \boldsymbol{g} \in \mathcal{G}(\boldsymbol{r},\overline{S}). \tag{E.126}$$

Since $l$ is $L_l$-Lipschitz continuous and $f_{\boldsymbol{g},\overline{S}}$ is $L_f$-Lipschitz continuous, the following holds (Shalev-Shwartz & Ben-David, 2014):

$$\mathfrak{R}_n^*(\tilde{\mathcal{G}}_{l,f}(\boldsymbol{r},\boldsymbol{S})) \leq L_l L_f \mathfrak{R}_n^*(\mathcal{G}(\boldsymbol{r},\boldsymbol{S})). \tag{E.127}$$

Consequently, if $\mathfrak{R}_n^*(\mathcal{G}(\boldsymbol{r},\boldsymbol{S}))$ and $\mathfrak{R}_n^*(\mathcal{G}_j(r_j,\overline{S}_j))$ for any $j \in [d]$ are monotonically decreasing with respect to $n$ and converge to 0, the error bound established in Theorem B.8 converges to 0 as $n \to \infty$. $\qquad\square$

# F. Detail Information of Experiments

## F.1. Details of Datasets

For real-world datasets, we utilize four datasets from OpenML (Vanschoren et al., 2013): *hls4ml_lhc_jets_hlf* (Pierini et al., 2020), *electricity* (Harries et al., 2014), *mv* (Luis, 2014), and *Run_or_walk_information* (Viktor, 2017). We refer to them as *Jets*, *Electricity*, *Mv*, and *Run-or-Walk*, respectively. Table F.1 summarizes these datasets. All binary features were constrained to take values of either 0 or 1. One-hot encoding was applied to all categorical features, and all continuous features were scaled to ensure their values fall within the range $[0,1]$.

*Table F.1.* Outline of datasets. binary, categorical, and numerical represent the number of features of each type, respectively.

| dataset | Jets | Electricity | Mv | Run-or-Walk |
|---|---|---|---|---|
| data size | 830000 | 45312 | 40768 | 88588 |
| binary | 0 | 0 | 2 | 0 |
| categorical | 0 | 0 | 1 | 0 |
| numerical | 8 | 7 | 7 | 6 |
| target | binary | binary | binary | binary |

## F.2. Details of Experimental setup

The experimental settings, except for the dataset, were almost identical to those used in the discrete WFL experiments (Sugiyama & Uchida, 2025). The differences lie in the settings for $\sigma^2$ and $\mathbb{E}_{p_*(\boldsymbol{x},y)}[\log p_*(\boldsymbol{X}^{\mathrm{w}}|\boldsymbol{X}^{\mathrm{o}},Y)]$. In this experiment, $\sigma^2$ was set to $(2\pi)^{-1}$. For continuous features, all were assumed to be finite-precision decimals, and $\mathbb{E}_{p_*(\boldsymbol{x},y)}[\log p_*(\boldsymbol{X}^{\mathrm{w}}|\boldsymbol{X}^{\mathrm{o}},Y)] \leq 0$ was assumed. Consequently, the bound was computed by setting $C_\sigma = 0$.

All other settings follow those in the discrete WFL experiment (Sugiyama & Uchida, 2025), as follows. The feature estimation models $\boldsymbol{g}$ and the label prediction model $f$ were represented as two-layer perceptrons with hidden layers of width 500, using ReLU as the activation function. All models were trained using Adam (Kingma & Ba, 2014) and the following hyperparameters: learning rate of 0.0005, batch size of 512, 100 epochs, and a weight decay of 0.0002. Logistic loss is employed for training $f$.

We summarize the method used to calculate the error bound presented in Theorem 4.3 for this experiment. First, the numerical values related to the error bound were set as $\delta = 0.0001$, $R_l(f_{\mathcal{F}}) = 0$, and $U_l = 2.0$. $R_l(f_{\mathcal{F}}) = 0$ was set under the assumption that $\mathcal{F}$ is sufficiently complex. In this experiment, the predicted label is determined by the label corresponding to the largest output value of $f$. Since scaling the outputs of $f$ does not affect the results, it was assumed that the maximum value of each element in $f$'s output is 1, and $U_l$ was set to 2.0. The logistic loss $l$, used in this experiment, is 1-Lipschitz continuous. Therefore, it follows that $\mathfrak{R}_n^*(\widetilde{\mathcal{F}}_l) = \mathfrak{R}_n^*(\mathcal{F})$ and $\mathfrak{R}_n^{\boldsymbol{g}}(\widetilde{\mathcal{F}}_l) = \mathfrak{R}_n^{\boldsymbol{g}}(\mathcal{F})$ (Lemma 26.9 in (Shalev-Shwartz & Ben-David, 2014)).

Similar to the experiment in discrete WFL (Sugiyama & Uchida, 2025), $\mathfrak{R}_n^*(\mathcal{F})$ and $\mathfrak{R}_n^g(\mathcal{F})$ were calculated using the upper bound on the Rademacher complexity of a multilayer perceptron derived by Neyshabur et al. (Theorem 1 in (Neyshabur et al., 2015)). For the parameters in this bound, we set $\mu = 1$ and $p = q = 2$. The parameter $\mu$ represents the upper bound on the $l_p$-norm of all parameters in the model, and $p$ specifies the type of $l_p$-norm. Since all parameters of $f$ can be scaled without affecting the inference results for predicting a single label, $\mu$ was set to 1.

## G. Additional Experiments

In Section 5, we experimentally validated our theoretical results under the setting where all WFs are continuous variables that may contain missing values. In this setting, since the exact values of WFs are not always missing, some instances have observed exact values for certain WFs. In contrast, this section investigates scenarios in which all WFs for all instances are observed either with substantial noise or as intervals that contain the exact values.

Figure G.3 illustrates the relationship between the generalization error of $f_{\boldsymbol{g},\overline{S}}$ and the number of training samples $n$, for several $\boldsymbol{g}$ with varying levels of estimation error. Figure G.4 compares these generalization errors with the theoretical error bound presented in Theorem 4.3. Consistent with the results in Section 5, these findings confirm that our derived error bound effectively captures the relationship between the estimation error of $\boldsymbol{g}$ and the rate at which the generalization error of $f_{\boldsymbol{g},\overline{S}}$ decreases as $n$ increases.

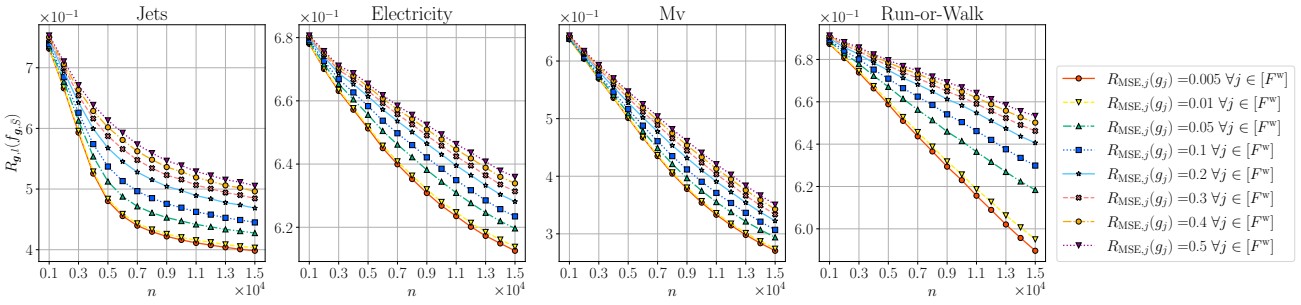

*Figure G.3.* The relationship between $R_{l,\boldsymbol{g}}(f_{\boldsymbol{g},\overline{S}})$ and the training data size $n$ for various MSEs of $\boldsymbol{g}$. This figure shows the cases in which all WFs for all instances are observed either with substantial noise or as intervals that contain the exact values.

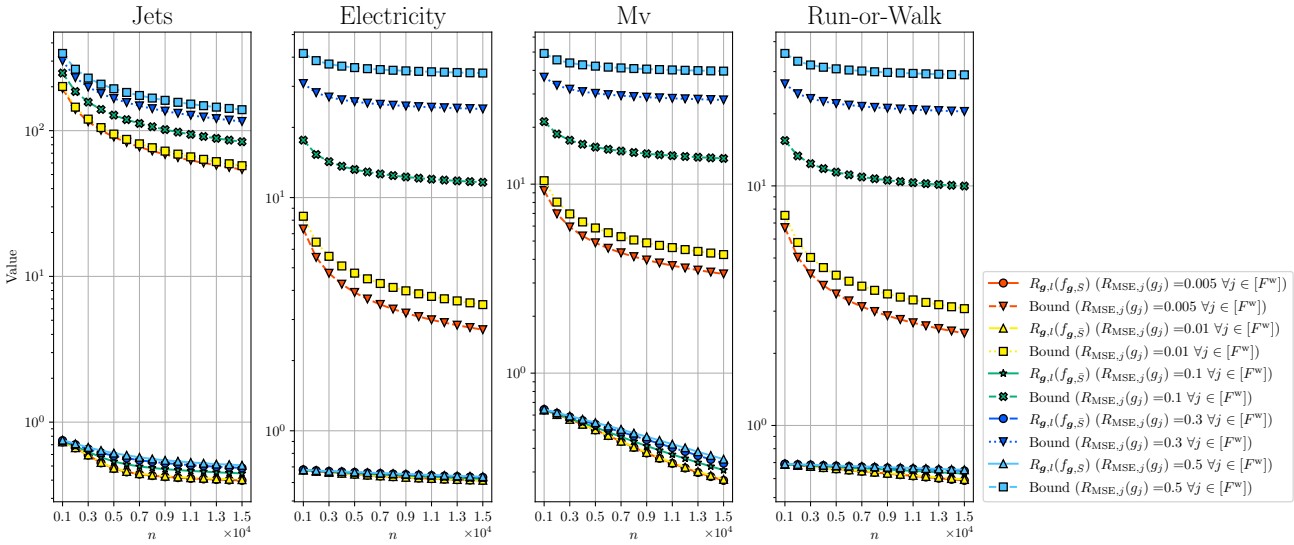

*Figure G.4.* A comparison between $R_{l,\boldsymbol{g}}(f_{\boldsymbol{g},\overline{S}})$ for various MSEs of $\boldsymbol{g}$ and the error bound derived in Theorem 4.3.

## H. Limitation

In this paper, we provided a unified theoretical analysis of continuous WFL, in parallel with the discrete WFL framework proposed by (Sugiyama & Uchida, 2025). At the same time, our analysis inherits several limitations similar to those in the discrete WFL setting (Sugiyama & Uchida, 2025).

The first limitation is that the derived error bounds do not take into account the feature importance of each WF with respect to the downstream prediction task. Extending the theoretical framework to incorporate feature importance remains an important direction for future research. The second limitation is that our analysis does not cover joint learning approaches where $f$ and $g$ are represented by a single model, nor does it handle settings in which the estimated outputs for some WFs are used as input features for estimating other WFs. Such more complex formulations are often important in practical applications, and developing methods and theoretical guarantees for them is a promising avenue for future work. It is considered that the theoretical results presented in this paper provide a solid foundation for addressing both of these future directions.

