# OpenReview forum: "Unified Analysis of Continuous Weak Features Learning with Applications to Learning from Missing Data"
_ICML.cc/2025/Conference — ICML 2025 poster_

### Official Review · Reviewer_dvad · 2025-03-06

**Overall Recommendation:** 4

**Summary:**

This paper proposes a unified theoretical framework for Continuous Weak Feature Learning (or continuous WFL), addressing scenarios where input features are low-quality due to missing data, measurement errors, or ambiguous observations. The authors introduce a novel risk-based formulation specifically tailored for continuous weak features, distinguishing it clearly from previous work that focused only on discrete weak features. The paper derives new theoretical bounds to characterize the interaction between feature estimation models and downstream prediction models. Furthermore, the paper provides conditions under which sequential and iterative learning algorithms achieve theoretical consistency. Experimental results on real-world datasets empirically validate the theoretical claims.

**Claims And Evidence:**

The main theoretical claims regarding the generalization error bounds and conditions for consistency appear rigorously presented, but their practical significance heavily relies on certain mathematical assumptions (e.g., Lipschitz continuity and bounded loss functions). While the empirical experiments generally support the theoretical findings, it remains unclear how sensitive these theoretical conditions are to deviations in practical scenarios. It would be helpful if authors could clarify these assumptions' practical implications or robustness.

**Essential References Not Discussed:**

At present, no glaring omissions of essential references are apparent. However, further scrutiny by reviewers more familiar with continuous weak features or semi-supervised learning might identify additional relevant studies.

**Experimental Designs Or Analyses:**

I examined the experimental section, particularly Section 5, focusing on the generalization error analysis on real-world datasets. The experimental design effectively illustrates the proposed theoretical results about error convergence. However, the datasets used (Jets, Electricity, etc.) represent relatively simple, low-dimensional scenarios. Extending these evaluations to scenarios involving modern generative models, such as LLMs or diffusion models, would provide stronger evidence for the applicability and utility of the theory in contemporary, high-impact research domains.

**Methods And Evaluation Criteria:**

The proposed theoretical framework and methods for evaluating the error bounds seem appropriate and logically sound. However, the evaluation heavily relies on relatively simple datasets from OpenML. Considering the complexity of recent machine learning applications, additional evaluation using more complex or realistic scenarios (such as datasets involving generative models or high-dimensional data) would enhance the practical relevance and validation of the proposed framework.

**Other Comments Or Suggestions:**

- The paper is missing an explicit impact statement. Given the theoretical nature of the work, it would be helpful if the authors provided a discussion on the broader implications of their findings, including potential societal or industrial impacts. Clarifying how Continuous WFL could influence real-world machine learning applications would strengthen the paper.

**Other Strengths And Weaknesses:**

**Strengths**

1. Presents a unified theoretical framework clearly extending discrete WFL theory to continuous cases.
2. Provides rigorous theoretical conditions for sequential and iterative learning methods ensuring consistency.

**Weaknesses**

1. The practical implications of theoretical assumptions are not fully discussed, potentially limiting the practical applicability.
2. Experimental validations are restricted to relatively simple and low-dimensional datasets, lacking evaluation on more complex, high-dimensional tasks. Applicability to contemporary high-impact domains such as Large Language Models (LLMs) or Diffusion models is not explored, which could limit the perceived relevance of the proposed method in current research contexts.

**Questions For Authors:**

- Given that many modern AI models deal with weak or missing features (e.g., masked tokens in LLMs, noise in generative models), it would be interesting to explore how your framework could be extended or adapted to these domains. Would the theoretical results hold in high-dimensional feature spaces commonly seen in LLMs?
- What are the practical trade-offs between sequential learning (impute-then-predict) and iterative learning (joint training of feature estimation and prediction)? While the theoretical results suggest conditions for consistency in both learning paradigms, it is unclear in which real-world scenarios one approach may be preferable over the other.

**Relation To Broader Scientific Literature:**

The paper effectively relates its contributions to previous theoretical work on discrete weak features learning, clearly highlighting the gap addressed by this work. Additionally, it integrates related work effectively, such as imputation methods (e.g., impute-then-regress) and complementary features learning, positioning its contributions within an established body of literature.

**Theoretical Claims:**

Due to limited familiarity with the detailed mathematical underpinnings of the theorems presented, I could not thoroughly verify the correctness of all proofs provided. However, the logical structure and steps outlined seem consistent with standard practices in theoretical machine learning. A thorough peer check from a reviewer deeply familiar with continuous week feature learning theory would be beneficial.

---

> ### Author Rebuttal · Authors · 2025-04-01
>
> We sincerely appreciate the time and effort you dedicated to evaluating our work and are grateful for your insightful feedback and constructive suggestions.
>
>
> **Practical Implications and Robustness of Mathematical Assumptions:**
>
> We think that our assumptions, such as Lipschitz continuity and bounded loss functions, are standard in theoretical studies and not particularly restrictive. Many commonly used loss functions naturally satisfy these properties, ensuring the validity of our theoretical results in practical settings.
>
> While some loss functions, like MSE and 0-1 loss, do not strictly meet these assumptions, this is not a practical concern. MSE satisfies them when target data values are bounded, which is always the case in computational settings. The 0-1 loss is rarely used directly due to optimization challenges, and surrogate losses like hinge loss, which are Lipschitz continuous, are typically employed.
>
> Even if a loss function does not fully satisfy these assumptions, our framework remains practically relevant. Since it integrates existing learning methods for $f$ and $g$, an effective learning algorithm in standard supervised learning is likely to be effective in WFL as well.
>
> **Evaluation on More Complex and Realistic Scenarios:**
>
> We appreciate your suggestion. As you pointed out, evaluating the framework on high-dimensional or more complex data distributions would further strengthen its practical relevance and validation. A similar concern was raised by Reviewer wW1H (Experiments Conducted on Other Types of Data). In the response, we have conducted additional experiments using different types of WFs to demonstrate the framework’s effectiveness in a broader range of scenarios. These experiments have further enhanced its practical relevance.
>
> Theoretically, our framework is not constrained by data complexity. However, empirical validation on high-dimensional datasets remains essential. We acknowledge that such evaluations would provide deeper insights and help identify future research directions. In this study, we focused on establishing the effectiveness of our framework in fundamental settings. Nevertheless, we plan to explore evaluations with more complex datasets in future work.
>
>
> **Potential Applicability to High-Impact Domains (e.g. Diffusion Model):**
>
>
> Our theoretical analysis establishes properties that are independent of the specific choices of learning methods for $f$ and $g$, meaning our framework is flexible and can incorporate models such as Diffusion Models. This flexibility is a key strength, suggesting that our approach is well-suited for contemporary high-impact domains. The precise performance when integrating Diffusion Models and other domains remains an open and intriguing question. In this study, we focused on developing and analyzing a general framework, and exploring its application to such advanced models is an important direction for future work.
>
>
> **Explicit Impact Statement of our paper:**
>
> We have addressed a similar question from Reviewer wW1H. Please refer to our response (top 2 response for reviewer wW1H) there for details.
>
>
> **Applicability to Modern AI Models and High-Dimensional Feature Spaces:**
>
> As mentioned earlier, our framework can be applied to various modern AI models. This is because it focuses on integrating different methods for learning $f$ and $g$. Moreover, the derived error bounds provide a theoretical analysis of how such $f$ and $g$ influence each other when using state-of-the-art learning methods. For instance, our framework enables a detailed understanding of how improvements in $g$'s error affect the learning efficiency of $f$.
> Additionally, our theoretical results hold regardless of the data distribution as long as the data values are bounded. Therefore, our theory remains valid even in high-dimensional feature spaces, offering valuable insights.
>
> **Trade-offs Between Sequential and Iterative Learning:**
>
> Intuitively, iterative learning is expected to be more effective than sequential learning, albeit at the cost of increased training time. This advantage arises because iterative learning can enhance the accuracy of both $g$ and $f$.
>
> First, in iterative learning, the learning of $g$ benefits not only from the observed values of WFs but also from the information of $Y$ through $f$, potentially leading to lower error compared to sequential learning. Moreover, according to Theorem 4.2, such an improved $g$ contributes to a more accurate $f$, further reinforcing the benefits of iterative learning. However, the quantitative evaluation of this trade-off depends on the specific methods and datasets used, making it difficult to determine a generalizable conclusion without large-scale empirical validation.

---

> > ### Comment · Reviewer_dvad · 2025-04-09
> >
> > Thank you for updating. I will keep my score.

---

### Official Review · Reviewer_Ck1h · 2025-03-20

**Overall Recommendation:** 3

**Summary:**

This paper proposes a unified analysis framework of weak features learning, where part of the features are inaccurate. The paper analyzes generalization performance of a class of learning algorithms, in which a feature predictor $g_j$ is learned for all dimension of "weak feature", and a classifier $f$ is learned to make the final prediction. The authors analyze the generalization error of $f$ when $g$ is fixed, as well as that of $g$ when $f$ is fixed, demonstrating that weak features learning is feasible under some conditions.

**Claims And Evidence:**

Generally yes.

**Essential References Not Discussed:**

No.

**Experimental Designs Or Analyses:**

No.

**Methods And Evaluation Criteria:**

Yes.

**Other Comments Or Suggestions:**

Typos: Line 237~238, lack a colon ":"; Line 239, "for any..." → "For any..."

**Other Strengths And Weaknesses:**

Strengths: The problem formulation is clear and general, the symbols are standard, and the proofs are relatively easy to follow.

Weaknesses:

My main concern is that the results seems a bit too straightforward, thus lack of insight more than supervised learning. Specifically, Theorem 4.3 and 4.4 demonstrate that a good generalization performance can be obtained when the feature predictors $g$ and the classifier $f$ can both be learned well. I think the problem of weak features learning is not merely a process of learning the value of weak features via ordinary features plus a process of classical supervised learning.

Due to this concern, I wonder if the authors could conduct a clearer discussion that separates the following topics: (1) the main challenge of weak features learning, (2) how the current analysis resolve the challenge, (3) the limitation of current analysis due to the development of the supervised learning theory, and (4) the limitation of current framework due to the mathematical model itself.

I think a convincing discussion of above topics in the paper will change my evaluation, even if the result of discussion is rather negative (weak feature learning is feasible only if the features are actually not weak).

**Questions For Authors:**

The authors made several assumptions throughout the paper. While each of the assumptions has been justified, I still have some questions regarding the necessity and generality of some assumptions:

(1a) In line 172-173, the performance of feature learner is measured via MSE. I wonder if this is necessary for the subsequent analysis, or just assumed for simplicity.

(1b) In line 209-210, the randomized feature estimation model is assumed to be Gaussian. Is Gaussian a common model in real-world scenarios? Is Gaussian a necessary assumption, or just assumed for simplicity?

Besides, I also have the following questions:

(2) In Theorems 4.3 and 4.5, the upper bounds include a quadratic term of $U_l$, thus I wonder when would this bound be non-trivial (except the cases that either $f$ or $g$ has learned very well).

(3) Moreover, Theorems 4.3 and 4.5 present the theoretical guarantee when either $f$ or $g$ is \emph{fixed} and the other is optimized. I'm curious if there is any overall guarantee when $f$ and $g$ is both optimizable (i.e., alternatively update, or simultaneously update).

**Relation To Broader Scientific Literature:**

No.

**Theoretical Claims:**

The theoretical claims are generally correct. However, some of the error bounds are a bit too loose, making them unable to provide theoretical insights, which is particularly important as the paper is mainly about "unified analysis". Please see weaknesses for the detailed comment.

---

> ### Author Rebuttal · Authors · 2025-04-01
>
> We are thankful for your careful examination of our paper and for their helpful suggestions to improve the clarity and depth of our research.
>
> **Clarification of the Theoretical Contribution of Continuous WFL:**
>
> Thank you for your suggestion. Our main contribution is a unified framework for continuous WFL with theoretical analysis. We acknowledge its limitations and summarize them below:
>
> (1) the main challenge of weak features learning:
>
> The main challenge in WFL is that simply combining existing learning theories for $f$ and $g$ does not guarantee consistency and convergence rates in WFL. Specifically, the learning of $f$ depends on $g$, and vice versa, but supervised learning theory does not account for this interdependence. Consequently, prior theoretical frameworks could not explicitly analyze how errors in $g$ (or $f$) affect the learning efficiency of $f$ (or $g$).
>
> (2) how the current analysis resolve the challenge:
>
> To address this issue, our analysis explicitly models the interaction between $f$ and $g$. We first develop mathematical tools to capture this mutual dependence (Lemmas 4.1 and 4.2). Using these tools, we derive error bounds that illustrate how the risk of $g$ (or $f$) influences the convergence rate of the risk of $f$ (or $g$) (Theorems 4.3 and 4.5). These error bounds provide precise insights into how the consistency and convergence rates of WFL evolve based on the risk of $g$ (or $f$), as detailed in Sections 4.2 and 4.3. For distinctions between discrete WFL and our setting, please refer to our response to Reviewer wW1H.
>
> (3) the limitation of current analysis due to the development of the supervised learning theory:
>
> As you pointed out, our analysis builds upon existing supervised learning theory. Consequently, our framework does not account for the feature importance of WFs in the downstream task, which remains a limitation.
>
> (4) the limitation of current framework due to the mathematical model itself:
>
> Moreover, since our mathematical model explicitly separates $f$ and $g$, it does not theoretically address approaches that jointly learn them as a single model. Addressing these challenges is an important direction for future work.
>
> For (1) and (2), we will clarify the significance and positioning of our theoretical results in the discussions following each analysis. For (3) and (4), we will explicitly state these limitations as future work in the Conclusion.
>
> **Question (1a) The reason for adopting MSE:**
>
> We adopt MSE for both analytical necessity and simplicity. It allows us to directly apply existing error bounds for learning $g$, which typically use MSE, to Theorem 4.3, where the error bound of $f$ is also expressed in MSE-based risk. Additionally, MSE enables key mathematical tools, such as Lemma 4.2, which captures the interaction between $f$ and $g$ and facilitates our theoretical analysis of WFL. While similar analyses might be possible with other metrics like mean absolute error, we have not yet explored this direction in depth.
>
> **Question (1b) The Reason for the Assumption of Feature Estimation Models:**
>
> The Gaussian assumption for feature estimation models is made for both analytical necessity and simplicity. This assumption enables us to relate the risk of $f$ to the MSE-based risk of $g$. However this assumption is not necessarily unrealistic, as Gaussian noise is often added to deterministic outputs of $g$ to represent prediction uncertainty, and Bayesian models frequently yield Gaussian predictive distributions. Thus, while it simplifies analysis, it remains relevant in practical settings.
>
> **Question (2) Non-triviality of Our Generalization Bounds and the Condition:**
>
> Although the formulas are complex, Theorem 4.3 provides a non-trivial bound, as the RHS of Eq. (4.11) does not include second-order terms of $U_l$​. The first term is purely first-order, while the second involves products of ${U_l}^{1/2}$, leading to at most a first-order dependence.
>
> Regarding Theorem 4.5, as you noted, it includes a quadratic term in $U_l$​. However, this term is scaled by $\sqrt{\log(1/\delta) / 2n}$​, which becomes small when $n$ is large. Thus, the quadratic term’s overall impact remains limited, ensuring that Eq. (4.13) remains non-trivial in large-sample regimes.
>
> **Question (3) Applicability of Our Theoretical Bounds to Alternative or Simultaneous Update:**
>
> Theorems 4.3 and 4.5 are applicable to an alternative update. Theorem 4.3 provides a generalization bound for $f$ given any $g$, while Theorem 4.5 does so for $g$ given any $f$. During training, Theorem 4.3 applies when updating $f$ with a fixed $g’$, and Theorem 4.5 applies when updating $g$ with a fixed $f’$. This allows generalization error analysis at each step of an alternative update process.
>
> However, our current framework does not provide theoretical guarantees for simultaneous update, which remains an important direction for future work.
>
>
> **About Typos:**
>
> Thank you for pointing that out. We will correct it.

---

> > ### Comment · Reviewer_Ck1h · 2025-04-02
> >
> > I thank the authors for their detailed rebuttal. My concerns have been largely addressed, and I have no further questions. I have raised my score accordingly, pending the incorporation of the promised clarifications in the future revision.

---

> > > ### Author Response · Authors · 2025-04-02
> > >
> > > Thank you again for your valuable time and dedicated effort. We truly appreciate your thoughtful evaluation and the constructive feedback you provided throughout the review process.

---

### Official Review · Reviewer_wW1H · 2025-03-22

**Overall Recommendation:** 2

**Summary:**

This paper aims to provide a systemical theoretical framework for continuous weak feature learning (WFL). Previous studies focus on discrete WFL while neglecting the continuous weak features. Moreover, they have not addressed the fundamental questions such as the influence of feature estimation and label prediction models on each other, and the precise conditions for theoretical guarantees. Built upon these motivations, this paper proposes to construct a general and systematical theoretical framework for continuous WFL. Furthermore, the authors extend the framework with discrete WFL to construct a more general framework.

**Claims And Evidence:**

Weaknesses:
- The importance of continuous WFL is underexplored since the value of discrete WFL has not yet been acknowledged.
- Moreover, there is no adequate explanation of continuous WFL, as well as its difference from discrete WFL. It is better to provide some examples and their applications, otherwise this paper will be hard to follow.
- There is an over-claim regarding the weak features. The claim of weak features includes missingness, measurement errors, or ambiguous observations, while the experiments are only conducted on missing data.

**Essential References Not Discussed:**

N/A

**Experimental Designs Or Analyses:**

Strengths:
- The experimental results can support the proposed idea.

Weaknesses:
- The experiments are only conducted on learning with missing data.

**Methods And Evaluation Criteria:**

Strengths:
- The experimental settings and evaluation criteria are properly provided.

**Other Comments Or Suggestions:**

See above.

**Other Strengths And Weaknesses:**

Weaknesses:
- The reference for "Anonymous. A unified framework for generalization error analysis of learning with arbitrary discrete weak features" is confusing. This paper cannot be searched on the web now. How will you refer to this citation after your paper is accepted?

**Questions For Authors:**

See above.

**Relation To Broader Scientific Literature:**

Strengths:
- This paper is related to learning from missing data. The authors have adequately discussed the works in such areas.

**Theoretical Claims:**

Strengths:
- The proposed framework has appropriately analyzed the error bounds with feature estimation and label prediction models. The theoretical analysis makes sense.

Weaknesses:
- The hypothesis of random weak features is a bit weak. In practice, the weak features may be relative to the exact features or the observed features.

---

> ### Author Rebuttal · Authors · 2025-04-01
>
> Thank you for your insightful review and the detailed feedback, which will greatly help us enhance the quality of our paper.
>
> **The importance of continuous WFL:**
>
> Existing research on specific cases of WFL, such as ItR and CFL, has established them as a recognized research area. The discrete WFL framework, detailed in our supplemental material, provides a unified approach for handling various discrete WFs. However, numerous real-world scenarios involve continuous WFs, such as missing or noisy values, observational errors, or interval-based observations containing exact values. For instance, continuous WFs are caused by measurement errors or insufficient precision in sensor readings. Additionally, anonymized personal data where numerical attributes like age, blood pressure, or purchase frequency are provided as intervals. These cases frequently arise in medical and industrial applications, highlighting the significance of continuous WFL. Since discrete WFL alone cannot theoretically accommodate these scenarios, developing a framework for continuous WFL is crucial.
>
> **The theoretical difference between discrete WFL and continuous WFL:**
>
> As stated in the introduction, discrete WFL relies on the discrete nature of WFs and cannot be trivially extended to continuous cases. This is evident when comparing the proofs of Theorem 3.1 and Lemma 4.2 in continuous WFL with the proofs of Theorem 3.1 and Lemma 4.1 in discrete WFL, which employ fundamentally different approaches. Moreover, while previous studies have addressed individual cases of continuous WFs, no unified theoretical framework exists. Our work formulates and analyzes such a framework, making a significant contribution to the field. To better convey the importance of our research, we will expand the introduction in the Camera Ready version. Additionally, we will clarify the differences in proof techniques by adding further details in the Appendix.
>
> The theoretical significance of continuous WFL and its distinction from discrete WFL are discussed in our response to Reviewer Ck1h (Reply Section: Clarification of the Theoretical Contribution of continuous WFL). We kindly refer you to that section for further details.
>
> **Experiments Conducted on Other Types of Data:**
>
> While our experiments focus on missing data, the proposed framework can be applied to scenarios involving measurement errors and ambiguous observations as well. We are currently conducting additional experiments to evaluate the performance under these conditions. The preliminary results indicate that the experimental results conducted on other types of WFs remain qualitatively similar to the missing data case. We will include these experimental results in the Appendix of the Camera Ready version.
>
> **The hypothesis of random weak features:**
>
> We acknowledge that restricting the probabilistic model $q_g$​ for modeling the randomness of weak features (WFs) to a Gaussian distribution is a limitation of our approach. However, this theoretical hypothesis allowed us to derive rich analytical insights, such as the relationship between the risk of $f$ and the MSE-based risk of $g$. Extending beyond this assumption is an important direction for future work.
>
> Regarding the assumption that the variance $\sigma^2$ is constant, we clarify that it can be generalized to depend on observed ordinary features $X^\mathrm{o}$, i.e., $\sigma^2(x^{\mathrm{o}})$. In this case, the current error bound involving $\sigma^2$ remains valid by replacing $\sigma^2$ with $\max_{x^{\mathrm{o}}}\sigma^2(x^{\mathrm{o}})$. We did not explicitly state this in the paper and will clarify it in the Camera Ready version.
>
> **About Citation “Anonymous. A unified …”:**
>
> The cited paper is our previous work on discrete WFL, which is currently under review. Following ICML submission guidelines, we have anonymized it and included it in the supplementary material. Regarding the citation after acceptance, if the discrete WFL paper is accepted by then, we will update the reference accordingly. Otherwise, we will upload the discrete WFL paper to arXiv and update the citation with the corresponding information.

---

### Decision · Program_Chairs · 2025-05-01

**Decision:**

Accept (poster)

**Comment:**

This paper provides a unified theoretical framework for continuous weak feature learning, which is distinguished from previous work focusing on discrete weak feature learning. It also conducts experiments on real-world datasets to validate the effectiveness. The reviewers have raised concerns related to the practicality of theoretical assumptions, the intuitive theoretical results, and the relatively simple validation datasets. During the rebuttal phase, the authors have addressed most of the concerns of the reviewers. Although some concerns regarding the theoretical assumptions have not been completely addressed, the theoretical contribution of this paper is somewhat meaningful. The authors need to carefully revise their paper according to the comments, and include necessary evaluation experiments under more scenarios (According to their rebuttal to Reviewer wW1H, they have conducted additional experiments under measurement errors and ambiguous observations). Consequently, I recommend a weak acceptance of this paper.